

# Application of pore water stable isotope method to characterise a wetland system

Katarina David[1]; Wendy Timms.[1]; Cath E. Hughes [2], Jagoda Crawford[2]; Dayna McGeeney[3]

[1]School of Minerals and Energy Resource engineering, and Connected Waters Initiative, University of New South Wales, Sydney, Australia
[2]Australian Nuclear Science and Technology Organisation, Sydney, Australia
[3]Australian Museum, Sydney, Australia

*Correspondence to*: Katarina David (k.david@unsw.edu.au)





**Abstract**

Three naturally intact wetland systems (swamps) were characterized based on sediment cores, analysis of surface water, groundwater and porewater stable isotopes. These swamps are classified as temperate highland peat swamps on sandstone (THPSS) and in Australia they are listed as threatened ecological communities.

This study is the first application of the stable isotope direct vapour equilibration method in a wetland, enabling quantification of the contributions of evaporation, rainfall and groundwater to swamp water balance. This technique enables understanding of the depth of evaporative losses and the relative importance of groundwater flow within the swamp environment without the need for intrusive piezometer installation at multiple locations and depths. Additional advantages of the stable isotope direct vapour equilibration technique include detailed spatial and vertical depth profiles of δ18O and δ2H, with good accuracy comparable to the porewater compression technique.

Depletion of δ18O and δ2H in porewater with increasing depth (to around 40–60 cm depth) was observed in two swamps, but remained uniform with depth in the third swamp. Within the upper surficial zone, the measurements respond to seasonal trends and are subject to evaporation in the capillary zone. Below this depth the pore water δ18O and δ2H signature approaches that of groundwater indicating lateral groundwater contribution. Significant differences were found in stable pore water isotopes for samples collected after dry weather period compared to wet periods where recharge of depleted rainfall was apparent.

The organic rich soil in the upper 40–60 cm retains significant saturation following precipitation events and maintains moisture necessary for ecosystem functioning. An important finding for wetland and ecosystem response to changing groundwater conditions (and potential ground movement) are the observations that basal sands underlay the swamps, allowing relatively rapid drainage at the base of the swamp and interaction with lateral groundwater contribution.

Based on the novel stable isotope direct vapour equilibration analysis of swamp sediment, our study identified the following important processes: rapid infiltration of rainfall to the water table with longer retention of moisture in the upper 40–60 cm and lateral groundwater flow contribution at the base. This study also found, that evaporation estimated using stable isotope direct vapour equilibration method is more realistic compared to reference evapotranspiration (ET). Importantly, if swamp discharge data were available in combination with pore water isotope profiles, an appropriate transpiration could be determined for these swamps. Based on the results, the groundwater contribution to the swamp is a significant component of the water balance during dry period. Our methods could complement other monitoring studies and numerical water balance models to improve prediction of the hydrological response of the swamp to changes in water conditions due to natural or anthropogenic influences.



# 1 Introduction

The stable isotopes of water ($\delta^{18}$O and $\delta^2$H) have been widely used to understand the groundwater and surface water interaction and recharge processes in aquifer systems (Barnes and Allison, 1988; Cuthbert et al, 2014). The understanding of stable isotope variation and signatures in subsurface allows the identification of flowpaths (Person et al, 2012), hydrological processes on a

catchment scale (Rodgers et al., 2005; Vitvar et al., 2005), climate change (Huneau et al., 2003; Tadros et al., 2016) and aquifer heterogeneity (Hendry and Wassenaar, 2009). Although less common than liquid water isotope studies, pore water (vapour) stable isotope techniques have been applied to investigate the groundwater flux and interpret the paleoenvironment (Hendry et al, 2013; Harrington et al, 2013), determine slope runoff contribution to groundwater (Garvelmann et al, 2012) and characterize muliti-layered sedimentatry sequence (David et al, 2015). Pore water stable isotope analysis was less common

due to sampling difficulties (Sodenberg et al, 2011) and high cost (Harrington et al, 2013), until advances in laser spectroscopy improved the speed and accuracy of the analysis (Wassenaar and Hendry, 2008; Hendry et al, 2015).

International examples of wetland research indicate that it remains challenging to quantify some components of the water balance (Bijoor et al., 2011), since only a few studies have investigated the groundwater contribution (Hunt et al., 1996). The application of stable isotopes $\delta^{18}$O and $\delta^2$H is typically limited to surface water isotopes to understand the hydrology of the

swamp system (Nyarko et al., 2010; Bijoor et al, 2011), the effect of transpiration (Wang and Yakir, 2000) and to improving the water balance study (Schwerdtfeger et al., 2014; Levy et al, 2016). Recent international research on wetland and lake systems focuses on stable water isotopes and environmental tracers (Mandl et al., 2017; Meier et al. 2015; Kaller, et al. 2015) to understand paleoclimate and processes in these systems.

The significance of groundwater in maintaining the function of swamp ecosystem has been discussed in international literature

(Chang et al, 2009; Kaller et al, 2015). In Australia, swamp studies have evaluated geomorphology (Fryirs et al, 2016; Cowley et al, 2016), management (Kohlhagen et al, 2013), relationship between vegetation and groundwater (Hose et al, 2014), the processes that result in denudation and sedimentation in the headwaters of the swamps (Prosser et al, 1994), natural and anthropogenic vegetation change in swamps (Bickford and Gell, 2005) and the impact of mining subsidence (CoA, 2014b). However, there is limited literature on the importance of groundwater storage, flow and source of water to maintain moisture

in swamp systems.

The Temperate highland peat swamps on sandstone (THPSS) swamps in Eastern Australia are endangered ecological communities with endemic flora and fauna that are dependent on hydrological balance. The direct influences on the water regime of these swamps are changes in weather patterns, natural storm activity (Smith et al. 2001), fire (Middleton and Kleinebecker, 2012; CoA, 2014a) and the effects of mining subsidence (CoA, 2014b). Ecologically, the THPSS swamps are

sensitive to changing swamp moisture content (CoA 2014a; Young, 2017), and the importance of groundwater in these systems has been discussed by Eamus and Froend 2006, Fryirs et al, 2014; Hose et al, 2014.

Based on the substantial direct and indirect evidence of saturation within the TPHSS swamps (Newnes Plateau shrub swamp) such as vegetation patterns, piezometer records, presence of certain plant species, age dating (Benson and Baird, 2012) and





spring discharge (Johnson, 2006), there is a clear indication that the maintenance of groundwater levels and groundwater discharge to swamps is necessary for the health of such wetland system (Clifton and Evans, 2001).

Despite the awareness of these factors, there is limited research to predict swamp behaviour and ecological response under changing water conditions (Mitsch and Gosselink 2000; Cowley et al, 2016). Furthermore, studies describing the impact of

environmental changes on swamp ecology are rare and there is insufficient understanding of natural variation in swamp ecology over time (CoA, 2014a).

Long drought periods in Australia result in temporary drying of water bodies, pooling of water and reduction in baseflow contribution to wetlands (Lake, 2003). Following such extreme settings, the rainfall may not be sufficient for a swamp to recover its original condition (Bond et al, 2008; Middleton and Kleinebecker, 2011). For example, Smith et al. (2001) report

loss of swamps in Africa and Australia as a result of climate change. Vegetation removal, drainage of swamp and undermining are known to be critical human impacts (Kohlhagen et al, 2013; Valentin et al, 2005). As such, mining and urbanisation have degraded and considerably damaged the TPHSS swamps (CoA, 2014b), although the actual impact often cannot be quantified due to limited baseline and monitoring data (Paterson, 2004). However, it is generally recognized that rock fracturing, changes in elevation gradient and catchment conditions can compromise the stability and integrity of these swamps (CoA, 2014b).

The objective of this research was to improve understanding of intact swamps under natural conditions by characterizing the sediments, waters and organic materials. For this we investigate for the first time using direct equilibration method the vertical profiles of stable $\delta^{18}O$ and $\delta^{2}H$ isotopes of pore water within the swamp. We then compare those to stable isotopes of groundwater, rainfall and surface water as distinct endpoint members. Along with logs of sediment lithology, organic and carbon content of sediments these stable isotope results enabled the development of a conceptual model of the swamp water

cycle.

## 2. Site description

The research site is located west of Sydney, NSW, Australia, in the World Heritage listed Blue Mountains on the Newnes Plateau (**Figure 1**) between Lithgow and Blue Mountains local government areas. The elevation of the plateau ranges from 1000-1200 m Australian Height Datum (mAHD).





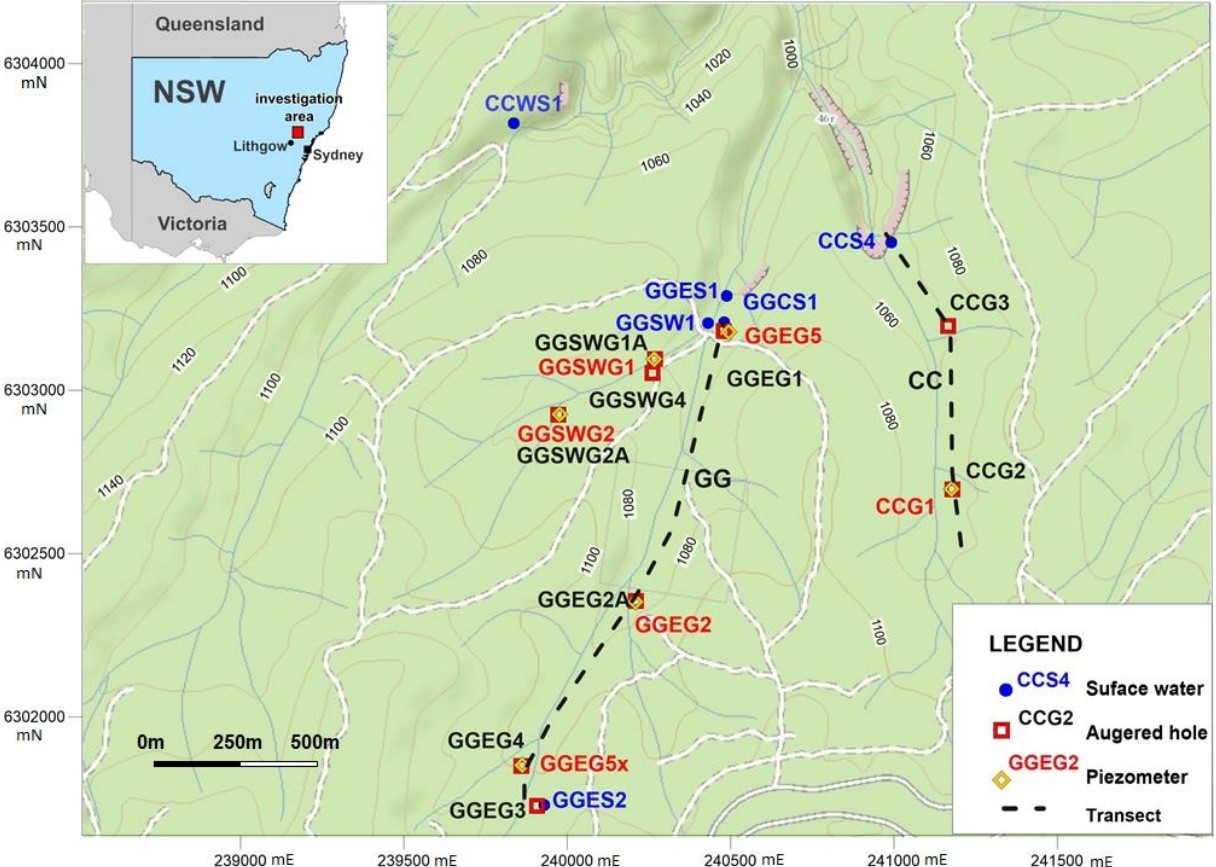

**Figure 1: Map of selected Newnes Plateau swamps with location of samples and transects.**

The Triassic Narrabeen Sandstone outcrops over most of the study area. It comprises mainly quartzose sandstone and minor claystone and shale (Yoo et al, 2001). The swamps on the Newnes Plateau are classified as shrub swamps (OEH, 2017) based

on the dominant shrub ecological community, and the highest elevation of any sandstone based swamp in Australia. These swamps occur in low slope headwaters of the Newnes Plateau as narrow and elongated sites with impeded drainage (OEH, 2017) and are also classified as TPHSS belonging to both headwater and valley infill types (CoA, 2014a). Mapping by Keith and Benson (1988) and Benson and Keith (1990) indicates that the shrub swamps cover 650 ha of land on the Newnes Plateau, with the largest swamp being 40 ha and average size less than 6 ha. Keith and Myerscough (1993) relate the swamps to other

upland swamps in the Sydney Basin in terms of biogeography. However, the difference from other TPHSS is the presence of a long-term permanent water table (Benson and Baird, 2012).

The three swamps selected: identified as CC (swamp area 7 ha, catchment area 150 ha), GG (swamp area 11 ha, catchment area 190 ha) and GGSW (swamp area 5 ha, catchment area 57 ha), are in the upper Carne Creek catchment. Carne Creek is a tributary of the Wolgan River (catchment area 5310 ha) that ultimately flows to the Hawkesbury River and Pacific Ocean. The

15 three swamps selected thus have a total area approximately 0.043% of the Wolgan River catchment. Except for the headwaters



including these swamps, the Wolgan River has been designated part of the Colo Wild River area recognizing substantially unmodified conditions and high conservation value (NSW Government, 2008). They are located to the north and west of pine plantation that was recently clear-felled, however this forestry activity is located in the catchment of other swamps that were not part of this study. The swamps in this study are located above and to the east of the current location of underground mining operations. The swamps are elongated with gentle gradient and typically terminate with a sandstone rockbar. The groundwater level in the swamps responds rapidly to rainfall recharge (Centennial Coal, 2016) and there is indication that swamp systems are fed by groundwater discharge (Benson and Baird, 2012).

The climate on the Newnes Plateau is temperate with higher rainfall in November to March and lower rainfall from April to October. Average yearly rainfall at the closest long term meteorological station in Lidsdale (Bureau of Meteorology (BoM) station 63132, 12 km west of the study area) is 765 mm (890 mAHD) and 1270 mm at Mt Wilson (BoM station 63246 21 km SE of the study area) (1010 AHD). The temperature varies from an average 19.6°C in summer to 5.8°C in winter (Lidsdale).

## 3. Methods

### 3.1 Fieldwork and sampling

The fieldwork was undertaken during 2016 and 2017 with the swamps in a natural state and recovered from earlier wildfire in 2013. The first sampling event 24th to 25th May 2016 occurred following an extremely dry weather period of three months below the long-term average rainfall (BoM Lithgow station 0630132, 900 m AHD, 13 km SW of the study area with 139 years of data records) for February to April. A repeat sampling on 25th to 26th October 2016 occurred after four months of above average rainfall from June to September. Sampling on 30thMay 2017 occurred under different climate conditions with both above and below average rainfall trend in the months preceding the sampling event. **Figure 2** shows the variation in monthly long term rainfall (139 years) and comparison with rainfall in 2016 and 2017.





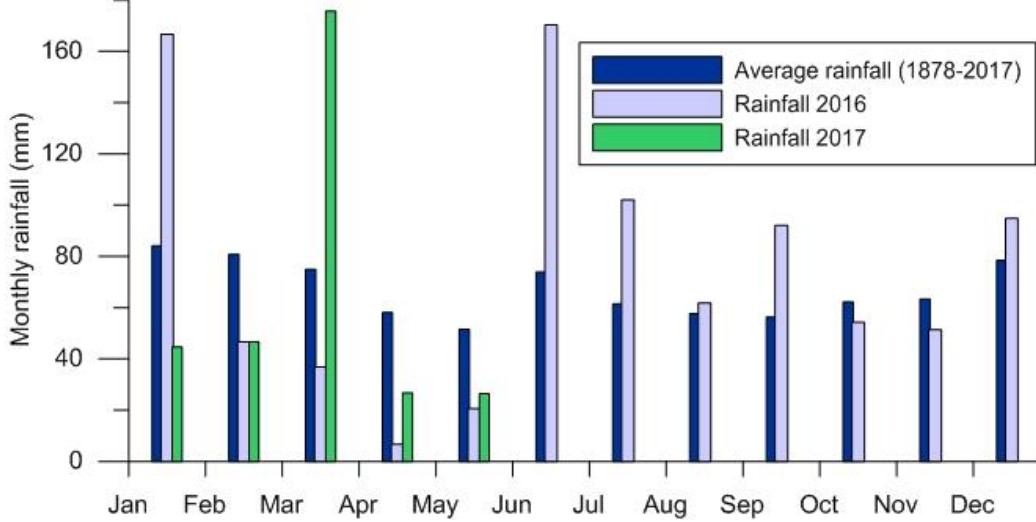

**Figure 2: Long term average monthly rainfall at Lithgow station (Bureau of Meteorology Station No. 063226) compared to rainfall during 2016 and 2017.**

In total seven sediment cores were obtained by coring using hand auger (40 mm diameter) to rock refusal (between 0.45 to 1.4 m), and two transects (CC and GG) were prepared along the length of two swamps (**Figure 1**) The hand augered holes were restored by returning excess material to the hole immediately after sampling. The coring on CC transect was repeated in October 2016 at a distance of less than 0.5 m from the original hole. The coring locations were selected to represent the swamp stratigraphy from upstream to downstream and to provide a spatial coverage across the three swamps. In addition, three augered

locations were selected such that they were adjacent to an existing piezometer (CCG1 on transect CC and GGEG2A and GGEG4 on GG transect). The purpose of this, in addition to determining the stable isotope profiles, was to enable comparison with the groundwater level measurements and to collect water samples from the underlying sandstone aquifer where possible. The nature and lithology of the augered core weredescribed in the field. Sediment cores were divided into subsamples of 10-20 cm length, were packed into Ziplock bags and kept in cool storage for later analysis of stable isotope composition of pore

water, moisture content and organic matter content. The samples for pore water analysis were temporarily double packed in ziplock bags by minimising the airspace in the bag, stored in the cooled ice box and vacuum packed the same afternoon after collection.

Groundwater was sampled directly from the augered hole, field parameters were measured immediately (pH, EC, DO, temperature) and samples field filtered (0.45 micron). This was repeated for all three sampling events; however, some bores

were dry and some not accessible. Groundwater from existing piezometers (CCG1, GGEG2, GGEG5x, GGEG5 and GGSWG1) was gauged and sampled by bailing three volumes and then the same procedure was followed as for the augered holes. Surface water samples were collected at the downgradient end of the swamp but also at one upgradient location (GGES2) where this was possible.





For this study ANSTO provided event based $\delta^{18}O$ and $\delta^2H$ for precipitation from Mt Werong for the period covered in this research. Mt Werong (Hughes and Crawford, 2013) is located around 70 km south of this research site, however, within the same climatic environment and at similar elevation to the investigated swamps.

## 3.2 Sample analysis

The swamp sediment samples were analysed for $\delta^{18}O$ and $\delta^2H$ by $H_2O_{(water)}$-$H_2O_{(vapour)}$ porewater equilibration (Wassenaar *et al*, 2008) and off-axis ICOS. The Los Gatos (LGR) water vapour analyser (WVIA RMT-EP model 911-0004) located at UNSW, Australia was used for sample analysis. Samples (n=54) were prepared by inflating the 1 L sample bags with dry air and allowing vapour equilibration over a period of between 17 to 24 hours. Following the equilibration period condensation

was noted on the side of some bags. More importantly the required 23,000 to 28,000 ppm $H_2O$ was present in the headspace allowing for accurate sampling (Hendry et al, 2015). It is therefore considered that the sampled vapour represents the water stored in the macropore space. The sample was collected by perforating the bag containing sample with sharp needle and transferring the pore water vapour sample directly from the bag via plastic tube to the LGR vapour analyser.

The analysis of the vapour sample was undertaken along with the standards prepared in the similar manner to the core samples.

The equilibration time for standards was around 20 minutes. Three standards were run after every third sample. $\delta^{18}O/\delta^2H$ vapour values were initially corrected using the fractionation factor to equivalent of pore water sample. The fractionation factor was estimated considering the equilibration temperature for conversion from vapour to liquid stage (Majoube, 1971). The readings were then corrected with two secondary $\delta^{18}O/\delta^2H$ standards (Los Gatos 2A -16.14‰ $\delta^{18}O$ and -123.6‰ $\delta^2H$ and 5A -2.8‰ $\delta^{18}O$ and 9.5‰ $\delta^2H$) and one primary SMOW standard run during the analysis.

Replicate sample analyses (mean difference of 6 samples) indicate reproducibility of results within 0.68‰ $\delta^2H$ and 0.04‰ $\delta^{18}O$ uncertainty. Reported instrument precision of 0.5‰ $\delta^2H$ and 0.15‰ $\delta^{18}O$ over 10 seconds and drift of 0.75‰$\delta^2H$ and 0.3‰ $\delta^{18}O$ over 15 minutes was minimised by correcting the readings. The data set for each sample was corrected for drift by back correction of standards within each set and then applying the same regression analysis to the relevant samples. For each sample the standard deviation and instrument drift error were calculated.

Water samples (surface water and groundwater, n=21) were analysed for $\delta^{18}O$ and $\delta^2H$ by isotope ratio mass spectrometry (IRMS) using LGR analyser located at UNSW Australia. Two LGR standards and VSMOW standard were used to correct the samples.

Gravimetric water content (ASTM D2974-14, 2014 and ASTM D2216-10, 2010 was measured by weighing the sample (n=70), drying at 100°C for 24 hours and re-weighing (Reynolds, 1970). The analysis was undertaken at the School of Mining

Engineering, UNSW Australia. Organic matter content was measured by loss on ignition method (LOI), by weighing (following initial drying at 100°C) and drying in furnace oven at 550°C. The analysis was conducted at the Water Research Laboratory, UNSW Australia.



Precipitation samples were analysed at the ANSTO Environmental Isotope Laboratory using a cavity ring-down spectroscopy method on a Picarro L2120-I Water Analyser (reported accuracy of ±1.0, ±0.2‰ for $\delta^2H$ and $\delta^{18}O$ respectively). The lab runs a minimum of two in-house standards calibrated against VSMOW/VSMOW2 and SLAP/SLAP2 with samples in each batch.

## 4. Results

5  ### 4.1. Stratigraphy, organic matter and moisture content

Four stratigraphic units are recognised along both Newnes Plateau swamp transects CC and GG (**Figures 3 and 4**), similar to a general classification derived by Fryirs et al (2014) for TPHSS in Blue Mountains and Southern Highlands region. These units are typically from the base upward medium to coarse sand, medium sand to clayey sand, silt to sandy clay and organic rich soil (sandy) at the top.

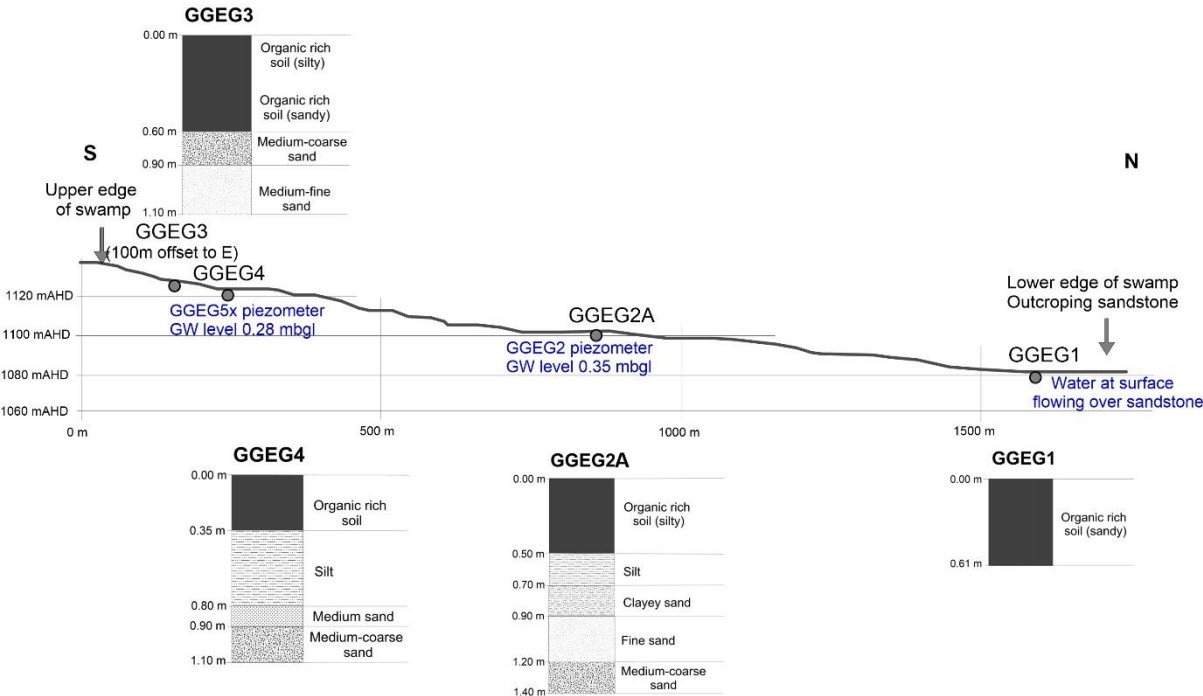

10  **Figure 3: Interpreted long–section of swamp GG with groundwater levels as measured in May 2016.**





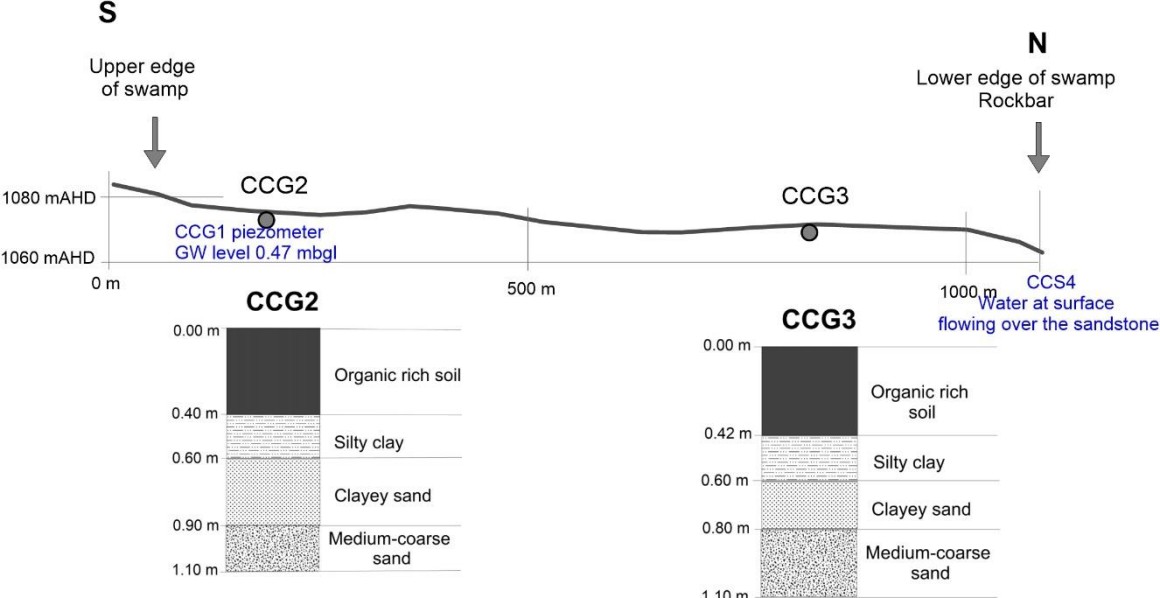

**Figure 4: Interpreted long–section of swamp CC with groundwater levels as measured in May 2016.**

The base of the swamp is comprised of quartz sandstone, the Banks Wall Sandstone of the Narrabeen Group. The alluvial sands (with sub-angular quartz grains) overlying the sandstone are off-white medium to coarse grain sands with occasional

quartz grains up to 2.5 mm in diameter. These sands are overlain by medium sand grading to fine sand in the GG transect, with 15% organic matter and a minor clay component. However, in the CC transect this layer is missing and sand transitions upwards to clayey sand with iron staining. The total thickness of these two sandy units varies from 10 to 50 cm increasing in downgradient direction. At the most downgradient site on GG transect the sand layer is absent. Typically, the basal sand is overlain by a silt and silty clay that is thickest in the middle of the swamp (20–45 cm). The silt is dark in colour and contains

approximately 40% organic matter with strong organic smell. The uppermost unit is an organic rich soil or peat (20-60 cm thick), occasionally silty with abundant roots.

The groundwater is shallow, and its level varied in piezometers (installed to 1.5 m depth) in May 2016 from 0.35 m below ground level (bgl) in GGEG2 to 0.47 m bgl in CCG1 (Figures 3 and 4). The groundwater level in augered holes was similar to that in shallow piezometers, however there was a significant difference represented by a rise of up to 0.4 m at all measured

locations following the wetter period. During this wetter period, groundwater levels recorded at GGSWG1 and GGEG2 were 0.05 m bgl and 0.09 m bgl, respectively. No overland flow was observed at any time, and the swamps did not have a formed channel. The only surface water observed in the swamps was at the lower edge of the swamp and flowing over the rockbar.

The swamp sediments are variably saturated, with gravimetric water content measurements exceeding 100% weight (dry mass basis) in the top 30 cm. This is typical for high organic matter proportion (GG samples) (**Figure 5**). It should be noted that

100% gravimetric water content relates to water holding capacity and organic content of the material. Within the same vertical profile, the organic matter content varied with depth and decreased from 60% to 10%. At a depth from 60 to 120 cm the





gravimetric water content decreased to an average of 17% for CC and 32% for GG swamp during both May and Oct 2016 sampling periods. The average organic matter decreased to 3.7% for all swamp locations below 80 cm depth.

During May 2016, following the dry period, upgradient and downgradient samples in CC swamp had similar gravimetric water content. A clear distinction was observed after wet weather period between the upgradient CCG2, having overall lower

5 gravimetric water content, and downgradient CCG3, with higher gravimetric water content. This difference can be explained by higher permeability in the upgradient part of the swamp resulting in quicker drainage, and/or increased groundwater contribution in the lower part of the swamp. A trend with an increase in moisture content downstream has been observed in all three swamps. However, at GGEG, the undulating topographic gradient means that changing moisture conditions exist along the length of the swamp. An overall increase in moisture content to around 80 cm depth in CCG3, was also recorded following

10 the wet weather period although the increase was not statistically significant (p>0.05).

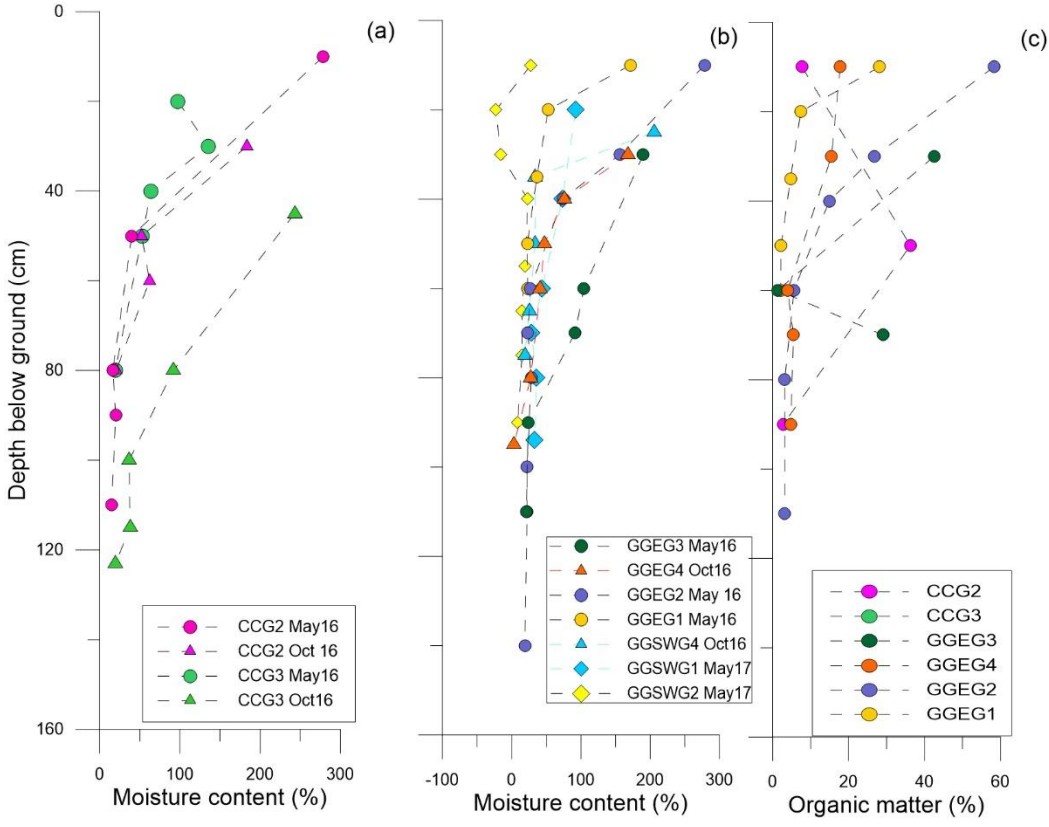

**Figure 5: Gravimetric water content (% weight) for CC (a), GG and GGSW (b) and organic matter content in CC and GG swamps (c) shown with depth.**





## 4.2 Stable isotopes of water and pore water

The relationship between the $\delta^{18}O/\delta^2H$ of surface water, groundwater and swamp pore water is presented in **Figure 6**. This figure also shows the local meteoric water line (LMWL) for Lithgow ($\delta^2H=7.99\delta^{18}O+16.6$; Hughes and Crawford, 2013) and weighted rainfall average for Mt Werong which is based on the past 12 years of data ($\delta^{18}O=-6.9$, $\delta^2H=-37$). The $\delta^{18}O$ of rainfall varies seasonally with higher values in summer (0 to 5‰) and lower in winter (-6 to -10‰).

Stable isotope data from precipitation events at Mt Werong were plotted (excluding the rainfall below 5 mm) for three periods (Jan to May 2016, May to October 2016, and Jan to May 2017). The stable isotope data for these events plot on or close to previously defined LMWL for Lithgow (note that the LMWL for Mt Werong of $\delta^2H=8.08\ \delta^{18}O+16.6$ (Hughes and Crawford, 2013) has a similar slope but higher intercept than that for Lithgow.





**Figure 6: Stable δO18/δ2H composition of surface water, groundwater, swamp pore water, and local rainfall for Lithgow (Hughes and Crawford, 2013) May 2016 (a), October (2016) (b) and May 2017 (c).**

For May 2016 with dry and warm antecedent conditions, pore water stable isotope ranges were -7.2 to 3.1‰ δ¹⁸O and -45.7 to -22.3‰ δ²H (Figure 6a). The regression lines for pore water samples at CC and GG in May 2016 deviated from the Lithgow LMWL along local evaporation lines with slopes of 4.2 and 4.6 respectively. Two major rainfall periods (27 mm two weeks prior to May 2016 sampling and 153.5 mm in January 2016) had no noticeable influence on the swamp pore water isotope composition. The intersection points of the regressed trend lines of pore water and LMWL plot within the depleted δ¹⁸O and δ²H rainfall range. The swamp pore water is more negative and therefore likely to be from bigger, more depleted events in the autumn and winter of the prior year (including 230 mm in April 2015: -7.6‰ δ¹⁸O, -39.2‰ δ²H; 108 mm in August 2015: -





9.8‰ $\delta^{18}$O, -61.6‰ $\delta^{2}$H; and two smaller but highly depleted events in June and July). This agrees with annual weighted averages at Mt Werong of -34.9 and -46.5‰ $\delta^{2}$H in 2015 and 2016, respectively.

Stable isotopes for swamp pore water collected in October 2016 (Figure 6b) ranged from -4.5 to -7.5‰ $\delta^{18}$O and -25 to -47‰ $\delta^{2}$H. A major rainfall event in June 2016 (92.8 mm at Lithgow and 109 mm at Mt Werong, -17.7‰ $\delta^{18}$O, -126.3‰ $\delta^{2}$H) had

not obviously affected swamp pore water, so it seems unlikely that two smaller rainfall events three weeks prior (25 mm and 17 mm) to sampling could have had an influence. Pore water stable isotope values from samples collected in October 2016 with wet and cool antecedent conditions, plot along the LMWL very close to weighted rainfall average. This is consistent with a winter rainfall signature.

The pore water samples collected in May 2017 from GGSW swamp lie along a slope of 6 which agrees with wetter period in

early 2017 compared to 2016. Such lower slope indicates evaporation trend in air where relative humidity is low (Gat, 1996). Samples from CC swamp collected in May 2017 are more enriched in $\delta^{2}$H (i.e. have a higher D-excess (*d*), defined as $d= \delta^{2}$H-8$\delta^{18}$O (Dansgaard, 1964)) than previously collected samples. Rainfall samples for bigger rainfall events in the period from December 2016 to May 2017 plot along the LMWL, except the events in the April prior to the 2017 sampling which have a significantly higher D-excess (*d*=24.5, 68 mm).

The pore water samples collected in May 2017 from GGSW swamp lie along a slope of 6 which agrees with wetter period in early 2017 compared to 2016. Such lower slope indicates evaporation trend in air where relative humidity is low (Gat, 1996). Samples from CC swamp collected in May 2017 are more enriched in $\delta^{2}$H (i.e. have a higher D-excess (*d*), defined as $d= \delta^{2}$H-8$\delta^{18}$O (Dansgaard, 1964)) than previously collected samples. Rainfall samples for bigger rainfall events in the period from December 2016 to May 2017 plot along the LMWL, except the events in the April prior to the 2017 sampling which have a

significantly higher D-excess (*d*=24.5, 68 mm). The porewater returned to the LMWL between May and October 2016, and shifted to the left of the LMWL for the May 2017 sampling.

Although the same rainfall events generally affect both Mt Werong and Newnes, and occur at the same time, the amount of rainfall at Newnes is typically smaller compared to Mt Werong. Whilst we would expect that larger rainfall events would lead to the most significant infiltration and recharge of groundwater, and therefore influence the porewater signature more, the data

seems to suggest that small recent rainfall events are very important in October 2016 and May 2017, following the wetter conditions experienced in the second half of 2016 and early 2017.

The importance of smaller rainfall events on recharge is also consistent with gravimetric water content data which remained stable throughout the wetter and drier period at depth below 0.8 m in CC and 0.6 m in GG and GGSW swamps. Another contributing factor may be that groundwater provides a moderating effect particularly during wetter periods, reducing the

effects that evaporation has on porewater isotope composition.

Groundwater samples collected in October 2016 and May 2017 are enriched relative to rainfall weighted average for Mt Werong (2005-2017). Surface water samples collected mainly at the downstream point of the swamp plot close to the LMWL and are depleted relative to pore water samples for $\delta^{18}$O, and $\delta^{2}$H, and relative to large rainfall events preceding the sampling event.




Surface water samples (-6.5 to -7.7‰ $\delta^{18}$O and -37.0 and -44.4‰ $\delta^2$H) plot within the range of $\delta^{18}$O and $\delta^2$H for groundwater (-6.2 to -7.9‰ $\delta^{18}$O and -32.4 and -44.7‰ $\delta^2$H). The statistical significance of the difference between the isotopic composition of surface water and swamp groundwater on both GG and GGSW transects was analysed by comparing the means of $\delta^{18}$O and $\delta^2$H (October 2016 and May 2017) for these two datasets using a *t*-test. Based on the mean we test the hypothesis that there is no statistical difference between the datasets (surface water and swamp groundwater). The calculated *p*-value was significantly more than 0.05 (for $\delta^{18}$O p=0.34 and for $\delta^2$H p=0.27;(n=20), indicating that the null hypothesis cannot be rejected and there is no significant difference between these two datasets. So, the reason for similarity of surface and groundwater samples is assumed to be short infiltration time to water table and mixing with lateral regional flow.

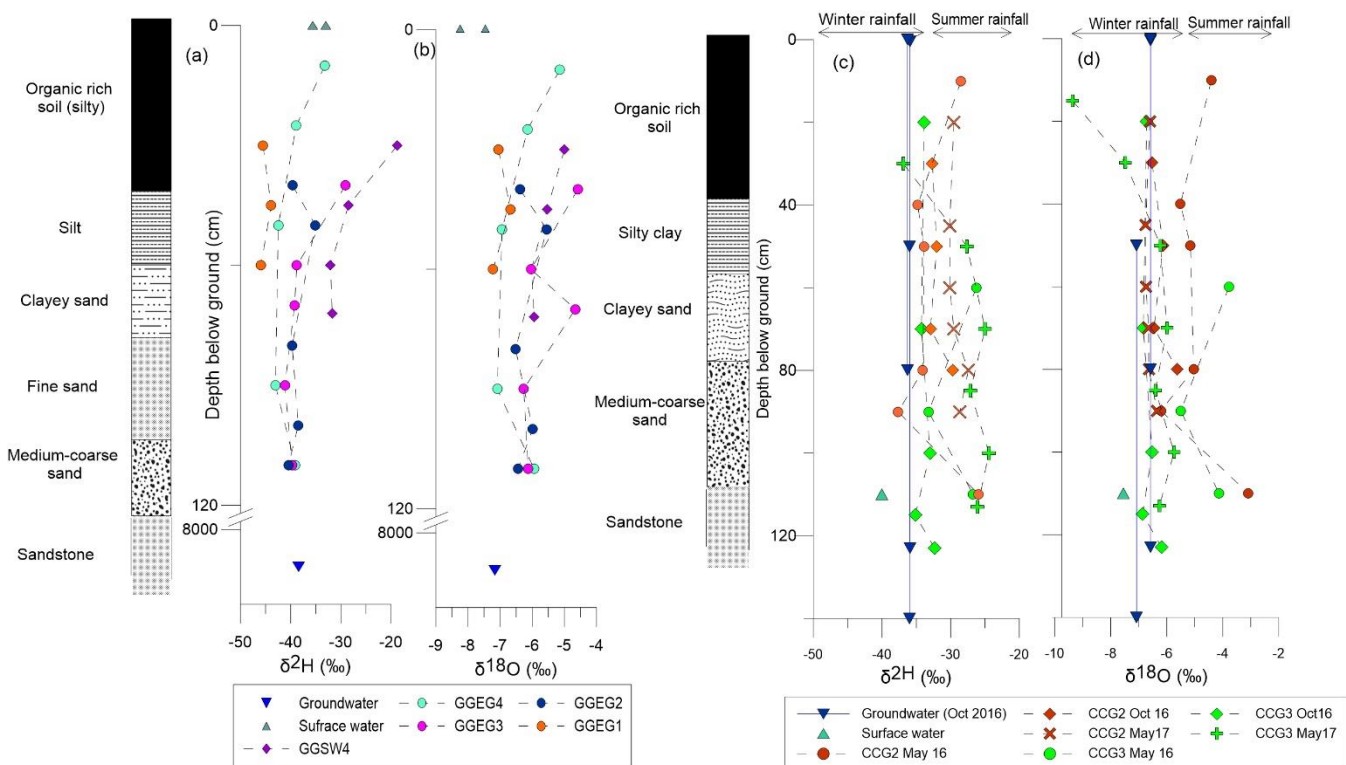

**Figure 7: $\delta^{18}$O and $\delta^2$H variation with depth in GG and GGSW swamps (May 2016) with typical lithology log. Groundwater sample represents groundwater flow in sandstone collected at the downstream point of the GG swamp (a and b). $\delta^{18}$O and $\delta^2$H variation with season and depth in CC swamp (May and October16 and May17) with typical lithology log. Groundwater represents cumulative water through the swamp within shallow piezometer and augered hole (c and d). Groundwater was not collected at all locations in May 2016. Depth of augured holes was not exactly the same in all sampling events.**

The $\delta^{18}$O and $\delta^2$H data for pore water are plotted with depth along with surface water and groundwater from GG and GGSW swamps (**Figure 7a and 7b**). Seasonal pore water and groundwater variations (May and October 2016 sampling) for CC swamp are compared to rainfall isotopic signature collected at Lithgow (Hughes and Crawford, 2013) (**Figure 7c and 7d**). The $\delta^{18}$O values of pore water (May 2016) in GG and GGSW swamps (**Figure 7**) show a tendency of depletion with depth





with greater variability at a depth of 40-65 cm. Below 100 cm depth, the $\delta^{18}O$ of pore water approach the swamp groundwater and groundwater signature in the underlying sandstone.

It can be observed that pore water samples from CC swamp from both upstream (location CCG2) and downstream (location CCG3) become depleted after longer wet and cool antecedent conditions with a $\delta^{18}O$ shift of around 1–3 ‰ (Figure 7c and 7d). $\delta^{18}O$ and $\delta^{2}H$ for pore water at CCG2 location during May and October 2016 has statistically significant difference between the wet and dry period ($\delta^{18}O$ (p=0.003) and $\delta^{2}H$ (p=0.02)), similar to $\delta^{18}O$ of pore water at CCG3 (p=0.01). The CC samples collected in May 2017 have similar $\delta^{18}O$ of pore water to October 2016 samples however slightly more depleted due to significant rainfall in March 2017.

Groundwater samples collected from piezometers screened across both top of sandstone and the base of swamp sediments (CCG1 and GGEG2) had similar $\delta^{18}O$ signature as pore water at a depth below 110 cm. Surface water $\delta^{2}H$ for October 2016 is more negative than pore water (-37.7‰ $\delta^{2}H$) in the upper 70 cm and is a reflection of typical winter rainfall signature.

## 4.3 Water and mass balance

During dry periods swamp pore water is subject to evaporation and becomes isotopically enriched. Therefore, the fractional loss of water through evaporation can be quantified if other water loss processes do not isotopically fractionate (Gonfiantini, 1986) or/and if the stable isotope composition of inflow and outflow and site weather data is known (Lawrence et al, 2007). To evaluate the evaporative losses based on isotopic composition of water, we used the Barnes and Allison (1988) analytical model to represent the change in isotopic profile in unsaturated soils due to evaporation. This model, based on deterministic approach, was selected because of the fact that the stable isotopes diffusivities very slowly with water content  and a relatively good agreement is reported with experimental results (Barnson and Allison, 1988; Shanafield et al, 2015).  The disadvantage of using the soil profile to estimate the evaporation is that the assumption of the steady state needs to be made and there is some uncertainty in dispersivity and tortuosity (Shanafield, 2015).

We applied the model to pore water data from all three sampling periods considering realistic input variables into the model as given in Table 1. The model was run with evaporation factor adjusted such that it matched the observed data; all other parameters remain constant.

Volumetric water content was calculated from the measured gravimetric water content (Figure 5) and bulk density. Bulk density was obtained from known lithology and measured data (Cowley et al, 2016) and porosity data from a swamp study by Walzsak et al, 2002. To estimate effective liquid diffusivity of isotopes, tortuosity values were obtained from literature (Maidment, 1993; Shackelford and Daniel, 1991; Barnes and Allison, 1988). A linear relationship was identified between particle size and tortuosity, and the final estimated tortuosity values are given in **Table 1**.

**Table 1: Input variables for the unsaturated Barnes and Allison (1988) model**





The results for unsaturated soil modelling at all sampled depth points based on $\delta^{18}O$ and $\delta^2H$ indicate an evaporative loss in the unsaturated zone of 4 mm to 9 mm/day in May 2016 (dry) period, and <1 mm/day for the wetter and cooler period between May and October 2016. Evaporation of less than 1 mm was estimated in CC swamp in both wet and dry periods, and at the upstream point on GGSW swamp.

5    The model was not sensitive to temperature, modelling at both 21.9°C and 10°C resulted in only minor differences in evaporation (<0.04 mm/day). Model results for drier and wetter periods are presented in **Figure 8**. The data for the May 2016 period (dry) shows a clear evaporative enrichment profile towards the surface (upper 0.4 to 0.6 m) and uniform $\delta^2H$ with depth (**Figure 8a and 8b**). The enrichment at surface is related to evaporation, but at depth fractionation is not occurring. During the wet period (**Figure 8c and 8d**) the $\delta^2H$ depletion at the surface is related to a big rainfall event, 10 days before sampling

10    (**Figure 6c**). No changes in isotopic composition were observed below a depth of 0.6m.

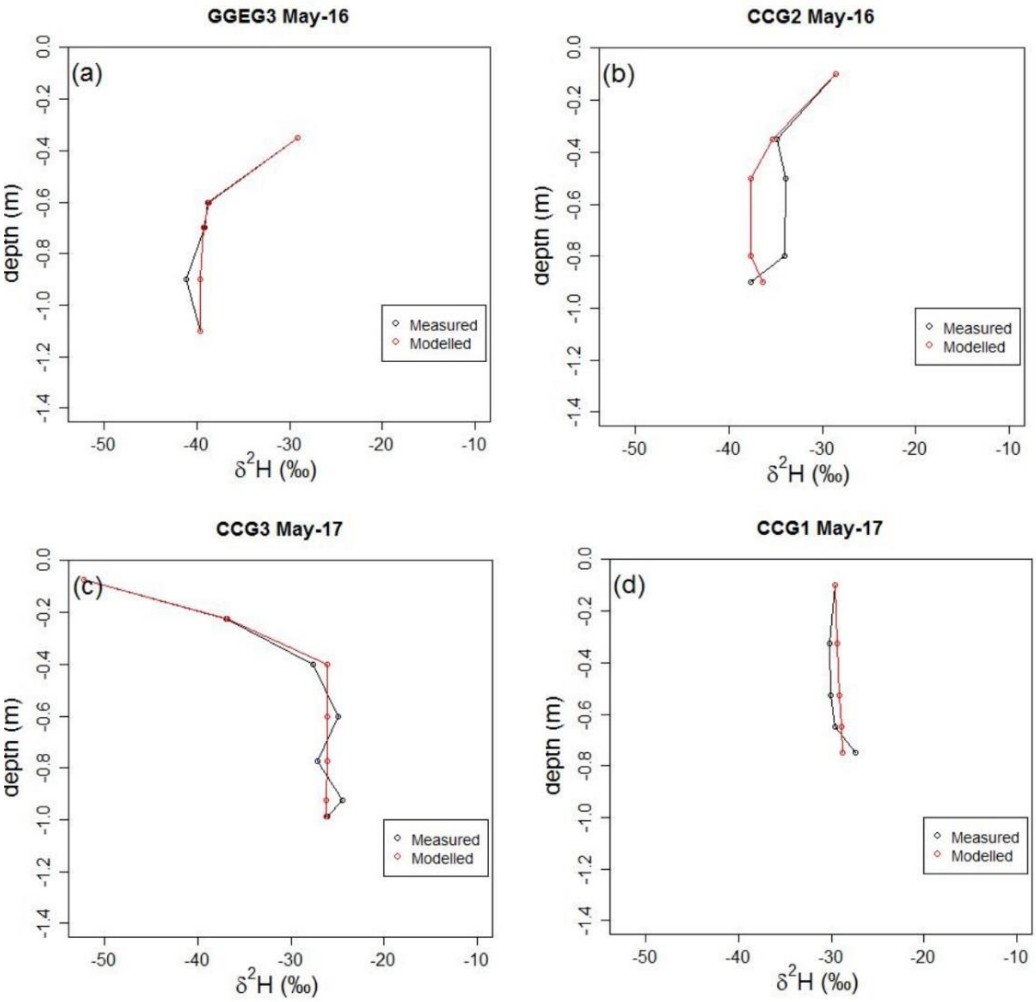

**Figure 8: Modelled vs measured $\delta^2H$ in unsaturated soil profile after dry period (a, b) and wet period (c,d)**





## Discussion

### 5.1 Swamp stratigraphy, geomorphology and groundwater condition

Swamp sediments are thin (up to 1.5 m) and are deposited directly on the sandstone basement. Typically, the organic soil or peat is 40-60 cm thick, underlain by unconsolidated alluvial sand and sandy silt with organic rich thin bands. The geomorphology of the Newnes swamps is consistent with the intact swamp classification as reported by Fryirs et al. (2016), with moisture and organic matter content as reported in Blue Mountain swamps by Cowley et al. (2016). The lithology indicates that the sediment transport is alluvial, however limited and occurring over relatively short distances (length of the swamp).

An important finding of this research is that no evidence was observed for a clay rich layer with sealing properties at the base of these swamps. This indicates that the swamp sediments are in direct hydraulic connection with the underlying sandstone, although there is likely to be a decrease in permeability at this interface (**Figure 9**).

Based on stable gravimetric water content below 0.4 m in CC and 0.6 m depth in GG and GGSW swamps, these elongated gentle gradient (50 mm/m average (Cardno, 2014)) shrub swamps are likely supported by groundwater recharge. Further indirect evidence for groundwater (in sandstone) interaction with swamp sediments and long-term saturation is the consistency of vegetation habitat (*Boronia deanei,* - wet heath shrub species) (Benson and Baird, 2012), measurement of groundwater levels (Centennial Coal, 2016), and age of swamp sediments (dated at 11,000 years, Chalson and Martin, 2009). Chalson and Martin (2009) undertook radiocarbon dating on pollen from a swamp on the Newnes plateau and found that the calibrated ages were 11,000 to 7,500 years (sampling depth 55-90 cm) and decreasing to 1,800 years at 40 cm depth. These ages support the existence of the swamps through the Holocene and their long-term interaction with groundwater.

We measured groundwater levels at the base of the swamp and within the underlying sandstone to a depth of around 10 m bgl at the downstream end of GG swamp indicating that these two units could be hydraulically connected. Typically, the groundwater levels in TPHSS (CC swamp) rise and decline in response to rainfall recharge (Centennial Coal, 2016) with very little lag time. Rapid infiltration and discharge is supported by low groundwater salinity (measured in this study) resulting from limited leaching of salts from the swamp. However, not all rainfall leads to recharge at the groundwater table. Given high moisture and organic matter content and evidence of seasonal precipitation in $\delta^{18}O$ and $\delta^{2}H$ profiles (p<0.05) in the upper swamp horizons, we conclude that in this zone the high water holding capacity increases residence time following the initial infiltration (vertical groundwater flow). The $\delta^{18}O$ and $\delta^{2}H$ of pore water in this variably saturated zone exhibit summer evaporation trends and a winter rainfall signature. The lateral groundwater discharge to the swamp is characterised by longer residence time compared to water exchange through the swamp and based on depleted $\delta^{18}O$ values and minor change between the sampling events. The similarity in EC and pH between surface and groundwater (not reported here) further supports relatively rapid infiltration and possibility of both lateral and upward local groundwater discharge that provides baseflow to the swamp.





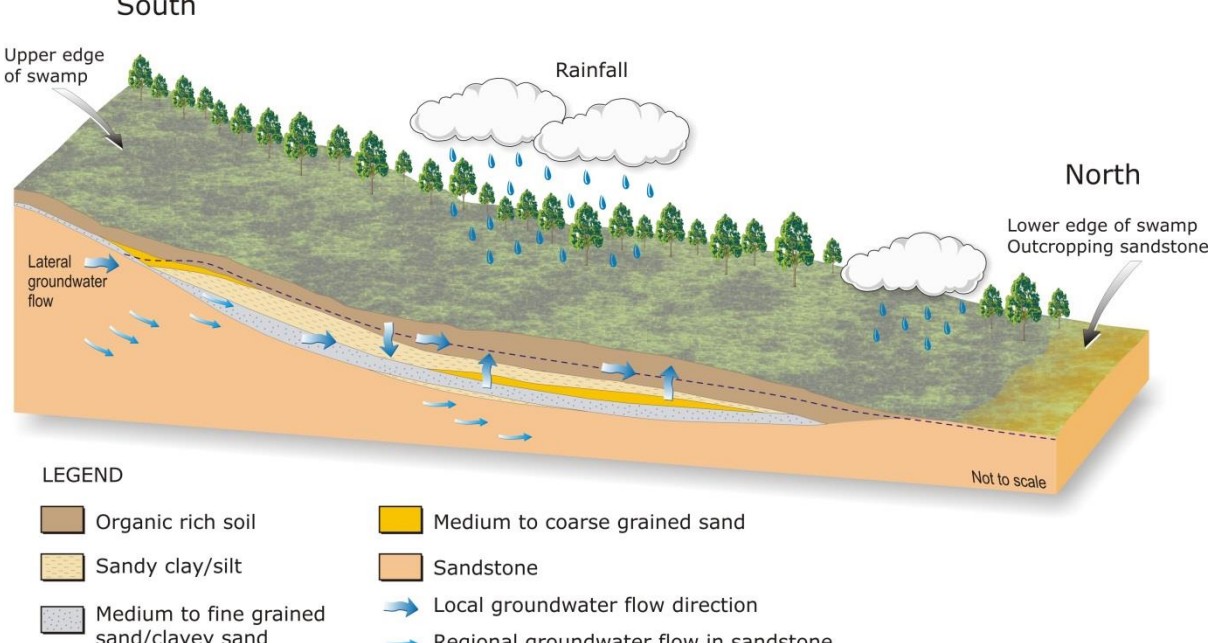

**Figure 9: Conceptual representation of water dynamics in the swamp system.**

To validate this conceptual model, a simple water/mass balance was completed based on the evaporative losses estimated by analytical model (Barnes and Allison, 1988). Using the results from dry weather period February to May 2016, we obtain

evaporation estimates ranging from 1 to 9 mm/day. The evaporation occurs in the top 0.4 m of the vertical profile, with an absence of fractionation below this depth where porewater isotopes values are similar to groundwater. These evaporation rates suggest high evaporation compared to rainfall in the same time period (**Table 2**).

The rainfall data from Lithgow BoM station 63132 and reference evapotranspiration (ET) data from Nullo Mountain BoM station 62100 (94 km north of the study site in the same mountain range and similar elevation and climate) indicate that in this

dry period (February to May 2016) the ET significantly exceeded the rainfall (Table 2). With a reference ET ranging from 1.7 to 4.4mm/day and E ranging mainly from 4 to 9 mm/d, the ET/E ratio for these swamps would be 0.7 to 0.3. This ratio is at the lower end of measured wetland ET/E ratio for typical wetlands indicating that reference ET could underestimate that based on realistic evaporation rates obtained by matching the modelled to observed data.

The $ET_c$ for typical wetland vegetation (sedge) in temperate climate is in the range from 0.8 to 1.2 (Allen et al, 1998; Mohamed

et al, 2012) and 0.7 was reported in a swamp in Murrumbidgee, Australia (Linacre et al, 1969). The $ET_c$ in our case is less than the estimated E based on stable isotope data. As transpiration does not fractionate, the actual evapotranspiration in the dry and warm period would have to be greater than the estimated evaporation. This would result in more water balance losses, requiring more water be supplied from other sources.

Runoff represents only small component of the water budget for several reasons. Firstly, the 10% slope gradient of the ridges,

3% slope gradient along the swamp floor and densely vegetated sides and base of the swamp minimize the runoff significantly.



Secondly, the upper soil layer is peat with significant water holding capacity compared to other soil types, and as indicated by the gravimetric water content measured in CC and GG swamps.

A simple mass balance comprising the rainfall (input), runoff (input) from the catchment considered two different approaches in dry period using E or $ET_c$. When $ET_c$ (output), was used, March had excess water with a deficit in February, April and May of between 10 and 60 mm.

However, the same mass balance calculated with E using 4 mm/day, had water deficit of between 10 and 113 mm/month for any month. If E of 9 mm/dayis used in the water balance, water deficit occurs in every month in the range from 10 to 260 mm/month (Table 2). Either way, two important output components are not considered in this mass balance: transpiration and discharge at the rockbar downgradient of the swamp. The estimation of these two components is uncertain, but inclusion in the water balance would increase the water deficit further. Importantly, if swamp discharge data were available in combination with pore water isotope profiles, an appropriate crop transpiration could be determined for these swamps, a factor that is typically a large unknown in water balance studies.

It is evident that given the water deficit, even without two output components, an additional water source must have maintained the groundwater levels in the swamp. We therefore conclude that groundwater is a significant contributor to swamp water balance, particularly during dry periods. For example, in GG swamp the groundwater levels are in the range from 0.28 to 0.38m bgl but the E rates are high enough to evaporate groundwater from surface to at least 0.6m.

Furthermore, measured loss of moisture as shown in Figure 5 indicates that significant loss occurs in such dry weather period in the top 40 cm (up to 150% by weight), while lower parts of the swamp remain saturated. The estimate of groundwater contribution in the drier period (February to May 2016) ranges from 10 to over 113 mm/month if measured 4 mm/day of evaporation is considered, and up to -260 mm/day if E of 9 mm/day is considered. The water balance was undertaken for dry period only as evaporation from soil profile using stable isotopes was considered to be most accurate during that period. Thus, even in the months where water balance is positive, ia groundwater contribution is likely, as evident from discharge at the rockbar observed at the end of dry period.

Although there is a compelling explanation for significant groundwater contribution, the actual volume cannot be estimated without knowledge of groundwater recession rate and/or measurement of discharge from the swamp. Figure 10 shows the water deficit and estimated groundwater contribution to each of the swamps for the $ET_C$ and E methods. The relative groundwater contribution is dominant in the dry weather period when it exceeds total rainfall. This groundwater contribution range represents the minimum and conservative value given that discharge from the swamp is not included in the water balance.

**Table 2: Water/mass balance components: measured rainfall and ET data (Lithgow and Nullo Mountain), runoff, measured E and estimated groundwater contribution (negative values are groundwater contribution).**





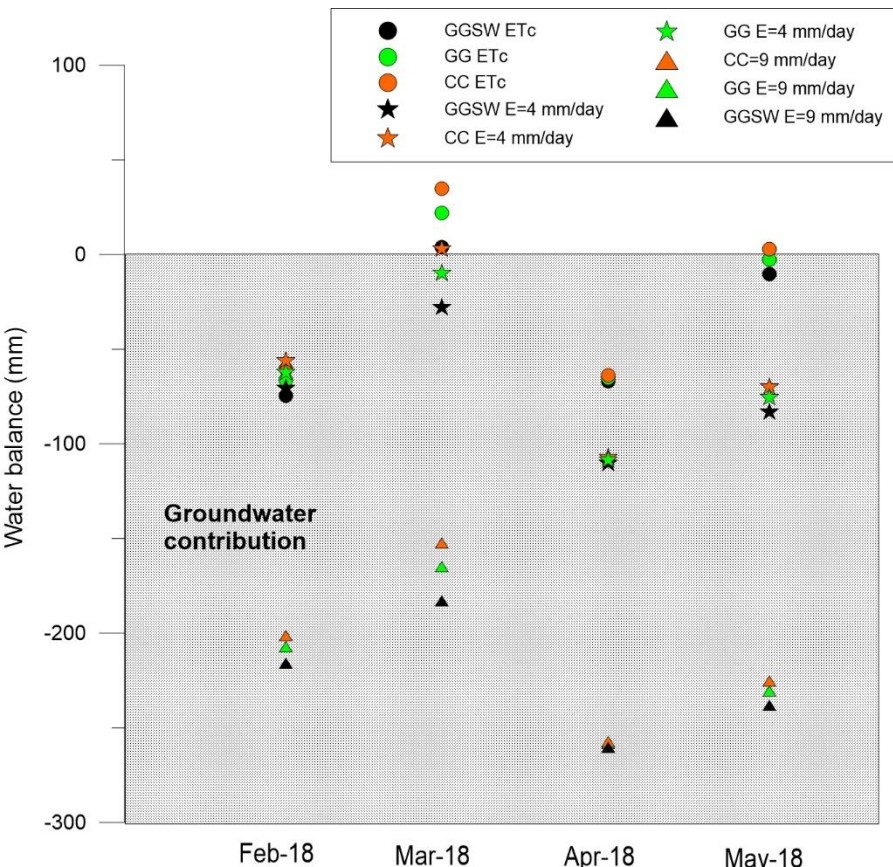

**Figure 10: Water balance during dry period estimated using ETc and E for each of the swamps.**

## 5.2 Groundwater movement within the swamp system

The vertical depth profiles of pore water $\delta^{18}O/\delta^2H$ can provide time series information by tracing the influence of the rainfall

isotopic signature in recharging water. The pore water direct vapour equilibration method is used for the first time in a swamp environment and results compared with clearly defined end members which included surface water, rainfall and groundwater. Constraining the interpretation of isotope results with these end members enabled groundwater inputs to be identified.

The evaporation response in the upper 40 cm is consistent with depth of penetration dependent on evaporation rate, soil type and time between rainfall events (Mathieu and Bariac, 1996; Malayah et al, 1996; dePaolo et al., 2004). As evaporation

proceeds, capillary rise of groundwater reduces the $\delta^{18}O$ enrichment closer to the surface. Moisture content data reveals variability at 30–70 cm depth, which is also observed in $\delta^{18}O$ and $\delta^2H$ profiles and is related to interlayering of fine and coarser grained material, consistent with other studies (dePaolo et al., 2012). The intercept of the pore water regression line with the LMWL is depleted relative to weighted average rainfall. Therefore, the isotope signature in the partially saturated zone (variable from 0.05 m bgl to 0.4 m bgl in the swamp) in the summer period (May 2016 sampling event) is a result of

evaporation.



There is a difference (p<0.05, n=18) between samples collected after dry and warm versus wet and cool antecedent conditions. The October 2016 (cool weather) samples from CC swamp are typically depleted in $\delta^{18}O$ and $\delta^2H$ and we conclude that these values are within the range of winter rainfall isotope values. Below 100 cm depth the values of $\delta^{18}O$ remain uniform and consistent with groundwater values but also with surface water. We infer this to represent groundwater derived from vertical infiltration and laterally from sandstone. We therefore consider the main processes to be rapid infiltration through the swamp sediments to water table but at the same time high water retention in the upper horizons, and slow lateral exchange of pore water below the vadose zone.

The vertical topographic difference from the swamp headwaters to the downstream end of the swamp (typically a sandstone rockbar) is around 40 m. This elevation difference is too small to result in any difference in isotopic signature, therefore, given the spatial response and assuming a homogeneous environment with vertical flow, pore water $\delta^{18}O$ and $\delta^2H$ should be similar (Garvelmann et al, 2012). However, observed variation in profiles is not uniform, and is caused by vertical rainfall infiltration in the upper part of the profile and lateral flow at the base. The lateral flow within the swamp sediments is further enhanced by regional groundwater flow contribution from the valley sides. Such lateral flow is reported in these swamps where sandstone is underlain by a claystone layer (Corbett et al, 2014).

Factors such as fine-grained content of lithological units, reported by other studies (dePaolo et al., 2012), have been found to result in a bigger shift to depleted $\delta^{18}O$ and $\delta^2H$ and variation in isotope signature with depth. The reason for this is related to hydraulic conductivity of the unconsolidated soil. For example, the biggest variation in $\delta^{18}O$ and $\delta^2H$ was observed in silt and clayey sand units (**Figure 7a** and **7b**) which contain higher percentage of particles <2 µm. Contrary to Garvelmann et al's (2012) observations, we did not find the variability in $\delta^{18}O$ and $\delta^2H$ to be a result of soil saturation and depth of vadose zone only, but also as a function of lithology and different grain size material (peat, organic soil with sand and silt). Variations in particle size, porosity and permeability would then influence groundwater flow and storage.

**Conclusion**

The hydrogeological and isotopic characterisation of these swamp environments provides a baseline understanding for future comparison of any hydrological changes due to natural or human activities. This study is the first application of the vapour equilibration method for determining stable isotopes of pore water in a wetland system. This unique pore water isotope approach, combined with other data and information has significantly improved a conceptual model of wetland hydrology. The pore water stable isotope method allows efficient sample collection without permanent disturbance, collection of vertically discretized data at any practicable frequency and without the need for sample squeezing.

Although groundwater is often assumed to be important for the sustainability of wetlands and swamp ecosystems, very little is known on how these systems would respond to changing groundwater conditions. The existing literature recognizes that natural variation in swamp ecology is not well understood given the complexity and interaction of groundwater and surface water.



This study found, for several upland peat swamps, groundwater is a dominant component of the water balance its contribution being larger than rainfall in the dry weather period. This finding is consistent with environmental tracer studies finding that 19-80% of water in Blue Mountains swamps is from groundwater, particularly in steeper and rounder catchments (Young, 2017). Furthermore, these swamp groundwater systems appeared to be in hydraulic connection with the underlying sandstone,

given similar groundwater levels, and the lack of a clayey layer at the base of the swamp. Although rainfall infiltration to water table occurs rapidly, the high water holding capacity of upper organic rich layers maintains the moisture for long periods. These processes are confirmed by the results of the water/mass balance, in particular during dry periods. The majority of flow through the swamp system is via lateral groundwater flow and its velocity depends on heterogeneity within this layer and its hydraulic conditions. Upward or downward flow between the swamp system and underlying rock is controlled by relatively

groundwater levels, the slope, and the hydraulic conductivity contrast at the interface.

The conceptual model presented here provides a valuable benchmark from which to assess potential changes in these swamps following underground mining and forestry activity. The improved understanding in the water balance in these swamps also has implications in other areas of the Blue Mountains where urbanization has a significant impact on upland swamps. The role that catchments have on the health of a swamp is important in supporting its flora and fauna, with groundwater likely to be a

primary factor that contributes to the long-term survival of the ecosystem. The protection of ecological community (Gorissen et al, 2017) is therefore dependent on maintenance of catchment stability and groundwater baseflow contribution if forestry activity and ground movement or deformation due to mining occur in the swamp catchment.

Measurement of pore water stable isotopes of peat and sediment within the swamp ecosystem provides direct information on the depth at which the evaporation occurs and understanding of the water cycle. E obtained from stable isotope direct

equilibration method was found to be more realistic than reference ET. In particular, based on current research of the water balance in wetland and swamp systems and ecology around the world, the application of this method could be beneficial to define water availability for flora and fauna in swamps where a thick organic soil/peat and sedimentary layer exists.

**Data availability**

The underlying research data can be accessed in the supplementary information.

**Acknowledgement**

This study was funded independently by UNSW. The authors would like to acknowledge the support of Centre for Water Initiative and School of Biological and Earth Sciences for assistance with sample analysis. Rainfall isotope analysis was funded

independent by ANSTO. We thank Prof A. Baker for providing constructive suggestions for this paper. We acknowledge Bob





Cullen for collecting rainfall samples at Mt Werong, Barbara Gallagher, Jennifer van Holst (ANSTO) and Fang Bian (UNSW) for analysis of rainfall samples, the Sydney Catchment Authority for providing rainfall data, and Karina Meredith (ANSTO) for advice on evaporation modelling.

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

**Table 1: Input variables for the unsaturated Barnes and Allison (1988) model.**

| Parameter | Value | Comment |
| --- | --- | --- |





| Temperature (°C) BoM Lithgow station 63132 | 1°C to 26°C | Monthly mean minimum and maximum, 10°C average |
|---|---|---|
| | | Peat |
| | 0.6 | Organic sand/clay |
| Bulk density (g/cm³) | 1.1–1.2 | Clean sand |
| | 1.7 | Basal sand |
| | 1.2 | |
| | 90 | Peat |
| | 70 | 20% peat and sand |
| Porosity (%) | 50 | 5% peat and sands |
| | 38 | Sands |
| | 0.39 | Organic rich sandy soil |
| | 0.66 | Medium-coarse sand |
| Tortuosity | 0.48 | Medium sand |
| | 0.21 | Fine sand |
| | 0.19 | Clayey sand |

**Table 2: Water/mass balance components: measured rainfall and ET data (Lithgow and Nullo Mountain), runoff, measured E and estimated groundwater contribution (negative values are groundwater contribution).**

| | Feb-16 | Mar-16 | Apr-16 | May-16 |
|---|---|---|---|---|
| Total monthly rainfall Lithgow station (SN63132) (mm/month) | 28.8 | 61.2 | 6.2 | 26 |
| Reference ET Nullo Mountain (SN62100) (mm/month) | 119.9 | 92 | 76.6 | 51 |
| Evaporation (pore water stable isotope profiles) mm/month | 117–267 | 123–273 | 120–270 | 123–273 |
| Runoff estimate (mm/month) | | | | |
| CC | 31 | 66 | 7 | 28 |
| GG | 25 | 53 | 5 | 23 |
| GGSW | 16 | 35 | 4 | 15 |
| Balance deficit mm/month (groundwater component) | | | | |





| | | | | |
|------|----------|----------|-------------|------------|
| CC | -63 to -207 | 4 to -146 | -110 to -257 | -69 to -219 |
| GG | -69 to -213 | -9 to -158 | -111 to -258 | -75 to -224 |
| GGSW | -78 to -221 | -27 to -177 | -113 to -260 | -82 to -232 |