# Peer review of "Application of the pore water stable isotope method and hydrogeological approaches to characterise a wetland system"

_Hydrology and Earth System Sciences, 2018_

## Referee Comment (RC1) · Anonymous Referee #1 · 25 May 2018

General Comments

This paper contains a novel method for characterising the hydrology of wetlands and swamps and presents a scientifically robust model of temperate upland swamp hydrology that fits within the context of current research into similar ecosystems. It is a well written paper with high scientific significance. One issue is that the terminology for describing groundwater within the swamps and regional groundwater aquifers is not differentiated. A major part of the paper is concerned with connectivity of the swamp aquifers with regional groundwater yet the term groundwater is used to describe both aquifers. One way of differentiation may be to call swamp groundwaters 'swamp water' or 'swamp water table' and regional groundwater 'groundwater' or 'sandstone aquifer' or similar and use these terms consistently throughout. Another issue is that while the

paper presents the application of the stable isotope direct vapour equilibration method to quantify water sources, it does not discuss this method in great detail. A paragraph (or two) to describe the data accuracy of the vapour method against the more conventional sampling method would be useful as would a more detailed discussion of the circumstances in which it could be used. Characterising flow paths within individual sedimentary units is one area where this method would be hugely advantageous. More detail is also required in describing methods and a description of the regional hydrogeology in the site description would be of use. See specific comments for detailed critique.

Specific Comments

Introduction: Page 3 Line 27:Change the term "hydrological balance".Swamp flora and fauna are dependent on the high water tables that are characteristic in THPSS. The term hydrological balance does not adequately describe this Page 3 Line 30: The term groundwater in this instance is confusing. Do you mean swamp groundwater or groundwater from the surrounding sandstone aquifer? Page 4 Line 1: Again groundwater terminology is confusing. Would suggest 'swamp water levels' or 'swamp groundwater' when referring to the swamp water table and 'regional groundwater' or 'sandstone aquifer' when referring to the bedrock aquifer.

Site Description: Need a description of regional hydrogeology to give a better picture of likely groundwater interactions. Page 6 Line 4: Does this mean the longwalls are located directly below the swamps?

Methods, Fieldwork and Sampling: More details of piezometers are required. Depth, installation method, construction materials etc Details of groundwater bore required including installation method, construction materials and depth Include a section on statistics and software used

Page 7 Line 5: Was a Russian D corer used to recover samples? If not how were samples recovered intact from a conventional auger? Page 7 Line 12: Swamp groundwater

or regional groundwater? How were sandstone aquifer samples collected? Was the existing piezometer drilled within the bedrock?

Results: Page 10 Line 14-15: wouldn't this just be collected rainwater? Page 10 Line 20: This sentence would be better placed within the methods Page 11 Line 7: It also may be the result of lateral throughflow along the longitudinal gradient, particularly within the sandy units Page 15 Line 8:Or that the surface water sample points are located in the discharge zone for groundwater flow Page 15 Lines 10-14: Figure caption is confusing. Change groundwater terminology. Page 16 Line 28: Probably should be explained in the methods Page 17 Line 5: In that case it would be informative to relate enrichment to relative humidity to assess whether that has more influence on evaporation than temperature

Discussion: Page 18 Line 10: This sentence should be combined with line 11 below to strengthen this argument. As it is, the sentence hangs without supportive evidence Line 14: I'm not sure this statement holds up. Long water residence times within the swamp water table may be occurring to sustain this vegetation community Line 14-measurement of groundwater. Measurement of groundwater levels is not evidence of aquifer connectivity. Consistency of swamp water tables and lack of significant drawdowns in dry periods may however be linked to aquifer connectivity. See Cowley et al 2018 "The hydrological function of upland swamps in eastern Australia: The role of geomorphic condition in regulating water storage and discharge" Line 18:Again this statement does not represent evidence of groundwater interactions per se. It is speculation. Reword Line 20: Measurement of GW levels above & below sandstone is not an indication of connection. GW level comparison of both aquifers may be, as might be comparison of isotopic signatures. Reword. Line 22: Rapid infiltration and discharge of what? Swamp or sandstone aquifer? Line 22: Where are measurements of groundwater salinity? Line 23: "resulting from limited leaching of salts from the swamp".Not sure you have the evidence for this statement Line 23: "recharge of the groundwater table". Swamp or sandstone aquifer? Line 29: reference required for EC & pH results Page

22: Line 4: groundwater from sandstone aquifer? swamp groundwater? Line 9: 'Isotopic signature' of precipitation? Line 14: A cross section of underlying hydrogeology would add to this conceptual model of swamp hydrology

Technical Corrections Abstract: Line 6: Add 'Endangered' before the word ecological and 'Under state and federal legislation' after communities Page 5 Figure 1: An Aerial photo or satellite base map would be better to define swamp boundaries than a topographic map. Page 7 line 13: Space needed between 'were' and 'described' Page 10 Figure 4: Where are the profiles and sediment logs for GGSWG swamp? Page 11 Figure 5: These charts may be better shown by putting the sampling periods together on one graph rather than separating the swamps. That would make it easier to flip between then and the rainfall charts. Putting sediment logs down the left hand side may make comparisons between sediment, moisture content and organic matter easier Page 15 Figure 7: Why are the surface water sample points low down in the depth profile in c and d but at the surface in a and b? Put them all at the surface Page 18 Line 21: THPSS Page 19 Figure 9: Change the colour of the Medium to fine grained sand/clayey sand unit. It appears at first glance to be indicating a water table Page 19 Line 14: Explain ETc Page 20 Line 7: space between day and is Page 21 Figure 10: This graphic does not effectively display the data in table 2. A simple column graph may be more effective. I don't understand why you used 2018 dates. Would it not be better to use sampling period dates? You need to explain why these dates were chosen Page 23 Line 9: Missing word after 'relatively' Line 15: Gorissen reference should go after the word 'ecosystem'. Insert 'this' before 'ecological' Page 32-33 Table 3: should be Table 1. It's difficult to determine which numbers pertain to which parameter. Either move the parameters or put borders around columns and rows. Move column 3 down do that the first 'peat' is in lone with the first bulk density number Page 33: Table 4 should be table 2

References: References cited in text that are not in reference list: Huneau et al, 2003, Hendry et al 2013, Hendry, 2008, Hunt et al, 1996, Mandl et al, 2017, Bickford and

Gell, 2005, Middleton and Kleinebecker, 2012, Johnson, 2006, Valentin et al, 2005, Gatt, 1996, Dansgaard, 1964, Linacre et al, 1969, Mathieu and Bariac, 1996, dePaolo et al 2012.

References in reference list not cited in text:

Andersen, M., Barron, O., Bond, N., Burrows, R., Eberhard, S., Emelyanova, I., Fensham, R., Froend, R., Kennard, M., Marsh, N., Pettit, N., Rossini, R., Rutlidge, R., Valdez, D. and Ward, D.: Research to inform the assessment of ecohydrological 10 responses to coal seam gas extraction and coal mining. Department of the Environment and Energy, Commonwealth of Australia, 2016.

Bukata, B.J., Osborne, T.Z. and Szafraniec, M.I.: Soil nutrient assessment and characterisation in a degraded Cnetral Florida 30 Swamp. Water Air Soil Pollut, 226: 307, 2015.

Centennial Coal.: Springvale Mine Extension project, Environmental Impact Statement , Available online http://majorprojects.planning.nsw.gov.au/index.pl?action=view_job&job_id=5594 Accessed on 20 April 2017, 2014.

Cloern, J. E., Canuel, E. A. and Harris, D.: Stable carbon and nitrogen isotope composition of aquatic and terrestrial plants of the San Francisco Bay estuarine system. Limnology and Oceanography, 47(3): 713-729, 2002.

Denk, T. R. A., Mohn, J., Decock, C., Lewicka-Szczebak, D., Harris, E., Butterbach-Bahl, K., Kiese, R. and Wolf, B. D.: The nitrogen cycle: A review of isotope effects and isotope modeling approaches. Soil Biology and Biochemistry 105: 121-137, 2017.

Deegan, L. A. and Garritt, R.H.: Evidence for spatial variability in estuarine food webs. Marine Ecology Progress Series 147: 31-47, 1997.

Fry, B.: Stable isotope ecology, Springer, LA, pp 317, 2006.

Gardner, W.H.: Water content. In: (ed. A. Klute) Methods of Soil Analysis, Part 1. Bibliography 165 Physical and Mineralogical Methods. Agronomy Monograph No.9 (2nd edn). pp493- 544, 1986.

Gorissen, S., Mallinson, J.Greenlees, M. and Shine,R.: The impact of fire regimes on populations of an endangered lizard in montane south-esastern Australia. Austral. Ecol. 40, 170-177, 2015.

Heaton, T.H.E.: Isotopic studies of nitrogen pollution in the hydrosphere and atmosphere: a review. Chem. Geol. 59, 87-102, 20 1986.

Huneau, F, Blavoux, B, Aeschbach–Hertig, W. and Kipfer, R.: Paleogroundwater of the Valreas Miocene aquifer (Southeastern France) as archives of the lgm/Holocene transition in the western Mediterranean region, IAEA report, IAEA-CN-80/24, 2013. Johnson, D.: Sacred waters: the story of the Blue Mountains gully traditional owners. Broadway, N.S.W.: Halstead Press, 237 pp, 2007.

Liu, Y., Sheng, L. and Liu, J.P.: Impact of wetland change on local climate in semi-arid zone of Northeast China. Chinese Geographical Science. 25,309-320, 2015.

Mandl, M. B., Shuman, B. N., Marsicek, J., Grigg, L.: Estimating the regional climate signal in a late Pleistocene and early Holocene lake-sediment $\delta$18O record from Vermont, USA. Quaternary Research (United States) 86(1): 67-78, 2016.

Potter, N.J., Chiew, F.H.S., Frost, A.J., Srikanthan, R., McMahon, T.A., Peel, M.C. and Austin, J.M.: Characterisation of 20 recent rainfall and runoff in the Murray-Darling Basin. A report to the Australian Government from the CSIRO Murray-Darling Basin Sustainable Yields Project. CSIRO, Australia. 40pp, 2008.

Rau G.H.: Carbon-13/carbon-12 variation in subalpine lake aquatic insects: Food source implications. Can. J. Fish. Aquat. Sci 37: 742-746, 1980.

Reddy, K.R. and DeLaune, R.L.: Biogeochemistry of wetlands science and applications. Boca Raton, FL: CRC Press, pp774, 30 2008.

Zhang, X., Sigman, D.M., Morel,F.M.M>, Kraepiel, A.M.L.: Nitrogen isotope fractionation by alternative nitrogases and past ocean anoxia. Proceedings of the National Academy of Sciences of the United States of America 111, 4782-4787, 2014.

---

## Referee Comment (RC2) · Anonymous Referee #2 · 9 Jun 2018

General comments:

The manuscript "Application of pore water stable isotope method to characterize a wetland system" by Katarina David et al. aims at characterizing an intact, vegetated wetland system located in the Australian highland and potentially endangered by anthropogenic activity, namely mining. For this purpose, the authors used hydrogeological and isotopic data and combined the latter with a model originally developed for arid zones. While the employed model yielded high evaporation rates based on water stable isotopes, no quantitative statements were made on transpiration or groundwater flow. Based on their findings, the authors developed a conceptual model that emphasizes the importance of lateral groundwater inflow for the persistence of such wetland systems.

While I personally support the motivation of this study and also the propagation of isotope methods I regret to say that I have several issues regarding this manuscript:

Title: The title does not fully reflect the work described here. Not only isotope data have been collected and used and the manuscript's main conclusion is based on other data.

Main outcome: The most important, yet qualitative finding of this study is groundwater flow through the wetland system. The presence of high groundwater levels despite high evaporation rate estimates led to the indirect conclusion that lateral groundwater inflow must be effective. This could already be expected, given just the combination of slope (5%), groundwater levels (high) and permeability of bedrock (sandstone –> high). The vulnerability of such systems following e.g. mining activity is based on potential changes of groundwater flow. The main effort of this study, however, was the estimation of evaporation rates which will most likely not be altered by e.g. mining activity. Furthermore, the application of the direct vapor equilibration method was never methodically restricted to non-wetlands and therefore does not constitute a challenge itself that needs to be emphasized.

Model selection: The selected model was developed and applied in desert regions with only vertical water flow, no vegetation, and no lateral groundwater contribution to the zone under investigation. I therefore doubt that invariance of isotopologue diffusivities is an exhaustive criterion for model selection. Furthermore, the assumption of steadystate conditions in a region with pronounced wet and dry season seems to be very far-fetched. The authors should either provide more details why the selected model is still applicable in a vegetated wetland environment with vertical and lateral water flow components or they should use a different approach that better considers subsurface water flow velocities and directions under both dry and wet conditions.

Data collection: The description of the calibration and validation routine of water stable isotope data is quite confusing. Why did the authors not use the linear regression be-
tween known values of liquid standards and raw readings of the respective headspace vapors to calibrate the unknown sample values? What did they correct the readings for? Or did they mean "calibrate" when they wrote "correct"? Why did they (have to) calculate individual fractionation factors? This would not have been necessary if all standards and samples had been stored sufficiently long and analyzed subsequently in a temperature-controlled environment following the principle of identical treatment. Also, reported uncertainties as measures of data quality are meaningless if they are not based on the authors' applied standard operation procedures.

Specific comments and Technical corrections:

P2 L7-8: "aiming at" instead of "enabling". The ability to quantify components of the water balance depends on environmental conditions, not on the method of data collection.

P2 L8-10: insert "potentially" before "enables". Otherwise this statement is too strong. The technique itself only enables collection of isotope data. Understanding of processes is a different thing.

P2 L11 & elsewhere: the numbers following the delta symbol have to be in superscript

P2 L12: is the porewater compression technique the most widely applied and accepted benchmark?

P2 L19-21: the finding of sand underlying the swamp can't really be credited to this study as it had been described before in studies cited by the authors.

P3 L9: multi, not muliti; sedimentary, not sedimentatry

P3 L11 and throughout MS: the references cited in the MS do not match the ones listed in the references section. I was therefore often unable to find the referenced statements in the cited literature

P3 L12, 16 & 19: why do the authors distinguish between Australian and international

**HESSD**
literature? Is one more relevant or trustworthy than the other?

P3 L17: insert "other" before "environmental tracers"

P3 L18: please specify the processes you are referring to

P3 L26: all THPSS terms should start with capital letters. Or none.

P3 L31 & elsewhere: citation style: put years in braces

P4 L3: "swamp behavior" is too sloppy, please specify

P4 L18: given that especially rainfall and surface water can vary on very short timescales with unknown time lags relative to soil water which in turn is subject to dispersion, how can these be considered distinct endmember points?

P5 L5: a verb is missing after "and"

P6 L2-4: if this does not affect the site under investigation, then why mention?

P6 L7: "fed by groundwater discharge" is a contradiction. I suggest to use the expression "fed by lateral groundwater inflow"

P6 L16-19: please provide numbers rather than "below average" or "above average"

P7 L5: can you please provide here already the number of samples as well as the achieved spatial depth resolution of sampling?

P7 L7: why had sampling holes to be restored or why is this important to be mentioned?

- P7 L7: excess of what? Why was not all soil material sampled?
- P7 L13: please specify what you mean by "nature"
- P7 L13: "were described" instead of weredescribed"

P7 L16: why were samples vacuum packed? How and how long were they stored prior to analysis?
P8 L1: insert "data" before "for precipitation"

P8 L9: 17-24 hours is too short for reaching complete isotopic equilibrium between soil water and headspace vapor. It should be several days, up to one week for clayey samples (Wassenaar et al, 2008). Or do the authors have indication that complete isotopic equilibrium between all relevant fractions of soil water was reached? If so please describe

P8 L10: I suppose this concentration range yields minimum data noise on a LGR instrument? Such a large range of vapor concentrations, however, probably indicates non-isothermal storage prior to and during analysis. Can the authors please comment on pre-analysis storage conditions with respect to the principle of identical treatment of samples and standards?

P8 L13: what kind of plastic tube? Was it diffusion-tight material? Else, the authors might have sampled a mixture of sample headspace and an unknown fraction of ambient vapor.

P8 L15: 20 minutes appear way too short. See also comment to P8 L9

P8 L15 & elsewhere: the expression " $\delta$ 18O/ $\delta$ 2H" is conceptually wrong and misleading given that this could be interpreted mathematically as a ratio of two isotope ratios

P8 L18: what was the volume of the standards? Did you prepare replicates or were the standards' headspaces analyzed repeatedly?

P8 L20-21: does this refer to liquid analyses or the direct vapor equilibration method?

P8 L21-22: how do these timespans compare to sampling time of the individual sample headspaces? If this is the precision reported by the manufacturer I would prefer to read about the precision the authors achieved when following their standard operation procedure (also in P9 L2)

P8 L26: LGR's technology is called e.g. OA-ICOS, but not IRMS
P8 L30: please provide a reference for the LOI method

P9 L5: this contradicts MS title which prominently mentions isotope data

P10 L4: what is "sub-angular" quartz?

P10 L5: please describe how grain size distributions were determined. If this information was taken from other publications it should appear in section 2 rather than the result section.

P10 L9: "dark" is not a color

P10 L10: can the authors please comment on the "organic smell". Volatile organic compounds likely have a strong effect on laser-based isotope analysis that needs to be considered for such samples.

P11 L5: "can be explained" seems to be not only descriptive and should appear in the discussion rather than the result section

P12 L1: please insert some specification after "of"

P12 L3 & 8: the intercepts in the linear equations should have a the "unit" ‰

P12 L9: it appears to me that the slope is slightly higher but the intercept is similar

P13 L9: "likely to be": see comment on P11 L5

P14 L3 & throughout MS + supplement: stable isotope data should be consistently reported with two decimal places for  $\delta$ 18O data and one for  $\delta$ 2H data

P14 L5 "it seems unlikely": see comment on P11 L5

P14 L10: this seems to be a misinterpretation potentially affecting overall results. This regression line is not to be mixed up with evaporation lines (See Benettin et al, 2018, for details). Further, it should not be interpreted in the results section

P14 L14: Since the authors present a LMWL, I would suggest to report Ic-excess
values (Landwehr & Coplen, 2006) rather than D-excess values

P14 L15-20: this paragraph appears twice

P14 L23: "we would expect": see comment on P11 L5

P14 L25: "seems to suggest": see comment on P11 L5

P14 L29: "may be": see comment on P11 L5

P15 L8: "assumed to be": see comment on P11 L5

P16 L1 & 7: insert "values" or equivalent before "of pore"

P16 L8-9: "due to": see comment on P11 L5

P16 L11: is the number in braces a  $\Delta\delta$ -value?

P16 L11: "is a reflection": see comment on P11 L5

P16 L13: isn't this just a water balance rather than a water and mass balance?

P16 L19: "very" = "vary"?

P16 L26-27: the description how data were collected should be provided in the method section

P17 L8 & 9: "is related to": see comment on P11 L5

P18 L12: "groundwater recharge" is misleading as it describes the replenishment of groundwater. I suggest to use "lateral groundwater inflow" instead.

P18 L11-18: the information on e.g. slope, vegetation, age dating should be provided in the section describing the study site (section 2)

P18 L22: why not report salinity data? Do they support the surprisingly high evaporation estimates?

P18 L26: insert "values" or equivalent before "of pore"
P18 L27: discharge: see comment on P6 L7

P18 L30: I would expect that upward flow supporting high evaporation rates would result in an increase of EC in surface water.

P20 L15-16: evaporation always occurs at the liquid-vapor interface (the "surface"). How can evaporation ("to at least 60cm") occur below the water table (28-38cm)?

P20 L19: "measured"? This number was rather calculated than measured

P20 L25: this figured should be presented and described in the results section

- P21 L5: "percolating" instead of "recharging"
- P21 L6: "clearly defined endmembers": see comment on P4 L18

P22 L1: difference in what?

P22 L9: "of precipitation" after "signature"

P22 L27-28: this conclusion has been drawn before (e.g. Wassenaar et al, 2008) and can't be credited to the present study

P22 L29-32: this statement is not conclusive and rather belongs to the introduction

Supplement: how come that foil weights differ by more than factor 4? Did the authors not use standardized sample bags (e.g. Ziploc) with comparable weights? If not this would be a violation of the principle of identical treatment.

References:

Benettin, P., T.H.M. Volkmann, J. Von Freyberg, J. Frentress, D. Penna, T. E. Dawson, and J.W. Kirchner (2018): Effects of climatic seasonality on the isotopic composition of evaporating soil waters. Hydrol. Earth Syst. Sci., 22, 2881–2890

Landwehr, J.M. and T.B. Coplen (2006): Line-conditioned excess: a new method for characterizing stable hydrogen and oxygen isotope ratios in hydrologic systems. In:
Isotopes in Environmental Studies, Aquatic Forum 2004: International Atomic Energy Agency, Vienna, Austria, IAEA-CSP-26, p. 132-135

Wassenaar, L. I., Hendry, M. J., Chostner, V. L., and Lis, G. P. (2008): High resolution pore water  $\delta$ 2H and  $\delta$ 18O measurements by H2O(liquid)–H2O(vapor) equilibration laser spectroscopy. Environ. Sci. Technol., 42, 9262–9267

---

## Author Comment (AC1) · 18 Jun 2018

Dear Reviewer, Thanks for the very constructive review and comments provided for the manuscript. We have included a detailed response to the questions below in blue. Reviewer's Comment This paper contains a novel method for characterising the hydrology of wetlands and swamps and presents a scientifically robust model of temperate upland swamp hydrology that fits within the context of current research into similar ecosystems. It is a well written paper with high scientific significance.

Response:We thank the reviewer for the positive feedback.

Comment:One issue is that the terminology for describing groundwater within the swamps and regional groundwater aquifers is not differentiated. A major part of the

paper is concerned with connectivity of the swamp aquifers with regional groundwater, yet the term groundwater is used to describe both aquifers. One way of differentiation may be to call swamp groundwaters 'swamp water' or 'swamp water table' and regional groundwater 'groundwater' or 'sandstone aquifer' or similar and use these terms consistently throughout.

Response:Thanks, the terminology has now been improved by differentiating between swamp groundwater and regional groundwater by introducing the new terminology as suggested. The swamp groundwater is now used to discuss swamp groundwater and regional groundwater to describe regional groundwater.

Comment:Another issue is that while the paper presents the application of the stable isotope direct vapour equilibration method to quantify water sources, it does not discuss this method in great detail. A paragraph (or two) to describe the data accuracy of the vapour method against the more conventional sampling method would be useful as would a more detailed discussion of the circumstances in which it could be used.

Response:The method has been described in detailed in referenced Wassenaar et al, 2008; Wassenaar and Hendry, 2008 and Hendry et al, 2015. Several additional sentences were added to both Sections 3.1 and 3.2 of the manuscript to add to the method description. Data accuracy of the vapour method is provided in Line 25, Pg7 and a comparison to the conventional sampling method and advantages of the method are provided as an additional paragraph in Line 30 Pg 7.

Comment:Characterising flow paths within individual sedimentary units is one area where this method would be hugely advantageous.

Response:We agree that characterising the flow paths in the individual sedimentary units in these particular swamps would be advantageous, however very limited data is available in the public domain. However, additional local geology and hydrogeology data is presented in the Section: Site Description to provide the background to understanding the context. It is expected that more groundwater data will be available in the

future to characterise the flow paths in the individual units for the researched swamps.

Comment:More detail is also required in describing methods and a description of the regional hydrogeology in the site description would be of use.

Response:More details are now also provided in Section 3.2 on describing different methods used in this research. Description of the regional hydrogeology is now provided in Section 2: Site description, based on the limited publicly available data for these swamps.

Specific review's comments

Introduction: Comment:Page 3 Line 27: Change the term "hydrological balance". Swamp flora and fauna are dependent on the high water tables that are characteristic in THPSS. The term hydrological balance does not adequately describe this.

Response:As suggested by the reviewer, the term ''hydrological balance" has been replaced with "high water table".

Comment:Page 3 Line 30: The term groundwater in this instance is confusing. Do you mean swamp groundwater or groundwater from the surrounding sandstone aquifer?

Response:Groundwater was taken from both swamps and sandstone aquifer, but to avoid confusion the term groundwater in this sentence relates to sandstone regional groundwater which is now termed regional groundwater. Another term has been introduced for groundwater in swamp and this is swamp groundwater. As a result, the Line30/Pg3 has added term swamp groundwater to include both.

Comment:Page 4 Line 1: Again groundwater terminology is confusing. Would suggest 'swamp water levels' or 'swamp groundwater' when referring to the swamp water table and 'regional groundwater' or 'sandstone aquifer' when referring to the bedrock aquifer.

Response:The term swamp groundwater has been adopted based on the reviewer's suggestion for swamp water while regional groundwater term describes the bedrock

aquifer. This has been corrected at Page4 line 1 and throughout the document.

Site Description: Comment:Need a description of regional hydrogeology to give a better picture of likely groundwater interactions.

Response:Description of the regional geology and hydrogeology is provided on Pg5/Line 12 to 24.

Comment:Page 6 Line 4: Does this mean the longwalls are located directly below the swamps?

Response:The longwalls are not directly underneath the swamps, but below ground to the southwest of the swamps, with the closest swamp to the longwall being GGSW. This has been reworded in the text to clarify.

Methods, Fieldwork and Sampling: Comment:More details of piezometers are required. Depth, installation method, construction materials etc Details of groundwater bore required including installation method, construction materials and depth

Response:Piezometers were installed by the mining company prior to our research study. To minimise disturbance to the swamp, all piezometers were installed by manual augering the 80-mm diameter hole to refusal and pushing the slotted 50 mm diameter PVC tube in the hole. A full PVC casing is attached to the top of the pipe. All piezometers in the swamp were installed to the base of the swamp, where auger refusal did not allow further progress. The typical installation depth is around 1m to 1.3 m. The bore installed in sandstone is 8. 5 m deep as shown on Figure 7, it is installed with 50 mm diameter PVC screen 3-meter length at the bottom, and extended with casing to the top. The top was sealed by grout, and a steel monument constructed to protect the bore. This information has been added to the manuscript Line10-17/Pg7.

Comment:Include a section on statistics and software used Response:For simple statistical analysis, an XLStat software package was used for analysis of moisture content, precipitation and organic matter content. Barnes and Allison (1988) model was setup

in R, an integrated set of software facilities for data manipulation, linear and non-linear modelling and graphical display. This information is added to the manuscript Lines1-3/Pg9. Comment:Page 7 Line 5: Was a Russian D corer used to recover samples? If not how were samples recovered intact from a conventional auger? Response:Yes, Russian D corer was used to recover samples. This information has been added to the manuscript Line15/Pg6.

Comment:Page 7 Line 12: Swamp groundwater or regional groundwater? How were sandstone aquifer samples collected? Was the existing piezometer drilled within the bedrock? ..to enable comparison with the swamp groundwater.

Response:This has been clarified in the manuscript. One sandstone aquifer sample was collected by emptying three well volume and then sampling, as described in the original manuscript Pg7/Line21.

Results:

Comment:Page 10 Line 14-15: wouldn't this just be collected rainwater?

Response: Yes, very likely the quick swamp groundwater levels rise is due to direct rainfall.

Comment:Page 10 Line20: This sentence would be better placed within the methods

Response:Thanks, this sentence was moved to Methods 3.2 Section.

Comment:Page 11 Line 7: It also may be the result of lateral throughflow along the longitudinal gradient, particularly within the sandy units

Response: Thanks, this was added to the manuscript Pg 11/Line9.

Comment:Page 15 Line 8: Or that the surface water sample points are located in the discharge zone for groundwater flow

Response:Thanks, the mixing of surface water with lateral regional groundwater is

likely occurring in the groundwater discharge zone where surface samples were collected. This was added to the manuscript Pg. 15/Line 9.

Comment:Page 15 Lines 10-14: Figure caption is confusing. Change groundwater terminology.

Response:Changed terminology to swamp and regional groundwater as per response to general comments. Changes were made to Figure 7 legend to avoid confusion.

Comment:Page 16 Line 28: Probably should be explained in the methods

Response:The Barnes and Allison (1988) model parameters were moved to Methods section Pg 9/Line 6-9

Comment:Page 17 Line 5: In that case it would be informative to relate enrichment to relative humidity to assess whether that has more influence on evaporation than temperature

Response:Barnes and Allison model does not specifically include humidity in the evaporation calculation. However, indirectly the effective diffusivities of isotopes are dependent on water content, and isotopes can diffuse in the vapour phase even without humidity gradient. The isotopic composition changes with depth by taking into consideration changes in water content. Where evaporation is proceeding, the production of heavy isotopes is affected by diffusion of water vapour and the kinetic effect includes the humidity factor. As described in Barnes and Allison: The kinetic effect is due to slightly different rates of diffusion of the different isotopic species through the 'atmospheric boundary layer'. In our case this is the unsaturated space in the pores. In the atmosphere it is affected by relative humidity and thus the degree of kinetic fractionation is affected by turbulence. The turbulence was one of the parameters used in the evaporation estimate.

Discussion:

Comment:Page 18 Line 10: This sentence should be combined with line 11 below to

strengthen this argument. As it is, the sentence hangs without supportive evidence

Response:Thanks, the sentence has been linked to the sentence on Line 11 to strengthen the argument.

Comment:Line 14: I'm not sure this statement holds up. Long water residence times within the swamp water table may be occurring to sustain this vegetation community

Response:We agree, the vegetation community is sustained by swamp groundwater, however swamp groundwater is maintained by regional groundwater (in addition to rainfall) in particular during dry periods and, as the reviewer suggests, in case where residence times are long. The sentence on line 14 indicates that consistency of vegetation during different weather periods, stable water levels and Holocene swamp sediment age all confirm that swamp system interacts with regional groundwater.

Comment:Line 14-measurement of groundwater. Measurement of groundwater levels is not evidence of aquifer connectivity. Consistency of swamp water tables and lack of significant drawdowns in dry periods may however be linked to aquifer connectivity. See Cowley et al 2018 "The hydrological function of upland swamps in eastern Australia: The role of geomorphic condition in regulating water storage and discharge"

Response:True, the wording has been changed to reflect the importance of consistency of water table and not the water level measurement. The sentence was corrected Pg20, Line 13.

Comment:Line 18: Again this statement does not represent evidence of groundwater interactions per se. It is speculation. Reword

Response:The sentence has been reworded, Pg20 Line 17-19

Comment:Line 20: Measurement of GW levels above & below sandstone is not an indication of connection. GW level comparison of both aquifers may be, as might be comparison of isotopic signatures. Reword.

[Figure]

Response:Thanks, the sentence has been reworded to indicate that comparison of groundwater levels indicates possible connection.

Comment:Line 22: Rapid infiltration and discharge of what? Swamp or sandstone aquifer?

Response:Rapid infiltration and discharge of swamp groundwater, - This sentence has been corrected.

Comment:Line 22: Where are measurements of groundwater salinity?

Response:The measurements of major ion composition and salinity are published separately in David et al, 2018. This has been updated in the manuscript.

Comment:Line 23: "resulting from limited leaching of salts from the swamp".Not sure you have the evidence for this statement

Response:This part of sentence-"resulting from limited leaching of salts from the swamp"- was deleted

Comment:Line 23: "recharge of the groundwater table". Swamp or sandstone aquifer?

Response:This is clarified to read: recharge of swamp groundwater system.

Comment:Line 29: reference required for EC & pH results

Response:Reference has been added in the sentence - David et al, 2018

Comment:Page 22: Line 4: groundwater from sandstone aquifer? swamp groundwater?

Response:Thanks, this part of sentence has been reworded as follows: -consistent with regional groundwater value

Comment:Line 9: 'Isotopic signature' of precipitation?

Response:Clarification has been made to this part of sentence as follows: too small to

result in any difference in pore water isotopic signature

Comment:Line 14: A cross section of underlying hydrogeology would add to this conceptual model of swamp hydrology

Response:We agree that the underlying hydrogeology would add to the conceptual model, however at this stage the detailed hydrogeology and evidence of interlayering of sandstone with thin siltstone in the Burralow Formation in these swamps is not available. Regional hydrogeology is described in Section 2. From shallow coring we do know that sandstone directly underlies the swamp, it is expected that more information will become available in time.

Technical Corrections Abstract:

Comment:Line 6: Add 'Endangered' before the word ecological and 'Under state and federal legislation' after communities

Response:Thanks, we have added the suggested wording.

Comment:Page 5 Figure 1: An Aerial photo or satellite base map would be better to define swamp boundaries than a topographic map.

Response:This is a good suggestion, and we have also prepared Figure 1 using the satellite map. However, the satellite map does not allow reader to appreciate the elevation changes in the swamp both in the downgradient direction and across the swamp. As a result, we have adopted the topographic map.

Comment:Page 7 line 13: Space needed between 'were' and 'described'

Response:Thanks, space was added.

Comment:Page 10 Figure 4: Where are the profiles and sediment logs for GGSWG swamp?

Response:Profile and sediment logs for GGSW swamp have been drawn and a new
figure was added to the manuscript, Figure 5.

Comment:Page 11 Figure 5: These charts may be better shown by putting the sampling periods together on one graph rather than separating the swamps. That would make it easier to flip between then and the rainfall charts. Putting sediment logs down the left-hand side may make comparisons between sediment, moisture content and organic matter easier

Response:Original Figure 5 (now Figure 6) has been updated based on review's suggestion. The sampling periods were separated, such that each is given on one graph for all three swamps. Sediment logs were already provided in Figures 3 to 5, so it would be a repetition to add them below new Figure 6.

Comment:Page 15 Figure 7: Why are the surface water sample points low down in the depth profile in c and d but at the surface in a and b? Put them all at the surface

Response:Thanks , this has been corrected and surface water samples are now all plotting at the top where they should be.

Comment:Page 18 Line 21: THPSS Page 19 Figure 9: Change the colour of the Medium to fine grained sand/clayey sand unit. It appears at first glance to be indicating a water table

Response:Thanks, to improve the clarity we have added the "water table "to the legend, as we considered that changing the colour of sand/clayey sand unit did not achieve this clarity.

Comment:Page 19 Line 14: Explain ETc

Response:The ETc is crop evapotranspiration which incorporates incorporates the ground cover, canopy properties and aerodynamic resistance of the particular crop into the calculation This definition been added to the manuscript.

Comment:Page 20 Line 7: space between day and is

Response:Thanks, space added.

Comment:Page 21 Figure 10: This graphic does not effectively display the data in table 2. A simple column graph may be more effective. I don't understand why you used 2018 dates. Would it not be better to use sampling period dates? You need to explain why these dates were chosen

Response:Thanks, we understand that the graphic maybe confusing compared to Table 2. The error in dates was corrected. We have looked at the option of using the column graph, but can not use this graph type as the values do not represent a range but one data point. Therefore, we have updated the Table 2 to clarify the values so that it is easier to see. We have also updated the graph such that legend clearly shows what ET is presented in the graph.

Comment:Page 23 Line 9: Missing word after 'relatively'

Response:Thanks, this has been reworded.

Comment:Line 15: Gorissen reference should go after the word 'ecosystem'. Insert 'this' before 'ecological'

Response:Reworded and changed as per reviewer's comments.

Comment:Page 32-33 Table 3: should be Table 1. It's difficult to determine which numbers pertain to which parameter. Either move the parameters or put borders around columns and rows. Move column 3 down do that the first 'peat' is in lone with the first bulk density number

Response:Thanks, Table 1 has been amended to make it easier to determine which numbers belong to which property.

Comment:Page 33: Table 4 should be table 2

Response:Thanks, this is amended now.

References:

Comment:References cited in text that are not in reference list: Huneau et al, 2003, Hendry et al 2013, Hendry, 2008, Hunt et al, 1996, Mandl et al, 2017, Bickford and Gell, 2005, Middleton and Kleinebecker, 2012, Johnson, 2006, Valentin et al, 2005, Gatt, 1996, Dansgaard, 1964, Linacre et al, 1967, Mathieu and Bariac, 1996, dePaolo et al 2012.

Response:All references missing from the list, or where incorrect year was attributed, have now been added to the list of references and checked in the manuscript. References in reference list not cited in text: Andersen, M., Barron, O., Bond, N., Burrows, R., Eberhard, S., Emelyanova, I., Fensham, R., Froend, R., Kennard, M., Marsh, N., Pettit, N., Rossini, R., Rutlidge, R., Valdez, D. and Ward, D.: Research to inform the assessment of ecohydrological 10 responses to coal seam gas extraction and coal mining. Department of the Environment and Energy, Commonwealth of Australia, 2016. Bukata, B.J., Osborne, T.Z. and Szafraniec, M.I.: Soil nutrient assessment and characterisation in a degraded Cnetral Florida 30 Swamp. Water Air Soil Pollut, 226: 307, 2015. Centennial Coal.: Springvale Mine Extension project, Environmental Impact Statement, Available online http://majorprojects.planning.nsw.gov.au/index.pl?action=view_job&job_id=5594 Accessed on 20 April 2017, 2014. Cloern, J. E., Canuel, E. A. and Harris, D.: Stable carbon and nitrogen isotope composition of aquatic and terrestrial plants of the San Francisco Bay estuarine system. Limnology and Oceanography, 47(3): 713-729, 2002. Denk, T. R. A., Mohn, J., Decock, C., Lewicka-Szczebak, D., Harris, E., Butterbach- Bahl, K., Kiese, R. and Wolf, B. D.: The nitrogen cycle: A review of isotope effects and isotope modeling approaches. Soil Biology and Biochemistry 105: 121-137, 2017. Deegan, L. A. and Garritt, R.H.: Evidence for spatial variability in estuarine food webs. Marine Ecology Progress Series 147: 31-47, 1997. Fry, B.: Stable isotope ecology, Springer, LA, pp 317, 2006. Gardner, W.H.: Water content. In: (ed. A. Klute) Methods of Soil Analysis, Part 1. Bibliography 165 Physical and

[Figure]

Mineralogical Methods. Agronomy Monograph No.9 (2nd edn). pp493- 544, 1986. Gorissen, S., Mallinson, J.Greenlees, M. and Shine,R.: The impact of fire regimes on populations of an endangered lizard in montane south-esastern Australia. Austral. Ecol. 40, 170-177, 2015. Heaton, T.H.E.: Isotopic studies of nitrogen pollution in the hydrosphere and atmosphere: a review. Chem. Geol. 59, 87-102, 20 1986. Huneau, F, Blavoux, B, Aeschbach–Hertig, W. and Kipfer, R.: Paleogroundwater of the Valreas Miocene aquifer (Southeastern France) as archives of the lgm/Holocene transition in the western Mediterranean region, IAEA report, IAEA-CN-80/24, 2013. Johnson, D.: Sacred waters: the story of the Blue Mountains gully traditional owners. Broadway, N.S.W.: Halstead Press, 237 pp, 2007. Liu, Y., Sheng, L. and Liu, J.P.: Impact of wetland change on local climate in semi-arid zone of Northeast China. Chinese Geographical Science. 25,309-320, 2015. Mandl, M. B., Shuman, B. N., Marsicek, J., Grigg, L.: Estimating the regional climate signal in a late Pleistocene and early Holocene lake-sediment _18O record from Vermont, USA. Quaternary Research (United States) 86(1): 67-78, 2016. Potter, N.J., Chiew, F.H.S., Frost, A.J., Srikanthan, R., McMahon, T.A., Peel, M.C. and Austin, J.M.: Characterisation of 20 recent rainfall and runoff in the Murray-Darling Basin. A report to the Australian Government from the CSIRO Murray-Darling Basin Sustainable Yields Project. CSIRO, Australia. 40pp, 2008. Rau G.H.: Carbon-13/carbon-12 variation in subalpine lake aquatic insects: Food source implications. Can. J. Fish. Aquat. Sci 37: 742-746, 1980. Reddy, K.R. and DeLaune, R.L.: Biogeochemistry of wetlands science and applications. Boca Raton, FL: CRC Press, pp774, 30 2008. Zhang, X., Sigman, D.M., Morel,F.M.M>, Kraepiel, A.M.L.: Nitrogen isotope fractionation by alternative nitrogases and past ocean anoxia. Proceedings of the National Academy of Sciences of the United States of America 111, 4782-4787, 2014.

Response:References which were not mentioned in the text were excluded from the reference list.

237, 2018.

[Figure]

**Fig. 1.**

(a) (b) (c) (d)

Depth below ground (cm)

Moisture content (%)   Moisture content (%)   Moisture content (%)   Organic matter (%)

**Legend (a):**
- GGEG1
- GGEG2
- GGEG3
- CCG3
- CCG2

**Legend (b):**
- GGSWG4
- CCG3
- CCG2
- GGEG4

**Legend (c):**
- CCG3
- CCG2
- GGEG1
- GGEG2
- GGEG4
- GGSWG2
- GGSWG1

**Legend (d):**
- CCG2
- CCG3
- GGEG3
- GGEG4
- GGEG2
- GGEG1

**Fig. 2.**

[Figure]

[Figure]

Fig. 3.

South

Upper edge
of swamp

Rainfall

North

Lower edge of swamp
Outcropping sandstone

Lateral
groundwater
flow

Not to scale

LEGEND

Organic rich soil

Sandy clay/silt

Medium to fine grained
sand/clayey sand

Medium to coarse grained sand

Sandstone

— – Water table

Local groundwater flow direction

Regional groundwater flow in sandstone

**Fig. 4.**

[Figure]

Water balance (mm)

100

0

-100

**Groundwater
contribution**

-200

-300

| ET$_C$ | | E= 4 mm/day | | E=9 mm/day | |
|---|---|---|---|---|---|
| ● | GGSW | ★ | GGSW | ▲ | CC |
| ● | GG | ★ | CC | ▲ | GG |
| ● | CC | ★ | GG | ▲ | GGSW |

Feb-16    Mar-16    Apr-16    May-16

**Fig. 5.**

---

## Referee Comment (RC3) · Anonymous Referee #3 · 21 Jun 2018

This review follows the assessment of Anonymous Reviewers 1 and 2 – this reviewer concurs with their suggestions and adds only the following comments.

I must reject this paper due to serious concerns about the accuracy of porewater isotope analytical methods. If there is no clear confidence in the analytical isotopic results the subsequent modeling does not matter.

This paper, if revised and resubmitted, further requires hard editing and a lot of trimming. Please conduct a thorough review for basic grammar and sentence structure. Check for imprecise or vague terminology usage. Please consider reducing the length – in many places there is unnecessary "filler text" (i.e. "International publications/example . . . why not just say "research has shown. . . (refs). Remove a lot of the ancillary information (detailed lithology) that is not explicitly needed for your objectives

of using pore water isotopes.

The entire Methods section, upon which this work hinges entirely, is insufficiently described or referenced. For example page 7 lines 13-17 – no citations are give for this sampling methodology. Section 3.2 needs to be entirely re-written – the analytical descriptions are incoherent. You need to give the delta values of all calibration standards. There is insufficient detail given to give confidence in the results.

I have grave doubts about the results for porewater stable isotopes and suspect the trends (or differences) between samplings may be due to evaporation artifacts. What was neglected to mention is the time lapsed between core sample collection (coring, stored in Ziplok) and the sample preparation (ie inflation) for isotope analysis. Was this storage hours, weeks, days? Ziplok bags are only good for a couple of days before evaporative loss occurs. If variable periods of times elapsed for the samplings, the samples could have been subjected to differential evaporative loss (ie why is the groundwater isotopic composition constant). There were no gravimetrics controls used, nor isotopic field controls to give confidence in this method (at least as it is described).

The Los Gatos "standards" used are not certified RMs, and should never be used for calibration. They have been revised at least 5 times due to improper storage (at LGR) in the past years.

SMOW / VSMOW do not exist – VSMOW2 does. Which was it?

The introduction is too long – suggest deleting lines Page 1, lines 12-18 (unrelated to wetlands)

Continuous line numbering would have been useful for reviewers.

Many places have a "the" or "a" added or missing. (i.e. title, Page 3 line 1, etc.).

Page 1 line 4, not climate change (aka CO2)-> rather, paleoclimate.

Page 4, line 8 "Following such extreme setttings. . . ( what does that mean?)

Page 4, line 16-20 – please rewrite the objectives in a clearer manner. A hypothesis would be a good place to start.

Page 8 line 25-26 - IRMS cannot be an LGR analyser!

Figure 6 caption error and use of d18O/d2H – the slash suggests 'or' when you mean 'and'. Superscripts missing. Suggest using the same Y-axis scaling on all figures. Why are the symbols for the same thing different in each panel? Very confusing to look at and compare!

---

## Author Comment (AC3) · 30 Jul 2018

Dear Reviewer, Thanks for the review and comments that have improved our manuscript. We have included a detailed response to the questions below in responses.

General Reviewer's Comment This review follows the assessment of Anonymous Reviewers 1 and 2 – this reviewer concurs with their suggestions and adds only the following comments. I must reject this paper due to serious concerns about the accuracy of porewater isotope analytical methods. If there is no clear confidence in the analytical isotopic results the subsequent modeling does not matter.

Response Thanks, it is acknowledged that the description of porewater isotope analytical methods was relatively brief. This section has now been improved and expanded to include additional details on the analytical method sampling technique. Section 3 provides more information on accuracy and confidence. This section has also has been updated based on comments from Reviewer 1 and 2.

General Reviewer's Comment This paper, if revised and resubmitted, further requires hard editing and a lot of trimming. Please conduct a thorough review for basic grammar and sentence structure. Check for imprecise or vague terminology usage. Please consider reducing the length – in many places there is unnecessary "filler text" (i.e. "International publications/ example : : : why not just say "research has shown: : : (refs). Remove a lot of the ancillary information (detailed lithology) that is not explicitly needed for your objectives of using pore water isotopes.

Response Thanks for the comments, further editing has been undertaken on this paper, and grammar and sentence structure checked. However, we respectfully disagree with the suggestion to trim the text. Our response is justified by Reviewer 1 who requested more ancillary information (detailed lithology), who for example requested that an additional cross-section be prepared to help with understanding the lithology. We have therefore added more information and additional cross-section in response to Reviewer 1 comments. We have reduced the length and removed the text which does not related to wetlands in the Introduction section. Please note some of the text is necessary to understand the background, and follows the response to Reviewer2. For example, the "international publications" is not a "filler text", the difference between the international and local literature is important as the swamp systems in Australia have formed under very different climate conditions compared to other swamps in the Northern hemisphere as discussed in Fryirs et al. (2014). This has resulted in different formation and evolution model for the upland swamps on sandstone in Eastern Australia. For all other items of discussion in the paper, the reviewer 3 comments were used to correct the text. The text in the manuscript has been reduced and edited as recommended by Reviewer.

General Reviewer's Comment - Method's section The entire Methods section, upon which this work hinges entirely, is insufficiently described or referenced. For example, page 7 lines 13-17 – no citations are give for this sampling methodology.

Response Thanks, this is acknowledged, and the entire Methods section and Section 3.1 has been considerably expanded and rewritten based on Reviewer's suggestion and suggestions from Reviewers 1 and 2 (please see below). For example, the citation is now provided for the sampling methodology (Wassenaar et al (2008) and Hendry et al (2015)).

"3.1 Fieldwork and sampling The fieldwork was undertaken during 2016 and 2017 with the swamps in a natural state and recovered from earlier wildfire in 2013. The first sampling event 24th to 25th May 2016 occurred following an extremely dry weather period of four months below the long-term average rainfall (BoM Lithgow Station 0630132, 900 m AHD, 13 km SW of the study area with 139 years of data records) for February to April (46.6 mm, 36.8 mm, 6.6 mm and 20.8 mm for each of the months). A total of 34 pore water samples and 5 surface and groundwater samples were collected. A repeat sampling on 25th to 26th October 2016 occurred after four months of above average rainfall from June to September (170.2 mm, 102 mm, 61.8 mm and 92 mm for each of the months). During October 2016 sampling event 14 pore water samples and 13 surface and groundwater samples were collected. Sampling on 30th May 2017 occurred under different climate conditions with both above and below average rainfall trend in the months preceding the sampling event. A total of 27 pore water samples and 6 surface and groundwater samples were collected in May 2017. The spatial depth resolution varied from 10 to 20 cm depending on the penetration of the corer. Figure 2 shows the variation in monthly long-term rainfall (139 years) and comparison with rainfall in 2016 and 2017. In total seven sediment cores were obtained by coring using a Russian D hand corer (40 mm diameter) to rock refusal (between 0.45 to 1.4 m), and three transects (CC, GG and GGSW) were prepared along the length of the swamps (Figures 3,4 and 5). The samples were geologically logged after extraction,

by noting the lithology, grain size and roundness, matrix and colour. The hand cored holes were restored by returning soil material to the hole immediately after sampling. This was undertaken to ensure no change occurred in this endangered and protected ecological system as a result of sampling. The coring on CC transect was repeated in October 2016 at a distance of less than 0.5 m from the original hole. The coring locations were selected to represent the swamp stratigraphy from upstream to downstream and to provide a spatial coverage across the three swamps. In addition, three cored locations were selected such that they were adjacent to an existing piezometer (CCG1 on transect CC and GGEG2A and GGEG4 on GG transect). The purpose of this, in addition to determining the stable isotope profiles, was to enable comparison with the swamp groundwater measurements and to collect regional groundwater sample from the underlying sandstone aquifer where possible. Sediment cores were divided into subsamples of 10-20 cm length, were packed into Ziplock bags and kept in cool storage for later analysis of moisture content and organic matter content. The samples for pore water analysis were temporarily double packed in ziplock bags by minimising the airspace in the bag, stored in the cooled ice box in accordance with the sampling protocol developed by Wassenaar et al (2008) and further improved by Hendry et al (2015). The same afternoon after collection, samples were packed in tough high-grade food storage plastic bags with air extracted, double sealed, separately stored in an additional plastic bag and were kept at a temperature of $4°C$ to prevent evaporation. Vacuum packing was required to minimise atmospheric moisture contamination. Storage time for samples after collection was 3 days in the cool environment ($4°C$) before they were analysed. All isotopic field controls during sampling and analysis were implemented; this included: quick storage in tough plastic bags, immediate double bagging during collection and vacuum packing the same afternoon with storage at $4°C$. Swamp groundwater was sampled directly from the cored hole, field parameters were measured immediately (pH, EC, DO, temperature) and samples field filtered (0.45 micron). This was repeated for all three sampling events; however, some bores were dry and some not accessible. Swamp groundwater and regional groundwater from existing piezometers (CCG1, GGEG2, GGEG5x, GGEG5 and GGSWG1) was gauged and sampled by bailing three volumes and then the same procedure was followed as for the cored holes. Swamp and sandstone piezometers were installed by the mining company prior to our research study. The swamp piezometers were installed to the base of the swamp, where auger refusal did not allow further progress. The typical installation depth was around 1 m to 1.3 m. To minimise disturbance of the swamp, all swamp piezometers were installed by manual coring an 80 mm diameter hole to refusal and pushing the slotted 50 mm diameter PVC tube into the hole. A full PVC casing was attached to the top of the pipe. The piezometer installed in the sandstone is 10.7 m depth with 50 mm diameter PVC casing that includes a 3-meter length of screen at the bottom of the hole. The piezometer installation was extended with casing to the top. The top was sealed by grout, and a steel monument constructed for protection. Surface water samples were collected at the downgradient end of the swamp but also at one upgradient location (GGES2) where this was possible. For this study ANSTO provided event based $\delta 18O$ and $\delta 2H$ data for precipitation from Mt Werong for the period covered in this research. Mt Werong (Hughes and Crawford, 2013) is located around 70 km south of this research site, however, within the same climatic environment and at similar elevation to the investigated swamps."

Reviewer's Comment -Section 3.2 needs to be entirely re-written – the analytical descriptions are incoherent. You need to give the delta values of all calibration standards. There is insufficient detail given to give confidence in the results.

Response Thanks, Section 3.2 was considerably expanded and fully re-written with additional detailed information on the sample analysis, secondary standards used along with their delta values and primary standard and reproducibility of the results. Please see below for fully revised Section 3.2.

"3.2 Sample analysis The swamp sediment samples were analysed for $\delta 18O$ and $\delta 2H$ by H2O(water)-H2O(vapour) porewater equilibration (Wassenaar et al, 2008; Wassenaar and Hendry, 2008) and off-axis ICOS. The Los Gatos (LGR) water vapour analyser (WVIA RMT-EP model 911-0004) located at UNSW, Australia was used for sample analysis. All samples and standards have been stored at 4°C prior to the analysis, and all have been allowed the same time on the laboratory bench in the temperature controlled laboratory. All samples have followed the same treatment. Samples (n=34, 14 and 27 for each of the sampling events) were prepared in the lab by transferring the sample to a tough clean zip lock bag. The 1 L sample bags were inflated with dry air and left on the laboratory bench at the controlled temperature for a period of between 17 to 24 hours to allow vapour equilibration. The timing of vapour equilibration is dependent on the compactness of the core sample, whether it is broken in pieces and if it is unconsolidated (Wassenaar et al, 2008). The timing varies for different geologic materials and must be determined experimentally (Hendry et al, 2015) for each material. Work by Wassenaar et al (2008) and David et al (2015) indicates that for compact, low permeability, consolidated materials around 3 days is required for core samples equilibration. The samples in this research are broken down, unconsolidated, saturated and high permeability therefore shorter equilibration time is considered justified. In addition, the optimal equilibration time in this research is considered to be achieved when headspace water content of 23,000 to 28,000 ppm $H_2O$ was measured in the bag. This headspace water content is important for accurate sampling (Hendry et al, 2015). Once the sample has reached complete isotopic equilibrium, the vapour was collected by perforating the bag containing sample with a sharp needle and transferring it directly from the bag to the LGR vapour analyser. The connection between the needle and the LGR inlet fitting was via the flexible, thick wall, soft plastic tube, fitted tightly with fittings on both sides. The tight fitting was required to limit the atmospheric air ingress into the LGR. The contamination by atmospheric air is considered negligible. This is based on the measurement of ambient air moisture of around 14,000 to 15,000 ppm, while the headspace for samples had a range of 23,000 to 28,000 ppm $H_2O$. The analysis of the vapour sample was undertaken along with the standards (1 ml) prepared in the similar manner to the core samples. The equilibration time for standards was around 20 minutes. A new set of three standards (one primary and two secondary) were run

after every third sample. It is not possible to sample the headspace repeatedly, as 1 L headspace only allows sampling once. Repeated inflating with dry air results in incorrect readings. Following each set of samples and standards, the analysis was suspended for a period of around 10-15 minutes, to allow the LGR to reach the stable atmospheric air readings and reduce any memory effect. The linear regression was used between $\delta$18O and $\delta$2H vapour values for standards (expressed as water using the fractionation factor) and liquid standards. The regression was used to calibrate the vapour results for samples. Calibration was undertaken with two secondary $\delta$18O and $\delta$2H standards (Los Gatos 2A -16.14‰ $\delta$18O and -123.6‰ $\delta$2H and 5A -2.80‰ $\delta$18O and 9.5‰ $\delta$2H) and normalised with one primary V-SMOW2 standard run during the analysis. LGR standards were stored in accordance with the protocol, at 4°C, and after usage the top was fully sealed to prevent any exchange with the atmosphere. The use of LRG as secondary standards has been used in other studies such as Penna et al (2010) on reproducibility and repeatability of the laser absorption spectroscopy measurements and was found that LGR standards performed satisfactory. Replicate sample analyses using direct vapour equilibration method (mean difference of 6 samples) indicate reproducibility of results in our research within 0.68‰ $\delta$2H and 0.04‰ $\delta$18O uncertainty. Reported instrument precision of 0.5‰ $\delta$2H and 0.15‰ $\delta$18O over 10 seconds and drift of 0.75‰$$\delta$2H and 0.3‰ $\delta$18O over 15 minutes was minimised by correcting the readings. The data set for each sample was corrected for drift by back correction of standards within each set and then applying the same regression analysis to the relevant samples. For each sample the standard deviation and instrument drift error were calculated. Following the standard operating procedures, the precision in this research was 0.6‰ $\delta$2H and 0.23‰ $\delta$18O over 70 seconds. Hendry et al (2015) report the analytical precision of the vapour equilibration method ($\pm$0.40‰ for $\delta$18O and $\pm$2.1‰ for $\delta$2H) to be comparable or better than physical extraction from cores using high-speed centrifugation, cryogenic micro-distillation, azeotropic and microwave distillation or isotope ratio mass spectrometry (IRMS) based direct equilibration methods as discussed in Kelln et al (2001). Based on work by Allison and Hughes (1983) and

Raves and Woods (1990), the direct vapour equilibration method is better than results obtained by physical and chemical extractions (Hendry et al, 2015). This is achieved through limiting fractionation losses by short storage time, single procedure once the samples are in the laboratory and use of standards and water isotopic data as a cross check. The advantage of the method is that it is particularly suitable for samples with high moisture content which shortens the equilibration time. Water samples (surface water, swamp water and groundwater, n=21) were analysed for $\delta18O$ and $\delta2H$ by the off axis- integrated cavity output spectrometry (OC-ICOS) technique using an LGR analyser located at UNSW Australia. Two secondary standards and V-SMOW2 standard were used to calibrate and normalise the samples. Gravimetric water content (ASTM D2974-14, 2014 and ASTM D2216-10, 2010 was measured by weighing the sample (n=70), drying at 100°C for 24 hours and re-weighing (Reynolds, 1970). The 100% gravimetric water content relates to water holding capacity and organic content of the material. The analysis was undertaken at the School of Mining Engineering, UNSW Australia. Organic matter content was measured by loss on ignition method (LOI), by weighing (following initial drying at 100°C) and drying in furnace oven at 550°C (Heiri et al, 2001). The analysis was conducted at the Water Research Laboratory, UNSW Australia. Precipitation samples were analysed at the ANSTO Environmental Isotope Laboratory using a cavity ring-down spectroscopy method on a Picarro L2120-I Water Analyser (reported accuracy of $\pm1.0$, $\pm0.2$‰ for $\delta2H$ and $\delta18O$ respectively). The lab runs a minimum of two in-house standards calibrated against VSMOW/VSMOW2 and SLAP/SLAP2 with samples in each batch. For simple statistical analysis of moisture content, precipitation and organic matter content, an XLStat software package (XL-Stat, 2017) was used. The Barnes and Allison (1988) model was implemented for this project using R (R core team, 2013), to investigate the evaporative losses based on isotopic composition of water. For the Barnes and Allison (1988) model volumetric water content was calculated from the measured gravimetric water content and bulk density. Bulk density was obtained from known lithology and measured data (Cowley et al, 2016) and porosity data from a swamp study by Walzsak et al (2002). To estimate

effective liquid diffusivity of isotopes, tortuosity values were obtained from the literature (Maidment, 1993; Shackelford and Daniel, 1991; Barnes and Allison, 1988)."

General Reviewer's Comment I have grave doubts about the results for porewater stable isotopes and suspect the trends (or differences) between samplings may be due to evaporation artifacts. What was neglected to mention is the time lapsed between core sample collection (coring, stored in Ziplok) and the sample preparation (ie inflation) for isotope analysis. Was this storage hours, weeks, days? Ziplok bags are only good for a couple of days before evaporative loss occurs.

Response Thanks, the updated Section 3.1 and 3.2 explains how samples were collected and prepared and how any potential evaporation was avoided during sampling, storage and preparation for and during the analysis. Samples were collected in accordance with the methods reported in Wassenaar et al (2008), they were stored at 4°C for three days after collection (coring) and were then prepared following methods of Hendry et al (2015), which improved on methods developed by Wassenaar et al (2008). The use of clear Ziplock bags, Isopack and clear bags for storage of samples for pore water analysis has been found (Hendry et al 2015) to result in evaporation loss and isotopic fractionation after 10-15 days after sample collection. In our research, we stored samples for only 3 days before analysis, in tough plastic bags (rather than Ziplock bags) sealed twice on each side after extracting air. Additionally, each vacuum packed (packed in the tough high-grade food plastic bag with air extracted) and double sealed sample was placed in an additional tough plastic bag with air space removed. Therefore, it is improbable that the results were artefacts of the storage process.

General Reviewer's Comment If variable periods of times elapsed for the samplings, the samples could have been subjected to differential evaporative loss (ie why is the groundwater isotopic composition constant). There were no gravimetrics controls used, nor isotopic field controls to give confidence in this method (at least as it is described).

Response We agree with the reviewer that different time periods were subject to different evaporation losses. This is observed in difference between the samples collected in May 2016 after dry and warm period compared to October 2016 after wet and cool period. All isotopic field controls during sampling and analysis were implemented; this included: quick storage in tough plastic bags, immediate double bagging during collection, vacuum packing with double seals in tough plastic food grade bags and double sealing. It is assumed that Reviewer relates gravimetric control to the weight of sample taken in the field. In the field the sample size was not weighted, as this was not considered important. What was considered important was that there was enough sample which will have a minimum of 5% moisture content to allow vapour equilibration method to be used. Water samples were also collected and analysed for cross refence with pore water. The isotopic field controls were added to Section 3.1 of the manuscript.

General Reviewer's Comment The Los Gatos "standards" used are not certified RMs, and should never be used for calibration. They have been revised at least 5 times due to improper storage (at LGR) in the past years.

Response: We used V-SMOW2 as primary reference standard, and Los Gatos standards as secondary. We are aware that there is SLAP which is the secondary standard distributed by IAEA, but have not used it. LGR standards in our lab were stored properly, at 4°C, and after usage the top was fully sealed to prevent any exchange with the atmosphere (opened once prior to this analysis). The use of LRG as secondary standards has been used in other studies such as Penna et al (2010) which undertook a study on reproducibility and repeatability of the laser absorption spectroscopy measurements and found LGR standards to work satisfactorily. In addition, the water samples analysed separately in a different LGR apparatus with in house standards and V-SMOW2 as the primary standard, returned the $\delta2H$ and $\delta18O$ similar to the pore water results, confirming the accuracy and calibration. Text has been added to manuscript in Section 3.1 to explain the LGR storage protocol was followed in the lab.

Reviewer's Comment SMOW / VSMOW do not exist – VSMOW2 does. Which was it?

Response It was V-SMOW2, this is now corrected in the manuscript.

Reviewer's Comment The introduction is too long – suggest deleting lines Page 1, lines 12-18 (unrelated to wetlands)

Response The text has been removed from Page 3, lines 12-18 as suggested.

Reviewer's Comment Continuous line numbering would have been useful for reviewers.

Response: This manuscript was prepared on the template suggested by HESS, and conversion to pdf is outside of the control of Authors of this paper.

Reviewer's Comment Many places have a "the" or "a" added or missing. (i.e. title, Page 3 line 1, etc.).

Response Thanks, "a" and "the" were added or removed, as suggested in the title and Page3 line 1, and at other places in the manuscript.

Reviewer's Comment Page 1 line 4, not climate change (aka CO2)-> rather, paleoclimate.

Response Thanks, this was reworded to paleoclimate in manuscript.

Reviewer's Comment Page 4, line 8 "Following such extreme setttings: : : ( what does that mean?)

Response: Thanks, this has been clarified as "long dry periods"

Reviewer's Comment Page 4, line 16-20 – please rewrite the objectives in a clearer manner. A hypothesis would be a good place to start.

Response: Line 16-20 related to objectives has been rewritten and hypothesis added. The paragraph now reads: "The objective of this research was to improve understanding of intact swamps under natural conditions by characterizing the sediments, waters and organic materials and developing the conceptual model for the swamp system. We hypothesise that groundwater is an important contributor to the swamp water balance

and for this we investigate for the first time using direct equilibration method the vertical profiles of stable $\delta$18O and $\delta$2H isotopes of pore water within the swamp. Groundwater contributions to pore water could be determined by comparison of end members stable isotope values. A conceptual model of the swamp water cycle was developed by combining stable isotope results with information from sediment lithology logs, organic and carbon content of sediments. "

Reviewer's Comment Page 8 line 25-26 - IRMS cannot be an LGR analyser!

Response: Thanks, this typing error is now corrected in manuscript to Off axis- integrated cavity output spectrometry (OA-ICOS).

Reviewer's Comment Figure 6 caption error and use of d18O/d2H – the slash suggests 'or' when you mean'and'. Superscripts missing. Suggest using the same Y-axis scaling on all figures. Why are the symbols for the same thing different in each panel? Very confusing to look at and compare!

Response Thanks, the caption has been changed to say $\delta$18O and $\delta$2H and superscripts added to Figure 6. Same y- scaling has been used on all figures as suggested, and all symbols for the same sample types are the same in each panel as recommended by the Reviewer3. Please see attached Figure 6 (now Figure 7).
* * *
The figure panels (a), (b), (c) show δ2H (‰) vs δ18O (‰) plots.

**(a)**
- Rainfall 61.5mm 5 months before sampling event
- y=4.24x -11.03
- Rain event 27mm two weeks before sampling
- y=4.62x -11.19
- LMWL
- Legend:
  - LMWL
  - Pore water GG May16
  - Pore water CC May16
  - Rainfall MtW big events Jan-May16
  - Rainfall MtW weighted ave Jan-May16 big events
  - Rainfall weighted average Mt Werong (2005-2017)

**(b)**
- Last rainfall event (16mm) 3 days before sampling
- LMWL
- Rainfall (25 and 17 mm) 3 weeks before sampling
- Major 109mm event 5 months before sampling -17.6‰ δ18O, -126.3‰ δ2H
- Legend:
  - Pore water GG Oct16
  - Pore water CC Oct16
  - Surface water Oct16
  - Groundwater (sandstone) Oct16
  - Swamp groundwater Oct16
  - Rainfall Mt Werong July-Oct16 big events
  - Rainfall MtW weighted ave Jun-Oct16 big events
  - LMWL
  - Rainfall weighted average Mt Werong (2005-2017)

**(c)**
- LMWL
- y=6x +9.3 R-squared = <<R2_0>>
- Second last rainfall event (36mm) 3 months before sampling
- 136mm event 2.5 months before sampling
- Last big rainfall event (26 mm) 10 days before sampling -13.7‰ δ18O, -95.1‰ δ2H
- Legend:
  - LMWL
  - Pore water GGSWG1 May17
  - Pore water GGSWG2 May17
  - Surface water May17
  - Swamp groundwater May17
  - Rainfall MtW Jan-May17 big events
  - Pore water CC May17
  - Rainfall weighted ave Jan-May17 big events
  - Rainfall weighted average Mt Werong (2005-2017)

**Fig. 1.**

---

## Author Response (AR1)

**Editor Decision: Reconsider after major revisions (further review by editor and referees) (20 Aug 2018) by Christine Stumpp**

Comments to the Author:

Three referees posted their comments and evaluated the manuscript. The referees had different opinions about the quality of the manuscript. Some of the main concerns were related to the methodology itself. In the manuscript, the methods and standard procedures for isotope analysis in the soil samples were described only briefly, and therefore it was difficult to evaluate the quality of the data.

The authors answered in detail on how soil samples were taken/stored and how the soil water isotope were analysed. It is important to include this in the revised version of the manuscript like all other concerns raised by the reviewers.

At the same time, the authors need to make sure that the manuscript gets shorter by deleting unnecessary information or being more precise with wording as outlined in the reviewer comments. According to some of the answers given, I recommend to go back to the text again and carefully check whether more improvments can be made, and I recommend putting some information into Supplementary Information.

Based on all the comments and the detailed answer about the used analytical method, the manuscript can be potentially reconsidered for publication in HESS after major revisions.

Dear Editor,

Thanks for your comments,

We have taken your and reviewers' comments into consideration and have made the requested changes.

Some of the main concerns were related to the methodology itself. In the manuscript, the methods and standard procedures for isotope analysis in the soil samples were described only briefly, and therefore it was difficult to evaluate the quality of the data.

The authors answered in detail on how soil samples were taken/stored and how the soil water isotope were analysed. It is important to include this in the revised version of the manuscript like all other concerns raised by the reviewers.

All reviewers' comments were addressed in the manuscript and the methodology and details on standard procedure provided. This included the collection and storage of samples and standards used in the analysis.

At the same time, the authors need to make sure that the manuscript gets shorter by deleting unnecessary information or being more precise with wording as outlined in the reviewer comments. According to some of the answers given, I recommend to go back to the text again and carefully check whether more improvments can be made, and I recommend putting some information into Supplementary Information.

We have removed the material from the manuscript that we considered was not needed. We also moved two graphs/figure in the supplement section and have thoroughly edited the document.

**Dear Reviewer1,**

Thanks for the very constructive review and comments provided for the manuscript. We have included a detailed response to the questions below in blue.

**Reviewer's Comment**

This paper contains a novel method for characterising the hydrology of wetlands and swamps and presents a scientifically robust model of temperate upland swamp hydrology that fits within the context of current research into similar ecosystems. It is a well written paper with high scientific significance.

**We thank the reviewer for the positive feedback.**

One issue is that the terminology for describing groundwater within the swamps and regional groundwater aquifers is not differentiated. A major part of the paper is concerned with connectivity of the swamp aquifers with regional groundwater, yet the term groundwater is used to describe both

aquifers. One way of differentiation may be to call swamp groundwaters 'swamp water' or 'swamp water table' and regional groundwater 'groundwater' or 'sandstone aquifer' or similar and use these terms consistently throughout.

Thanks, the terminology has now been improved by differentiating between swamp groundwater and regional groundwater by introducing the new terminology as suggested. The *swamp groundwater* is now used to discuss swamp groundwater and *regional groundwater* to describe regional groundwater. Changes were made throughout the document.

Another issue is that while the paper presents the application of the stable isotope direct vapour equilibration method to quantify water sources, it does not discuss this method in great detail. A paragraph (or two) to describe the data accuracy of the vapour method against the more conventional sampling method would be useful as would a more detailed discussion of the circumstances in which it could be used.

The method has been described in detailed in referenced Wassenaar et al, 2008; Wassenaar and Hendry, 2008 and Hendry et al, 2015. Several additional sentences were added to both Sections 3.1 and 3.2 (Pg 8 Line 25 to Pg 9 Line 10) of the manuscript to add to the method description. Data accuracy of the vapour method is provided on Pg 9 Line 25 to Line 35 and a comparison to the conventional sampling method and advantages of the method are provided as an additional paragraph in Pg 10 Line 0-10

**Characterising flow paths within individual sedimentary units is one area where this method would be hugely advantageous.**

We agree that characterising the flow paths in the individual sedimentary units in these particular swamps would be advantageous, however very limited groundwater data is available in the public domain. However, additional local geology and hydrogeology data is presented in Section 2 (Pg 5 Lines 22-34 to Pg 6 Line 0-8) to provide the background to understanding the context. It is expected that more groundwater data will be available in the future to characterise the flow paths in the individual units for the researched swamps.

**More detail is also required in describing methods and a description of the regional hydrogeology in the site description would be of use.**

More details are now also provided in Section 3.1 Pg 7 Line 5-10, Pg 7 Lines 21 to Pg 8 Line and Pg 8 Line 10-15, Section 3.2 Pg 8 Lines 25-34, Pg 9 Lines 0-25) on describing different methods used in this research. Description of the regional hydrogeology is now provided in Section 2: Site description (Pg 5 Lines 22-34 to Pg 6 Line 0-8), based on the limited publicly available data for these swamps.

**Specific review's comments**

**Introduction:**

Page 3 Line 27: Change the term "hydrological balance". Swamp flora and fauna are dependent on the high water tables that are characteristic in THPSS. The term hydrological balance does not adequately describe this.

As suggested by the reviewer, the term "hydrological balance" has been replaced with "high water table" at the same location in the document.

**Page 3 Line 30: The term groundwater in this instance is confusing. Do you mean swamp groundwater or groundwater from the surrounding sandstone aquifer?**

Groundwater was taken from both swamps and sandstone aquifer, but to avoid confusion the term groundwater in this sentence relates to sandstone regional groundwater which is now termed *regional groundwater*. Another term has been introduced for groundwater in swamp and this is *swamp groundwater*. As a result, the Line30/Pg3 (original document) has added term swamp groundwater to include both.

Page 4 Line 1: Again groundwater terminology is confusing. Would suggest 'swamp water levels' or 'swamp groundwater' when referring to the swamp water table and 'regional groundwater' or 'sandstone aquifer' when referring to the bedrock aquifer.

The term *swamp groundwater* has been adopted based on the reviewer's suggestion for swamp water while *regional groundwater* term describes the bedrock aquifer. This has been corrected at Page4 line 1 (in the original document) and throughout the document.

**Site Description:**

Need a description of regional hydrogeology to give a better picture of likely groundwater interactions.

Description of the regional geology and hydrogeology is provided on Pg5/Line 22 to Pg 6 Line 8. Page 6 Line 4: Does this mean the longwalls are located directly below the swamps? The longwalls are not directly underneath the swamps, but below ground to the southwest of the swamps, with the closest swamp to the longwall being GGSW. This have been reworded in the text to clarify on Pg 5 Line17-19.

**Methods, Fieldwork and Sampling:**

More details of piezometers are required. Depth, installation method, construction materials etc Details of groundwater bore required including installation method, construction materials and depth

Piezometers were installed by the mining company prior to our research study. To minimise disturbance to the swamp, all piezometers were installed by manual augering the 80-mm diameter hole to refusal and pushing the slotted 50 mm diameter PVC tube in the hole. A full PVC casing is attached to the top of the pipe. All piezometers in the swamp were installed to the base of the swamp, where auger refusal did not allow further progress. The typical installation depth is around 1m to 1.3 m. The bore installed in sandstone is 10.7 m deep as shown on Figure 7, it is installed with 50 mm diameter PVC screen 3-meter length at the bottom, and extended with casing to the top. The top was sealed by grout, and a steel monument constructed to protect the bore. This information has been added to the manuscript Line 8-16 /Pg8.

**Include a section on statistics and software used**

For simple statistical analysis, an XLStat software package was used for analysis of moisture content, precipitation and organic matter content. Barnes and Allison (1988) model was setup in R, an

integrated set of software facilities for data manipulation, linear and non-linear modelling and graphical display. This information is added to the manuscript Lines 22-29 /Pg10.

Page 7 Line 5: Was a Russian D corer used to recover samples? If not how were samples recovered intact from a conventional auger?

Yes, Russian D corer was used to recover samples. This information has been added to the manuscript Line5/Pg7.

Page 7 Line 12: Swamp groundwater or regional groundwater? How were sandstone aquifer samples collected? Was the existing piezometer drilled within the bedrock?

..to enable comparison with the *swamp groundwater*.. This has been clarified in the manuscript. One sandstone sample was collected by emptying three well volume and then sampling, the existing piezometer is installed to bedrock as described in the original manuscript Pg8/Line13.

**Results:**

Page 10 Line 14-15: wouldn't this just be collected rainwater?

Yes, very likely the quick swamp groundwater levels rise is due to direct rainfall.

Page 10 Line20: This sentence would be better placed within the methods

Thanks, this sentence was moved to Methods 3.2 Section.

Page 11 Line 7: It also may be the result of lateral throughflow along the longitudinal gradient, particularly within the sandy units

Thanks, this was added to the manuscript Pg 27/Line7.

Page 15 Line 8: Or that the surface water sample points are located in the discharge zone for groundwater flow

Thanks, the mixing of surface water with lateral regional groundwater is likely occurring in the groundwater discharge zone where surface samples were collected. This was added to the manuscript Pg. 32/Line 20.

Page 15 Lines 10-14: Figure caption is confusing. Change groundwater terminology.

Changed terminology to *swamp* and *regional groundwater* as per response to general comments. Changes were made to Figure 7 (now Figure 8) legend to avoid confusion.

Page 16 Line 28: Probably should be explained in the methods

The Barnes and Allison (1988) model parameters were moved to Methods section Pg 10/Line 22 Page 17 Line 5: In that case it would be informative to relate enrichment to relative humidity to assess whether that has more influence on evaporation than temperature

Barnes and Allison model does not specifically include humidity in the evaporation calculation. However, indirectly the effective diffusivities of isotopes are dependent on water content, and isotopes can diffuse in the vapour phase even without humidity gradient. The isotopic composition changes with depth by taking into consideration changes in water content. Where evaporation is proceeding, the production of heavy isotopes is affected by diffusion of water vapour and the kinetic effect includes the humidity factor. As described in Barnes and Allison: The kinetic effect is due to slightly different rates of diffusion of the different isotopic species through the 'atmospheric boundary layer'. In our case this is the unsaturated space in the pores. In the atmosphere it is affected by relative humidity and thus the degree of kinetic fractionation is affected by turbulence. The turbulence was one of the parameters used in the evaporation estimate.

**Discussion:**

Page 18 Line 10: This sentence should be combined with line 11 below

to strengthen this argument. As it is, the sentence hangs without supportive evidence

Thanks, the sentence has been linked to the sentence on Line 11 (original document) to strengthen the argument.

Line 14: I'm not sure this statement holds up. Long water residence times within the swamp water table may be occurring to sustain this vegetation community

We agree, the vegetation community is sustained by swamp groundwater, however swamp groundwater is maintained by regional groundwater (in addition to rainfall) in particular during dry periods and, as the reviewer suggests, in case where residence times are long. The sentence indicates that consistency of vegetation during different weather periods, stable water levels and Holocene swamp sediment age all confirm that swamp system interacts with regional groundwater as described in Pg 5 Line 22-27.

Line 14-measurement of groundwater. Measurement of groundwater levels is not evidence of aquifer connectivity. Consistency of swamp water tables and lack of significant drawdowns in dry periods may however be linked to aquifer connectivity. See Cowley et

al 2018 "The hydrological function of upland swamps in eastern Australia: The role of geomorphic condition in regulating water storage and discharge"

True, the wording has been changed to reflect the importance of consistency of water table and not the water level measurement. The sentence was corrected Pg5, Line 23.

Line 18: Again this statement does not represent evidence of groundwater interactions per se. It is speculation. Reword

The sentence has been reworded, Pg5 Line 26-28

Line 20: Measurement of GW levels above & below sandstone is not

an indication of connection. GW level comparison of both aquifers may be, as might be comparison of isotopic signatures. Reword.

Thanks, the sentence has been reworded to indicate that comparison of groundwater levels indicates possible connection. Pg 26 Line 24-26

Line 22: Rapid infiltration and discharge of what? Swamp or sandstone aquifer?

Rapid infiltration and discharge of swamp groundwater, - This sentence has been corrected. Pg26 Lines 28-29

Line 22: Where are measurements of groundwater salinity?

The measurements of major ion composition and salinity are published separately in David et al, 2018. This has been updated in the manuscript. Pg26 Lines 29

Line 23: "resulting from limited leaching of salts from the swamp".Not sure

you have the evidence for this statement

This part of sentence-"resulting from limited leaching of salts from the swamp"- was deleted

Line 23: "recharge of the groundwater table". Swamp or sandstone aquifer?

This sentence was removed as at was out of context.

Line 29: reference required for EC & pH results

Reference has been added in the sentence - David et al, 2018, Pg26 Lines 29

Page 22: Line 4: groundwater from sandstone aquifer? swamp groundwater?

Thanks, this part of sentence has been reworded as follows: -consistent with regional groundwater value Pg32, Line 26

Line 9: 'Isotopic signature' of precipitation?

Clarification has been made to this part of sentence as follows: too small to result in any difference in isotopic signature of precipitation, Pg 21 line 31

**Line 14: A cross section of underlying hydrogeology would add to this conceptual model of swamp hydrology**

We agree that the underlying hydrogeology would add to the conceptual model, however at this stage the detailed hydrogeology and evidence of interlayering of sandstone with thin siltstone in the Burralow Formation in these swamps is not available. Regional hydrogeology is described in Section 2. From shallow coring we do know that sandstone directly underlies the swamp, it is expected that more information will become available in time. Additional cross section was added as Fig 5.

**Technical Corrections**

Abstract: Line 6: Add 'Endangered' before the word ecological and 'Under state and federal legislation' after communities

**Thanks, we have added the suggested wording. Pg 1 Line 17**

**Page 5 Figure 1: An Aerial photo or satellite base map would be better to define swamp boundaries than a topographic map.**

This is a good suggestion, and we have also prepared Figure 1 using the satellite map. However, the satellite map does not allow reader to appreciate the elevation changes in the swamp both in the downgradient direction and across the swamp. As a result, we have adopted the topographic map. Page 7 line 13: Space needed between 'were' and 'described'

**Thanks, space was added.**

Page 10 Figure 4: Where are the profiles and sediment logs for GGSWG swamp?

Profile and sediment logs for GGSW swamp have been drawn and a new figure was added to the manuscript, Figure 5.

Page 11 Figure 5: These charts may be better shown by putting the sampling periods together on one graph rather than separating the swamps. That would make it easier to flip between then and the rainfall charts. Putting sediment logs down the left-hand side

may make comparisons between sediment, moisture content and organic matter easier Original Figure 5 (now Figure 6) has been updated based on review's suggestion. The sampling periods were separated, such that each is given on one graph for all three swamps. Sediment logs were already provided in Figures 3 to 5, so it would be a repetition to add them below new Figure 6.

Page 15 Figure 7: Why are the surface water sample points low down in the depth profile in c and d but at the surface in a and b? Put them all at the surface

Thanks , this has been corrected and surface water samples are now all plotting at the top where they should be. (now Figure 8)

**Page 18 Line 21: THPSS Page 19 Figure 9: Change the colour of the Medium to fine grained sand/clayey sand unit. It appears at first glance to be indicating a water table**

Thanks, to improve the clarity we have added the "water table "to the legend, as we considered that changing the colour of sand/clayey sand unit did not achieve this clarity.

**Page 19 Line 14: Explain ETc**

Crop evapotranspiration ( $ET_c$ ) calculation incorporates the ground cover, canopy properties and aerodynamic resistance for the specific crop into the calculation. This definition been added to the manuscript. Pg 25 Line 0-2

Page 20 Line 7: space between day and is

Thanks, space added.

Page 21 Figure 10: This graphic does not effectively display the data in table 2. A simple column graph may be more effective. I don't understand why you used 2018 dates. Would it not be better to use sampling period dates? You need to explain why these dates were chosen Thanks, we understand that the graphic maybe confusing compared to Table 2. The error in dates was corrected. We have looked at the option of using the column graph, but can not use this graph

type as the values do not represent a range but one data point. Therefore, we have updated the Table 2 (now Table 1) to clarify the values so that it is easier to see. We have also updated the graph such that legend clearly shows what ET is presented in the graph.

Page 23 Line 9: Missing word after 'relatively'

Thanks, this has been reworded. Pg34 Line 2-3

Line 15: Gorissen reference should go after the word 'ecosystem'. Insert 'this' before 'ecological' Reworded and changed as per reviewer's comments.

Page 32-33 Table 3: should be Table 1. It's difficult to determine which numbers pertain to which parameter. Either move the parameters or put borders around columns and rows. Move column 3 down do that the first 'peat' is in lone with the first bulk density number

Thanks, Table 1 has been amended to make it easier to determine which numbers belong to which property. Table 1 has been moved to Supplement

Page 33: Table 4 should be table 2

Thanks, this is amended now.

References:

References cited in text that are not in reference list:

Huneau et al, 2003, Hendry et al 2013, Hendry, 2008, Hunt et al, 1996, Mandl et al, 2017, Bickford and Gell, 2005, Middleton and Kleinebecker, 2012, Johnson, 2006, Valentin et al, 2005, Gatt, 1996, Dansgaard, 1964, Linacre et al, 1967, Mathieu and Bariac, 1996, dePaolo et al 2012.

All references missing from the list, or where incorrect year was attributed, have now been added to the list of references and checked in the manuscript.

References in reference list not cited in text:

Andersen, M., Barron, O., Bond, N., Burrows, R., Eberhard, S., Emelyanova, I., Fensham,

R., Froend, R., Kennard, M., Marsh, N., Pettit, N., Rossini, R., Rutlidge, R.,

Valdez, D. and Ward, D.: Research to inform the assessment of ecohydrological 10 responses to coal seam gas extraction and coal mining. Department of the Environment

and Energy, Commonwealth of Australia, 2016.

Bukata, B.J., Osborne, T.Z. and Szafraniec, M.I.: Soil nutrient assessment and characterisation in a degraded Cnetral Florida 30 Swamp. Water Air Soil Pollut, 226: 307, 2015.

Centennial Coal.: Springvale Mine Extension project,

Environmental Impact Statement, Available online

http://majorprojects.planning.nsw.gov.au/index.pl?action=view\_job&job\_id=5594 Accessed on 20 April 2017, 2014.

Cloern, J. E., Canuel, E. A. and Harris, D.: Stable carbon and nitrogen isotope composition of aquatic and terrestrial plants of the San Francisco Bay estuarine system.

Limnology and Oceanography, 47(3): 713-729, 2002.

Denk, T. R. A., Mohn, J., Decock, C., Lewicka-Szczebak, D., Harris, E., Butterbach-Bahl, K., Kiese, R. and Wolf, B. D.: The nitrogen cycle: A review of isotope effects and isotope modeling approaches. Soil Biology and Biochemistry 105: 121-137, 2017. Deegan, L. A. and Garritt, R.H.: Evidence for spatial variability in estuarine food webs.

Marine Ecology Progress Series 147: 31-47, 1997.

Fry, B.: Stable isotope ecology, Springer, LA, pp 317, 2006.

Gardner, W.H.: Water content. In: (ed. A. Klute) Methods of Soil Analysis, Part 1. Bibliography 165 Physical and Mineralogical Methods. Agronomy Monograph No.9 (2nd edn). pp493- 544, 1986.

Gorissen, S., Mallinson, J.Greenlees, M. and Shine, R.: The impact of fire regimes on populations of an endangered lizard in montane south-esastern Australia. Austral. Ecol. 40, 170-177, 2015.

Heaton, T.H.E.: Isotopic studies of nitrogen pollution in the hydrosphere and atmosphere: a review. Chem. Geol. 59, 87-102, 20 1986.

Huneau, F, Blavoux, B, Aeschbach–Hertig, W. and Kipfer, R.: Paleogroundwater of the Valreas Miocene aquifer (Southeastern France) as archives of the lgm/Holocene transition in the western Mediterranean region, IAEA report, IAEA-CN-80/24, 2013. Johnson, D.: Sacred waters: the story of the Blue Mountains gully traditional owners. Broadway, N.S.W.: Halstead Press, 237 pp, 2007.

Liu, Y., Sheng, L. and Liu, J.P.: Impact of wetland change on local climate in semi-arid zone of Northeast China. Chinese Geographical Science. 25,309-320, 2015.

Mandl, M. B., Shuman, B. N., Marsicek, J., Grigg, L.: Estimating the regional climate signal in a late Pleistocene and early Holocene lake-sediment \_180 record from Vermont, USA. Quaternary Research (United States) 86(1): 67-78, 2016.

Potter, N.J., Chiew, F.H.S., Frost, A.J., Srikanthan, R., McMahon, T.A., Peel, M.C. and Austin, J.M.: Characterisation of 20 recent rainfall and runoff in the Murray-Darling Basin. A report to the Australian Government from the CSIRO Murray-Darling Basin Sustainable Yields Project. CSIRO, Australia. 40pp, 2008.

Rau G.H.: Carbon-13/carbon-12 variation in subalpine lake aquatic insects: Food source implications. Can. J. Fish. Aquat. Sci 37: 742-746, 1980.

Reddy, K.R. and DeLaune, R.L.: Biogeochemistry of wetlands science and applications.

Boca Raton, FL: CRC Press, pp774, 30 2008.

Zhang, X., Sigman, D.M., Morel, F.M.M>, Kraepiel, A.M.L.: Nitrogen isotope fractionation by alternative nitrogases and past ocean anoxia. Proceedings of the National Academy of Sciences of the United States of America 111, 4782-4787, 2014.

References which were not mentioned in the text were excluded from the reference list.

**Dear Reviewer2,**

Thanks for the constructive review and comments provided for the manuscript. We have included a detailed response to the questions below in responses.

**General Reviewer's Comment**

While I personally support the motivation of this study and also the propagation of isotope methods I regret to say that I have several issues regarding this manuscript:

Title:

**Reviewer's Comment**

The title does not fully reflect the work described here. Not only isotope data have been collected and used and the manuscript's main conclusion is based on other data.

**Response**

We acknowledge the title does not fully describe the work completed so we propose the changed title as follows: "Application of pore water stable isotope method and hydrogeological approaches to characterise a wetland system", Pg Line 1-2

**Reviewer's Comment**

Main outcome: The most important, yet qualitative finding of this study is groundwater flow through the wetland system. The presence of high groundwater levels despite high evaporation rate estimates led to the indirect conclusion that lateral groundwater inflow must be effective. This could already be expected, given just the combination of slope (5%), groundwater levels (high) and permeability of bedrock (sandstone –> high). The vulnerability of such systems following e.g. mining activity is based on potential changes of groundwater flow. The main effort of this study, however, was the estimation of evaporation rates which will most likely not be altered by e.g. mining activity. Furthermore, the application of the direct vapor equilibration method was never methodically restricted to non-wetlands and therefore does not constitute a challenge itself that needs to be emphasized.

**Response**

We agree with the reviewer that the effectiveness of groundwater lateral flow in the swamp system could be deduced from a combination of topography and geology drivers. However, the relative effectiveness of groundwater flow in the swamp water balance, has yet to be thoroughly evaluated. Given that this study did not use the piezometer data but mainly relied on stable isotopes, we have provided a quantitative estimate which represents a conservative estimate of a minimum volume of groundwater contribution to the wetland. Evaporation in this research was one of the parameters of the water balance, used to estimate groundwater contribution. Mining is expected to change the groundwater flow paths, chemistry and ground surface elevation and therefore evaporation. Pg 3 Line 3-4 and Pg 2 Lines 3-7

It is also true, that the direct equilibration method was not restricted to non-wetlands, but the method reported in literature to date has been applied in low permeability unconsolidated or consolidated environments. However, it has never been applied in the relatively high permeability swamp system. We therefore believe that there is a merit in showing the value of applying the method in a swamp environment.

**Model selection:**

**Reviewer's Comment**

The selected model was developed and applied in desert regions with only vertical water flow, no vegetation, and no lateral groundwater contribution to the zone under investigation. I therefore doubt that invariance of isotopologue diffusivities is an exhaustive criterion for model selection.

Furthermore, the assumption of steady state conditions in a region with pronounced wet and dry season seems to be very far-fetched. The authors should either provide more details why the selected model is still applicable in a vegetated wetland environment with vertical and lateral water flow components or they should use a different approach that better considers subsurface water flow velocities and directions under both dry and wet conditions.

**Response**

Barnes and Allison (1988) model has been applied in experimental and field studies to both arid and non-arid regions. The assumption of steady state characteristics was considered appropriate as the analysis was mainly focused on the outcomes of the dry weather season. However, a wet period was also modelled to see if the model could match the data during this period, not during the transition from dry to wet which would clearly invalidate model assumptions. Stable diffusivities do not represent a complete or only criterion for model selection, in fact vapour diffusion has an impact on effective diffusivity such that it results in its increase therefore they are not stable. The effective diffusivities are the function of water content and they are proportional to the effect of porous medium. Given that the water content decreases as evaporation proceeds the tortuosity will also decrease. The factor that influences diffusivity is in effect a product of tortuosity and volumetric water content.

The selected steady unsaturated conditions model by Barnes and Allison, 1988 has not been restricted to arid regions, in fact, the model has been applied to clay mineral unsaturated experimental laboratory environment and vegetated and non-vegetated impacts. The model relies on Zimmermann (1967) observations of the effects of vegetation on the isotopic profiles and Barnes and Allison (1988) conclude that under vegetation the enrichment will depend on the effect of deep profile due to lower evaporation rates, lower water contents and effect of transpiration.

Further support for the selection of the Barnes and Allison (1988) model for the vegetated wetland environment is also shown in the recent work undertaken by Piayda et al (2017). They studied vegetation effects on soil water infiltration and distribution as well as dynamics of soil evaporation and grassland water use in a Mediterranean cork oak woodland. They found that regardless of the presence of vegetation or bare soil, the total evapotranspirative water loss of soil and understorey remains unchanged, but infiltration rates decreased by 24%. In their study, below-ground biomass was sampled with soil cores in 5, 15, 30 and 60 cm depth. In total, 80% of root biomass was found to be distributed between 5 and 15 cm depth. Only 5% was distributed above 5 cm and 15% between 20 and 35 cm depth.

Reference: Piayda A., Dubbert M., Siegwolf R., Cuntz M., Werner C., 2017. Quantification of dynamic soil-vegetation feedbacks following an istopically labelled precipitation pulse. Biogeosciences, 14, 2293-2306.

Furthermore, the modelling is considered to be applicable by focusing on vertical flow. We only included the samples from the base of the swamp (not sides) where vertical flow is dominant due to high permeability of the peat. We have not modelled the regional aquifer water balance in this study.

**Text changed at Pg 23 Lines 15-19**

**Data collection:**

**Reviewer's Comment:**

The description of the calibration and validation routine of water stable isotope data is quite confusing. Why did the authors not use the linear regression between known values of liquid

**standards and raw readings of the respective headspace vapors to calibrate the unknown sample values?**

Response

The description of the calibration of water stable isotope data is provided in Pg 10 Line 8-11. The calibration was undertaken as follows: a linear regression relationship was established between the known liquid standards i.e. two LGR standards and VSMOW/VSMOW2 and corrected vapours of the same standards. The standards' vapours were expressed as water using the fractionation factor at 25C. The regression equation was then applied to raw readings of the vapours for samples. This was used to calibrate and normalise the samples. This detail was not clearly explained in the manuscript and has now been added. The description of the calibration and validation routine of pore water stable isotope data has been significantly expanded and is given in Pg 9 Line 17 to Pg 10 Line 8.

**Reviewer's Comment What did they correct the readings for? Response The readings were also corrected for instrument drift. Pg 9 Line 30-35 to Pg 10 Line 1-2**

Reviewer's Comment Or did they mean "calibrate" when they wrote "correct"? Response: It is acknowledged that terminology in the manuscript needed to be more consistent. Standards were used to calibrate the samples as per the above response and corrected for instrument drift. Pg 9 Line 15 to Pg 10 Line 2

Reviewer's Comment Why did they (have to) calculate individual fractionation factors? This would not have been necessary if all standards and samples had been stored sufficiently long and analyzed subsequently in a temperature-controlled environment following the principle of identical treatment.

Response: the fractionation factor was not calculated individually, it was only applied to measured standard's vapours to express as water. This was used for regression with liquid standards. The regression was then applied to raw sample readings. Pg 9 Line 18. All samples and standards have been stored at 4°C prior to the analysis, and all have been allowed the same time on the lab bench in the temperature controlled laboratory and have followed the same treatment. Pg 8 Lines 24 to Pg 9 Line 4

**Reviewer's Comment Also, reported uncertainties as measures of data quality are meaningless if they are not based on the authors' applied standard operation procedures.**

Response: Standard operational procedures were followed, all samples were prepared in the same manner, at the same temperature and at the same time, stored at the same temperature, and were all allowed the same time prior to being prepared for analysis. The laboratory was temperature controlled. Pg 8 Lines 2 to Pg 9 Line 4

Specific comments and Technical corrections

Reviewer's Comment: P2 L7-8: "aiming at" instead of "enabling". The ability to quantify components of the water balance depends on environmental conditions, not on the method of data collection. Response Thanks, this is corrected in the manuscript. Pg 1 Line 19

Reviewer's Comment P2 L8-10: insert "potentially" before "enables". Otherwise this statement is too strong. The technique itself only enables collection of isotope data. Understanding of processes is a different thing.

**Response Thanks, the word has been inserted as suggested. Pg 1 Line 21**

Reviewer's Comment P2 L11 & elsewhere: the numbers following the delta symbol have to be in superscript

Response Thanks, all numbers following delta symbol are in superscript in the manuscript, the HESS conversion to pdf caused this change and converted them to normal numbers

**Reviewer's Comment P2 L12: is the porewater compression technique the most widely applied and accepted benchmark?**

Response This has been reworded to say, "with other reported physical and chemical techniques". Also, current literature provides good comparison and results comparable or better to those of other methods (Hendry et al, 2015) Pg 1 Line 24

**Reviewer's Comment P2 L19-21: the finding of sand underlying the swamp can't really be credited to this study as it had been described before in studies cited by the authors.**

Response Other cited literature describes swamps in the Blue Mountains and did not specifically relate to this area. The sand underlaying the swamps will influence the response of the swamp to disturbances such as mining, in contrast to lower permeability clay. The geology of the Newnes Plateau area where these swamps are located, comprises Burralow formation which is known to have thin interlayered siltstone and mudstone layers. Other studies (referenced in the manuscript) were undertaken in the area which is underlain by Banks Wall Sandstone, which is cleaner quartz sandstone. Pg 1 Line 35

Reviewer's Comment P3 L9: multi, not muliti; sedimentary, not sedimentatry

**Response Thanks, this is now corrected in the manuscript.**

Reviewer's Comment P3 L11 and throughout MS: the references cited in the MS do not match the ones listed in the references section. I was therefore often unable to find the referenced statements in the cited literature

**Response Thanks, this has been checked and corrected.**

**Reviewer's Comment P3 L12, 16 & 19: why do the authors distinguish between Australian and international literature? Is one more relevant or trustworthy than the other?**

Response The reason for distinguishing between Australian and international literature was due to climate in which these swamps were formed. In the northern hemisphere where most wetland work is reported, the swamps were formed since the last glaciation in Holocene, while in Australia they were formed in the non-glaciated period. A study by Fryers et al (2014) found that highly variable nature of hydrological regime in Australia these swamp systems have different formation and evolution compared to those in the northern hemisphere climates. The information referring to international literature has been omitted and paragraph changed. Pg 2, line 23-Pg 3 Line 4

**Reviewer's Comment P3 L17: insert "other" before "environmental tracers"**

Response Thanks the word has been inserted.

**Reviewer's Comment P3 L18: please specify the processes you are referring to**

Response This refers to hydraulic connectivity processes. This sentence was removed from manuscript as it was not directly related to the topic.

Reviewer's Comment P3 L26: all THPSS terms should start with capital letters. Or none.

Response All THPSS terms are now capitalised

Reviewer's Comment P3 L31 & elsewhere: citation style: put years in braces

Response Thanks, this is corrected throughout the text.

Reviewer's Comment P4 L3: "swamp behavior" is too sloppy, please specify

Response Changed to "hydrogeological changes in the swamps"

Reviewer's Comment P4 L18: given that especially rainfall and surface water can vary on very short timescales with unknown time lags relative to soil water which in turn is subject to dispersion, how can these be considered distinct endmember points?

Response Rainfall, and as a result surface water, do vary based on the rainfall intensity, source of rainfall (Eastern Coast Low or Western Through) and season, however they are considered the endpoints given their role in the input and output part of the water cycle. Also, these natural variations have been very well documented by rainfall sample collection, which has allowed us to compare detailed variations of rainfall and groundwater in this manuscript. The changes were made in the manuscript Pg 4 Line 4

Reviewer's Comment P5 L5: a verb is missing after "and"

Response Reworded as follows "and they occur at highest elevation"

Reviewer's Comment P6 L2-4: if this does not affect the site under investigation, then why mention?

Response The plantation was mentioned, as it is immediately next to the catchment boundary of the swamps in this study. However, other swamps not in this study can be affected by changes in the plantation. This sentence has been removed from manuscript.

Reviewer's Comment P6 L7: "fed by groundwater discharge" is a contradiction. I suggest to use the expression "fed by lateral groundwater inflow"

Response Thanks, changed as suggested. Pg 5 Line 21

Reviewer's Comment P6 L16-19: please provide numbers rather than "below average" or "above average "

Response Numbers added as suggested:" for February to April (46.6mm, 36.8 mm, 6.6 mm and 20.8 mm for each of the months)" and " above average rainfall from June to September (170.2 mm, 102 mm, 61.8 mm and 92 mm for each of the months" Pg 6 Line 19-23

Reviewer's Comment P7 L5: can you please provide here already the number of samples as well as the achieved spatial depth resolution of sampling?

Response This has been added to the manuscript as follows: A total of 34 pore water samples and 5 surface and groundwater samples were collected during May 2016. During October 2016 sampling event 14 pore water samples and 13 surface and groundwater samples were collected. A total of 27 pore water samples and 6 surface and groundwater samples were collected in May 2017. The spatial depth resolution varied from 10 to 20 cm depending on the penetration of the corer. Pg 6 Line 20 – 26.

**Reviewer's Comment P7 L7: why had sampling holes to be restored or why is this important to be mentioned?**

Response Because the field work was undertaken in a sensitive environmental area protected by State and Federal legislation and is part of the World Heritage Area, therefore the sampling holes could not be left open to modify conditions in the swamp, but were filled after sampling.

**Reviewer's Comment P7 L7: excess of what? Why was not all soil material sampled?**

Response Thanks, word "excess" replaced by "soil"

**Reviewer's Comment P7 L13: please specify what you mean by "nature"**

Response This includes colour, shape of the grains and size of grains. This has been explained in the manuscript now. Pg 7 Line 7-8

**Reviewer's Comment P7 L13: "were described" instead of were described"**

Response Thanks, the words are now separated.

**Reviewer's Comment P7 L16: why were samples vacuum packed? How and how long were they stored prior to analysis?**

Response Samples were vacuum packed, to avoid equilibration with the air in the bag. The samples were stored for three days in the cool environment at 4°C temperature. Pg 7 Line 25

**Reviewer's Comment P8 L1: insert "data" before "for precipitation"**

**Response Thanks the word is now inserted**

Reviewer's Comment P8 L9: 17-24 hours is too short for reaching complete isotopic equilibrium between soil water and headspace vapor. It should be several days, up to one week for clayey samples (Wassenaar et al, 2008). Or do the authors have indication that complete isotopic equilibrium between all relevant fractions of soil water was reached? If so please describe

Response The sample equilibration time is dependent whether the core is intact or broken into small pieces, and if it is unconsolidated. For each geologic material the equilibration time needs to be established experimentally. The core samples in Wassenaar et al (2008) study, were intact and consolidated, low permeability representing the worst-case scenario in terms of time required, therefore the time when samples were fully equilibrated was after 3 days. Similar findings were reported in David et al (2015) where testing was done on consolidated, low permeability sandstone and siltstone cores. However, in this research the geologic medium is the high permeability, loose, broken down and unconsolidated peat, organic material, sands and silt. As a result of this the

equilibration time was reduced to up to 24 hours. After this time the full equilibration was achieved which is evident in headspace water content of between 25000 to 29000 ppm, as reported in (Wassenaar et al, 2008; David et al, 2015). Pg 8 Line 28 to Pg 9 Line 6

Reviewer's Comment P8 L10: I suppose this concentration range yields minimum data noise on a LGR instrument? Such a large range of vapor concentrations, however, probably indicates non-isothermal storage prior to and during analysis. Can the authors please comment on pre-analysis storage conditions with respect to the principle of identical treatment of samples and standards?

Response The concentration range is required for the analysis, i.e. a minimum of 5% of moisture content per sample. Below this range the results are not accurate. The variation is due to different lithological characteristics of samples, different moisture content and hydraulic conductivity. Pg 8 Line 27 to Pg 9 Line 5

Before the analysis all samples and standards were stored in the cool place at 4°C and were then transferred to the lab bench when preparing for the analysis. The temperature in the lab is controlled. Pg 8 Line 24-29 and Pg 10 Line 1-2

Reviewer's Comment P8 L13: what kind of plastic tube? Was it diffusion-tight material? Else, the authors might have sampled a mixture of sample headspace and an unknown fraction of ambient vapor.

Response It is a flexible, thick walled plastic tube which is connected with fittings to LGR inlet on one side and with needle on the other side. The contamination by atmospheric air is considered negligible. Pg 9 Line 8 This is based on the measurement of ambient air moisture of around 14,000 to 15,000 ppm, while the headspace for samples had a range of 23,000 to 28,000 ppm H2O. Also, given high moisture content in the sample/bag the contamination is negligible based on mass balance. Pg 9 Line 10-13

Reviewer's Comment P8 L15: 20 minutes appear way too short. See also comment to P8 L9 Response The standards are pure liquid and therefore require little time for equilibration. Wassenaar et al (2008) report 5 minutes for liquid samples, David et al (2015) report 20 minutes. Pg 9, Line 14-15

Reviewer's Comment P8 L15 & elsewhere: the expression "\_18O/\_2H" is conceptually wrong and misleading given that this could be interpreted mathematically as a ratio of two isotope ratios Response Thanks, corrected to  $\delta^{18}$ O and  $\delta^{2}$ H throughout.

Reviewer's Comment P8 L18: what was the volume of the standards? Did you prepare replicates or were the standards' headspaces analyzed repeatedly?

Response The volume of standards was 1 ml (about 1-4 drops) and they were prepared as replicates after each set of 3 samples. It is not possible to sample the headspace repeatedly, as 1 L headspace only allows sampling once. Inflating with dry air again results in incorrect readings. Pg 9 Line 15-19

Reviewer's Comment P8 L20-21: does this refer to liquid analyses or the direct vapor equilibration method?

Response This relates to direct vapour equilibration method. This has been added to the manuscript to clarify. Pg 9, Line 30

Reviewer's Comment P8 L21-22: how do these timespans compare to sampling time of the individual sample headspaces? If this is the precision reported by the manufacturer I would prefer to read about the precision the authors achieved when following their standard operation

**procedure (also in P9 L2)**

Response The sampling time of the individual headspace was from 35 to 60 seconds. After each full set of samples and standards the analysis was suspended for about 10-15 minutes to allow the LGR instrument to reach the stable readings of the ambient air. Pg 9 Line 16-17 The instrument precision is reported in L21-22, but the reproducibility of the results resulting from this study is reported in L20-L21. Each dataset was corrected for the instrument drift, and when following the standard operating procedures, the precision in this research was 0.63‰  $\delta^2$ H and 0.23%  $\delta^{18}$ O over 70 seconds. Pg 9 Line 30 to Pg 10 Line 2

**Reviewer's Comment P8 L26: LGR's technology is called e.g. OA-ICOS, but not IRMS**

Response Corrected typo to Off axis -integrated(OC-ICOS) cavity output spectrometry. The LGR used for liquid samples was not the same instrument. Pg 10 line 10-13

**Reviewer's Comment P8 L30: please provide a reference for the LOI method**

**Response The reference for LOI is provided Heiri et al (2001) Pg 10 Line 13**

**Reviewer's Comment P9 L5: this contradicts MS title which prominently mentions isotope data**

Response The authors consider it important to have a section in the manuscript which defines the stratigraphy, organic matter and moisture content as this helps understand and support the isotope data. This was also requested by Reviewer 1 of this manuscript.

**Reviewer's Comment P10 L4: what is "sub-angular" quartz?**

Response Term "Sub-angular" defines the roundness of quartz or any other sediment grain. This is important as it can point to information on transport of the material.

**Reviewer's Comment P10 L5: please describe how grain size distributions were determined. If this information was taken from other publications, it should appear in section 2 rather than the result section.**

Response The description of the lithology in L5 and grain size was determined by logging the samples as soon as they were extracted. This included defining the lithology, grain size and colour. The logging was undertaken for this research on the core samples that were extracted and is not taken from other publications. P

A sentence was added to the methods section: The samples were geologically logged after extraction, by noting the lithology, grain size and roundness, matrix and colour. Additional sentence was added to results section: The cross -sections presented in Figures 3 to 5 were prepared on the basis of logged cores extracted as part of this research. g 6 Line 7 -8

**Reviewer's Comment P10 L9: "dark" is not a color Response Thanks, this has been clarified as follows: "dark grey"**

**Reviewer's Comment P10 L10: can the authors please comment on the "organic smell". Volatile organic compounds likely have a strong effect on laser-based isotope analysis that needs to be considered for such samples.**

Response The "organic smell' relates here to peat sediments, pointing out to the high presence of organic material. This is not related to volatile organic compounds, which are not present in these young (Holocene) sedimentary environments. This has been clarified in the manuscript. Pg 12 Line 6

Reviewer's Comment P11 L5: "can be explained" seems to be not only descriptive and should appear in the discussion rather than the result section

Response Thanks, this was moved to Section 5.1

**Reviewer's Comment P12 L1: please insert some specification after "of"**

Response The sentence has been reworded to read: The relationship between surface water, swamp groundwater, regional groundwater and swamp pore water the  $\delta^{18}$ O and  $\delta^{2}$ H data. Pg 15 Line 6-7

Reviewer's Comment P12 L3 & 8: the intercepts in the linear equations should have a the "unit" ‰ Response Thanks, the unit was added.

Reviewer's Comment P12 L9: it appears to me that the slope is slightly higher, but the intercept is similar

Response We agree with the reviewer's comment for May 2017, the sentence was added to the manuscript: The exception is May 2017 when the slope of LMWL is slightly higher, but the intercept is similar. This sentence was removed as it did not add value to the manuscript.

Reviewer's Comment P13 L9: "likely to be": see comment on P11 L5 Response Thanks, this sentence was moved to Section 5.2.

Reviewer's Comment P14 L3 & throughout MS + supplement: stable isotope data should be consistently reported with two decimal places for \_18O data and one for \_2H data Response Thanks, this is corrected now throughout the document.

Reviewer's Comment P14 L5 "it seems unlikely": see comment on P11 L5 Response This sentence was moved to Section 5.2.

Reviewer's Comment P14 L10: this seems to be a misinterpretation potentially affecting overall results. This regression line is not to be mixed up with evaporation lines (See Benettin et al, 2018, for details). Further, it should not be interpreted in the results section.

Response Thanks, the sentence was removed from the results section. We agree that the regression line should not be mixed up with evaporation line and as per Benettin et al (2018). The sentence that was removed tried to explain the slope of the regression line, and differentiate it from the wet weather line. The text explains the importance of small and big rainfall events on the pore water response. The presentation of the rainfall isotopic signature and absence of direct correlation with swamp sediment isotopic pore water data, confirms the ideas discussed in Benettin et al (2018) paper. It is not considered that the sentence in question has bearing on the overall results, as matching of observed data to the model provides a clear indication of evaporation in the upper horizon for May 2016 dry weather sampling period. Pg 31 Line 2-4

**Reviewer's Comment P14 L14: Since the authors present a LMWL, I would suggest to report lc-excess values (Landwehr & Coplen, 2006) rather than D-excess values**

Response We acknowledge that reporting lc-excess is a good concept, however in this research we consider that D-excess provides a better indication. The reason for this is because the slope of the LMWL is close to 8 similar to that of the GMWL. Another reason for reporting D-excess is to allow other researchers to use the results of this study by comparing to well-defined GMWL.

**Reviewer's Comment P14 L15-20: this paragraph appears twice**

Response Thanks, this is now corrected.

Reviewer's Comment P14 L23: "we would expect": see comment on P11 L5

**Response Thanks, this sentence was moved to Section 5.2**

Reviewer's Comment P14 L25: "seems to suggest": see comment on P11 L5 Response Thanks, this sentence was moved to Section 5.2

Reviewer's Comment P14 L29: "may be": see comment on P11 L5 Response Thanks, this sentence was moved to Section 5.2

Reviewer's Comment P15 L8: "assumed to be": see comment on P11 L5 Response Thanks, this sentence was moved to Section 5.2

Reviewer's Comment P16 L1 & 7: insert "values" or equivalent before "of pore" Response Thanks, word 'values' has been inserted

Reviewer's Comment P16 L8-9: "due to": see comment on P11 L5 Response Thanks, this was reworded as follows: "however slightly more depleted and similar to significant rainfall in March 2017" Pg22 Line 18-19

Reviewer's Comment P16 L11: is the number in braces a \_\_\_\_-value? Response Thanks, word "values" added

Reviewer's Comment P16 L11: "is a reflection": see comment on P11 L5 Response Thanks, this was reworded to: "and is similar to typical winter rainfall signature"

Reviewer's Comment P16 L13: isn't this just a water balance rather than a water and mass balance? Response True, corrected

Reviewer's Comment P16 L19: "very" = "vary"? Response Thanks, changed to "vary"

Reviewer's Comment P16 L26-27: the description how data were collected should be provided in the method section Response Thanks, this was moved to Section 3.2.

Reviewer's Comment P17 L8 & 9: "is related to": see comment on P11 L5 Response Thanks, sentence was moved to Section 5.1.

Reviewer's Comment P18 L12: "groundwater recharge" is misleading as it describes the replenishment of groundwater. I suggest to use "lateral groundwater inflow" instead. Response This has been updated as suggested.

Reviewer's Comment P18 L11-18: the information on e.g. slope, vegetation, age dating should be provided in the section describing the study site (section 2) Response The information as suggested by the reviewer has been moved to Section 2.

Reviewer's Comment P18 L22: why not report salinity data? Do they support the surprisingly high evaporation estimates?

Response The salinity data has been referenced here to another report (David et al, 2018), but please note the evaporation estimates are relatively high. This means that they are high for upland swamp and this specific environment but not high compared to evaporation at lower elevations. As a result, the evaporation does not cause an increase in salinity as expected in arid environments.

Also, as mentioned, the regional groundwater contribution of fresh water also means that salinity is not an issue in this area. Pg 26 Line 29-30

Reviewer's Comment P18 L26: insert "values" or equivalent before "of pore"

Response Thanks, the word 'values' has been added.

Reviewer's Comment P18 L27: discharge: see comment on P6 L7

Response Thanks, changed to "inflow"

Reviewer's Comment P18 L30: I would expect that upward flow supporting high evaporation rates would result in an increase of EC in surface water.

Response Upward flow is from the regional groundwater in sandstone, this water Is fresh and depth to water (4.5 m bgl) below evaporation influence. Therefore, no increase in EC is likely, in particular due to estimated groundwater contribution in the dry weather period.

Reviewer's Comment P20 L15-16: evaporation always occurs at the liquid-vapor interface (the "surface"). How can evaporation ("to at least 60cm") occur below the water table (28-38cm)? Response The sentence was not aiming to say that evaporation proceeds below the water table, but that if evaporation only was occurring, without regional groundwater inflow, then more water would be evaporated from the surface and the depth to water would be greater. This has been reworded. Pg29 Line 32-35

Reviewer's Comment P20 L19: "measured"? This number was rather calculated than measured Response Thanks, this has been updated as per suggestion.

Reviewer's Comment P20 L25: this figure should be presented and described in the results section Response This figure, and Table 2 (now Table 1) have both been moved to results Section on Water balance.

Reviewer's Comment P21 L6: "clearly defined endmembers": see comment on P4 L18 Response The text has been reworded to say "endmembers". As mentioned in the earlier response, although the rainfall signature changes, we can track this signature as there is a good dataset of stable isotope data for precipitation. A sentence was added as follows: Although the stable isotope data in precipitation changes in the short term, this end member is well constrained based on the good quality dataset for precipitation. PG 31, Line 7-8

Reviewer's Comment P22 L1: difference in what? Response Thanks, text added " in  $\delta^{18}$ O and  $\delta^{2}$ H values"

Reviewer's Comment P22 L9: "of precipitation" after "signature" Response Thanks, the suggestion was added

Reviewer's Comment P22 L27-28: This conclusion has been drawn before (Wassenaar et al, 2008) and can't be credited to the present study Response Thanks, reference added.

Reviewer's Comment P22 L29-32: this statement is not conclusive and rather belongs to the introduction Response Text has been moved to introduction Supplement: how come that foil weights differ by more than factor 4? Did the authors not use standardized sample bags (e.g. Ziploc) with comparable weights? If not this would be a violation of the principle of identical treatment.

Response The samples for gravimetric water content determination were dried in the oven, therefore they were placed in the aluminium foil cups. These cups, were not the same in weight for each sample. However, the standards under which this analysis was done (ASTM D2974-14, 2014 and ASTM D2216-10, 2010), allows for some change in weight. As a result, the equation to calculate gravimetric water content requires weighing of foil.

**Dear Reviewer3,**

Thanks for the review and comments that have improved our manuscript. We have included a detailed response to the questions below in responses.

**General Reviewer's Comment**

This review follows the assessment of Anonymous Reviewers 1 and 2 – this reviewer concurs with their suggestions and adds only the following comments. I must reject this paper due to serious concerns about the accuracy of porewater isotope analytical methods. If there is no clear confidence in the analytical isotopic results the subsequent modeling does not matter.

**Response**

Thanks, it is acknowledged that the description of porewater isotope analytical methods was relatively brief. This section has now been improved and expanded to include additional details on the analytical method sampling technique. Section 3 provides more information on accuracy and confidence. This section has also has been updated based on comments from Reviewer 1 and 2.

**General Reviewer's Comment**

This paper, if revised and resubmitted, further requires hard editing and a lot of trimming. Please conduct a thorough review for basic grammar and sentence structure. Check for imprecise or vague terminology usage. Please consider reducing the length – in many places there is unnecessary "filler text" (i.e. "International publications/ example : : : why not just say "research has shown: : : (refs). Remove a lot of the ancillary information (detailed lithology) that is not explicitly needed for your objectives of using pore water isotopes.

**Response**

Thanks for the comments, further editing has been undertaken on this paper, and grammar and sentence structure checked.

However, we respectfully disagree with the suggestion to trim the text. Our response is justified by Reviewer 1 who requested more ancillary information (detailed lithology), who for example requested that an additional cross-section be prepared to help with understanding the lithology. We have therefore added more information and additional cross-section in response to Reviewer 1 comments.

We have reduced the length and removed the text which does not related to wetlands in the Introduction section. Please note some of the text is necessary to understand the background, and follows the response to Review2. For example, the "international publications" is not a "filler text", the difference between the international and local literature is important as the swamp systems in Australia have formed under very different climate conditions compared to other swamps in the Northern hemisphere as discussed in Fryirs et al. (2014). This has resulted in different formation and evolution model for the upland swamps on sandstone in Eastern Australia. For all other items of discussion in the paper, the reviewer 3 comments were used to correct the text. The text in the manuscript has been reduced and edited as recommended by Reviewer.

**General Reviewer's Comment - Method's section**

The entire Methods section, upon which this work hinges entirely, is insufficiently described or referenced. For example, page 7 lines 13-17 – no citations are give for this sampling methodology.

**Response**

Thanks, this is acknowledged, and the entire Methods section and Sections 3.1 and 3.2 have been considerably expanded and rewritten based on Reviewer's suggestion and suggestions from Reviewers 1 and 2 (please see below).

For example, the citation is now provided for the sampling methodology (Wassenaar et al (2008) and Hendry et al (2015)).

**3.1 Fieldwork and sampling**

Fieldwork was undertaken during 2016 and 2017 with the swamps in a natural state and recovered from earlier wildfire in 2013. The first sampling event 24th to 25th May 2016 occurred following an extremely dry weather period of four months below the long-term average rainfall (BoM Lithgow Station SN63226, 900 m AHD, 13 km SW of the study area with 139 years of data records) for February to May (46.6 mm, 36.8 mm, 6.6 mm and 20.8 mm for each of the months). A total of 34 pore water samples and 5 surface and groundwater samples were collected. A repeat sampling on 25th to 26th October 2016 occurred after four months of above average rainfall from June to September (170.2 mm, 102 mm, 61.8 mm and 92 mm for each of the months). During October 2016 sampling event 14 pore water samples and 13 surface and groundwater samples were collected. Sampling on 30th May 2017 occurred under different climate conditions with both above and below average rainfall trend in the months preceding the sampling event. A total of 27 pore water samples, 3 surface and 3 groundwater samples were collected in May 2017. The spatial depth resolution varied from 10 to 20 cm depending on the penetration of the corer. **Figure 2** shows the variation in monthly long-term rainfall (139 years) and comparison with rainfall in 2016 and 2017.

Figure 2: Long term average monthly rainfall at Lithgow station (Bureau of Meteorology Station No. 063226) compared to rainfall during 2016 and 2017.

In total seven sediment cores were obtained by coring using a Russian D hand corer (40 mm diameter) to rock refusal (between 0.45 to 1.4 m), and three transects (CC, GG and GGSW) were prepared along the length of the swamps (**Fig 3,4 and 5**). Samples were geologically logged after extraction, by noting the lithology, grain size and roundness, matrix and colour. The hand cored holes were restored by returning soil material to the hole immediately after sampling. This was undertaken to ensure no change occurred to endangered and protected ecological system as a result of sampling. The coring on CC transect was repeated in October 2016 at a distance of less than 0.5 m from the original hole. The coring locations were selected to represent swamp stratigraphy from upstream to downstream and to provide a spatial coverage across the three swamps. In addition, three cored locations were selected such that they were adjacent to an existing piezometer (CCG1 on transect CC and GGEG2A and GGEG4 on GG transect). The purpose of this, in addition to determining the stable isotope profiles, was to enable comparison with the swamp groundwater measurements and to collect regional groundwater from the underlying sandstone aquifer where possible.

Sediment cores were divided into subsamples of 10-20 cm length, packed into Ziploc bags and kept in cool storage for later analysis of moisture content and organic matter content. The samples for pore water analysis were temporarily double packed in Ziploc bags by minimising the airspace in the bag, stored in the cooled ice box in accordance with the sampling protocol developed by Wassenaar et al (2008) and further improved by Hendry et al (2015). The same afternoon after collection, samples were packed in tough high-grade food storage plastic bags with air extracted, double sealed, separately stored in an additional plastic bag and were kept at a 4°C to prevent evaporation. Vacuum

packing was required to minimise atmospheric moisture contamination. All isotopic field controls during sampling and analysis were implemented; this included: quick storage in tough plastic bags, immediate double bagging during collection and vacuum packing the same afternoon. Storage time for samples after collection was 3 days in the cool environment (4°C) before they were analysed.

Swamp groundwater was sampled directly from the cored hole and field parameters were measured immediately (pH, EC, DO, temperature). This was repeated for all three sampling events; however, some bores were dry and some not accessible. Swamp groundwater and regional groundwater from existing piezometers (CCG1, GGEG2, GGEG5x, GGEG5 and GGSWG1) was gauged and sampled by bailing three volumes and then the same procedure was followed as the cored holes. Swamp and sandstone piezometers were installed by the mining company prior to our research study. Swamp piezometers were installed to the base of the swamp, where auger refusal did not allow further progress. The typical installation depth is around 1 m to 1.3 m. To minimise disturbance of the swamp, all swamp piezometers were installed by manual coring an 80 mm diameter hole to refusal and pushing the slotted 50 mm diameter PVC tube into the hole. A full PVC casing was attached to the top of the pipe. The sandstone piezometer is 10.7 m depth with 50 mm diameter PVC casing that includes a 3-meter length of screen at the bottom of the hole. The piezometer installation was extended with casing to the top. The top was sealed by grout, and a steel monument constructed for protection. Surface water samples were collected at the downgradient end of the swamp but also at one upgradient location (GGES2) where this was possible.

For this study ANSTO provided event based  $\delta^{18}$ O and  $\delta^{2}$ H data for precipitation from Mt Werong for the period covered in this research. Mt Werong (Hughes and Crawford, 2013) is located around 70 km south of this research site, however, within the same climatic environment and at similar elevation to the investigated swamps.

Reviewer's Comment -Section 3.2 needs to be entirely re-written – the analytical descriptions are incoherent. You need to give the delta values of all calibration standards. There is insufficient detail given to give confidence in the results.

**Response**

Thanks, Section 3.2 was considerably expanded and fully re-written with additional detailed information on the sample analysis, secondary standards used along with their delta values and primary standard and reproducibility of the results. Please see below for fully revised Section 3.2

**3.2 Sample analysis**

The swamp sediment samples were analysed for  $\delta^{18}$ O and  $\delta^{2}$ H by  $H_2O_{(water)}-H_2O_{(vapour)}$  pore water equilibration (Wassenaar et al, 2008; Wassenaar and Hendry, 2008) and off-axis ICOS. The Los Gatos (LGR) water vapour analyser (WVIA RMT-EP model 911-0004) located at UNSW, Australia was used for sample analysis. All samples and standards have been stored at 4°C prior to the analysis, and all have been allowed the same time on the laboratory bench in the temperature-controlled laboratory during preparation and have followed the same treatment. Samples (n=34, 14 and 27 for each of the sampling events) were prepared in the lab by transferring the samples to a tough Ziploc bag. The 1 L sample bags were inflated with dry air and left on the laboratory bench within the controlled temperature for a period of between 17 to 24 hours to allow vapour equilibration. Timing of vapour equilibration is dependent on compactness of the core sample, whether it is broken in pieces and if it is unconsolidated (Wassenaar et al, 2008). The timing varies for different geologic materials and must be determined experimentally (Hendry et al, 2015) for each material. Work by Wassenaar et al (2008) and David et al (2015) indicates that for compact, low permeability, consolidated materials around 3 days is required for core samples equilibration. The samples in this research are broken down, unconsolidated, saturated and high permeability therefore shorter equilibration time is considered justified. In addition, the optimal equilibration time in this research is considered to be achieved when headspace water content of 23,000 to 28,000 ppm H2O was measured in the bag. This headspace water content is important for accurate sampling (Hendry et al, 2015). Once the sample has reached complete isotopic equilibrium, the vapour was collected by perforating the bag containing sample with a sharp needle and transferring it directly from the bag to the LGR vapour analyser. The connection between the needle and the LGR inlet fitting was via a flexible, thick wall, soft plastic tube, fitted tightly with fittings on both sides. The tight fitting was required to limit the atmospheric air ingress into the LGR. The contamination by atmospheric air during sampling is considered negligible. This is based on the measurement of ambient air moisture of around 14,000 to 15,000 ppm, while the headspace for samples had a range of 23,000 to 28,000 ppm H2O.

Analysis of the vapour sample was undertaken along with the standards (1 ml) prepared in the similar manner to the core samples. The equilibration time for standards was around 20 minutes based on literature and air moisture (Wassenaar et al, 2008). A new set of three standards (one primary and two secondary) were run after every third sample. It is not possible to sample the headspace repeatedly using this technique, as 1 L headspace only allows sampling once (60-90 seconds). Repeated inflating of the same sample with dry air results in incorrect readings. Following each set of samples and standards, the analysis was suspended for a period of around 10-15 minutes, to allow the LGR to reach the stable atmospheric air readings and reduce any memory effect. Linear regression for  $\delta^{18}O$  and  $\delta^{2}H$  was established between the liquid values for standards and raw headspace vapour (fractionation factor at 25°C) readings for the same standards. Regression was used to calibrate the vapour results for samples. Calibration was undertaken with two secondary  $\delta^{18}$ O and  $\delta^{2}$ H standards (Los Gatos 2A -16.14 ‰  $\delta^{18}$ O and -123.6 ‰  $\delta^{2}$ H and 5A -2.80 ‰  $\delta^{18}$ O and 9.5  $\% \delta^2 H$ ) and normalised with one primary VSMOW/VSMOW2 standard run during the analysis. LGR standards were stored in accordance with the protocol, at 4°C, and ampules were fully sealed to prevent any exchange with the atmosphere. The use of LRG standards as secondary standards has been used in other studies such as Penna et al (2010) on reproducibility and repeatability of the laser absorption spectroscopy measurements and was found that LGR standards performed satisfactorily.

Replicate sample analyses using direct vapour equilibration method (mean difference of 6 samples) indicate reproducibility of results in our research within an uncertainty of 0.68 ‰ for  $\delta^2$ H and 0.04 ‰ for  $\delta^{18}$ O. Reported instrument precision of 0.5 ‰  $\delta^2$ H and 0.15 ‰  $\delta^{18}$ O over 10 seconds and drift of 0.75 ‰  $\delta^2$ H and 0.3 ‰  $\delta^{18}$ O over 15 minutes was minimised by correcting the readings. The data set for each sample was corrected for drift by back correction using standards within each set and then applying the same regression analysis to the relevant samples. For each sample the standard deviation and instrument drift error were calculated. Following the standard operating procedures, the precision in this research was 0.6 ‰ for  $\delta^2$ H and 0.23 ‰ for  $\delta^{18}$ O over 60 seconds.

Hendry et al (2015) report the analytical precision of the vapour equilibration method (±0.40 ‰ for  $\delta^{18}O$  and ±2.1 ‰ for  $\delta^{2}H$ ) to be comparable or better than physical extraction from cores using highspeed centrifugation, cryogenic micro-distillation, azeotropic and microwave distillation or isotope ratio mass spectrometry (IRMS) based direct equilibration methods as discussed in Kelln et al (2001). Based on work by Allison and Hughes (1983) and Revesz and Woods (1990), the direct vapour equilibration method precision is also better than for methods obtained by chemical water extractions (Hendry et al, 2015). This is achieved through limiting fractionation losses by short storage time, single procedure once the samples are in the laboratory and use of standards and water isotopic data as a cross check.

Water samples (surface water, swamp groundwater and regional groundwater, n=21) were analysed for  $\delta^{18}$ O and  $\delta^{2}$ H by the off axis- integrated cavity output spectrometry (OC-ICOS) technique using an LGR analyser located at UNSW Australia. Two secondary standards and VSMOW/VSMOW2 standard were used to calibrate and normalise the samples.

Gravimetric water content (ASTM D2974-14, 2014 and ASTM D2216-10, 2010 was measured by weighing the sediment samples (n=70), drying at 100°C for 24 hours and re-weighing (Reynolds, 1970). 100% gravimetric water content relates to water holding capacity and organic content of the material. The analysis was undertaken at the School of Mining Engineering, UNSW Australia. Organic matter content was measured by loss on ignition method (LOI), by weighing (following initial drying at 100°C) and drying in furnace oven at 550°C (Heiri et al, 2001). The analysis was conducted at the Water Research Laboratory, UNSW Australia.

Precipitation samples were analysed at the ANSTO Environmental Isotope Laboratory using a cavity ring-down spectroscopy method on a Picarro L2120-I Water Analyser (reported accuracy of  $\pm 1.0, \pm 0.2$  ‰ for  $\delta^2$ H and  $\delta^{18}$ O respectively). The lab runs a minimum of two in-house standards calibrated against VSMOW/VSMOW2 and SLAP/SLAP2 with samples in each batch.

For simple statistical analysis of moisture content, precipitation and organic matter content, an XLStat software package (XLStat, 2017) was used. The Barnes and Allison (1988) model was implemented for this project using R (R core team, 2013), to investigate the evaporative losses based on isotopic composition of water. For the Barnes and Allison (1988) model volumetric water content was calculated from the measured gravimetric water content and bulk density. Bulk density was obtained from known lithology and measured data (Cowley et al, 2016) and porosity data from a swamp study by Walczak et al (2002). To estimate effective liquid diffusivity of isotopes, particle size and tortuosity values were obtained from the literature (Maidment, 1993; Shackelford and Daniel, 1991; Barnes and Allison, 1988).

**General Reviewer's Comment**

I have grave doubts about the results for porewater stable isotopes and suspect the trends (or differences) between samplings may be due to evaporation artifacts. What was neglected to mention is the time lapsed between core sample collection (coring, stored in Ziplok) and the sample preparation (ie inflation) for isotope analysis. Was this storage hours, weeks, days? Ziplok bags are only good for a couple of days before evaporative loss occurs.

**Response**

Thanks, the updated Section 3.1 and 3.2 explains how samples were collected and prepared and how any potential evaporation was avoided during sampling, storage and preparation for and during the analysis. Samples were collected in accordance with the methods reported in Wassenaar et al (2008), they were stored at 4°C for three days after collection (coring) and were then prepared following methods of Hendry et al (2015), which improved on methods developed by Wassenaar et al (2008).

The use of clear Ziplock bags, Isopack and clear bags for storage of samples for pore water analysis has been found (Hendry et al 2015) to result in evaporation loss and isotopic

fractionation after 10-15 days after sample collection. In our research, we stored samples for only 3 days before analysis, in tough plastic bags (rather than Ziplock bags) sealed twice on each side after extracting air. Additionally, each vacuum packed (packed in the tough highgrade food plastic bag with air extracted) and double sealed sample was placed in an additional tough plastic bag with air space removed. Therefore, it is improbable that the results were artefacts of the storage process. Pg 7 Lines 19 to Pg 8 Line 4

**General Reviewer's Comment**

If variable periods of times elapsed for the samplings, the samples could have been subjected to differential evaporative loss (ie why is the groundwater isotopic composition constant). There were no gravimetrics controls used, nor isotopic field controls to give confidence in this method (at least as it is described).

**Response**

We agree with the reviewer that different time periods were subject to different evaporation losses. This is observed in difference between the samples collected in May 2016 after dry and warm period compared to October 2016 after wet and cool period. All isotopic field controls during sampling and analysis were implemented; this included: quick storage in tough plastic bags, immediate double bagging during collection, vacuum packing with double seals in tough plastic food grade bags and double sealing. It is assumed that Reviewer relates gravimetric control to the weight of sample taken in the field. In the field the sample size was not weighted, as this was not considered important. What was considered important was that there was enough sample which will have a minimum of 5% moisture content to allow vapour equilibration method to be used. Water samples were also collected and analysed for cross refence with pore water.

The isotopic field controls were added to Section 3.1 of the manuscript Pg 8 Line 2-3

**General Reviewer's Comment**

The Los Gatos "standards" used are not certified RMs, and should never be used for calibration. They have been revised at least 5 times due to improper storage (at LGR) in the past years.

Response: We used V-SMOW2 (VSMOW/VWMOW2) as primary reference standard, and Los Gatos standards as secondary. We are aware that there is SLAP which is the secondary standard distributed by IAEA, but have not used it. LGR standards in our lab were stored properly, at 4°C, and after usage the top was fully sealed to prevent any exchange with the atmosphere (opened once prior to this analysis). The use of LRG as secondary standards has been used in other studies such as Penna et al (2010) which undertook a study on reproducibility and repeatability of the laser absorption spectroscopy measurements and found LGR standards to work satisfactorily.

In addition, the water samples analysed separately in a different LGR apparatus with in house standards and V-SMOW2 as the primary standard, returned the  $\delta^2$ H and  $\delta^{18}$ O similar to the pore water results, confirming the accuracy and calibration.

Text has been added to manuscript in Section 3.2 (Pg 9, Line 24-30) to explain the LGR storage protocol was followed in the lab.

**Reviewer's Comment SMOW / VSMOW do not exist – VSMOW2 does. Which was it?**

Response It was VSMOW/VSWMOW2, this is now corrected in the manuscript. (Pg 9 Line 26)

Reviewer's Comment The introduction is too long – suggest deleting lines Page 1, lines 12-18 (unrelated to wetlands)

**Response**

The text has been removed from Page 3, lines 12-18 as suggested.

**Reviewer's Comment**

Continuous line numbering would have been useful for reviewers.

Response: This manuscript was prepared on the template suggested by HESS, and conversion to pdf is outside of the control of Authors of this paper.

**Reviewer's Comment**

Many places have a "the" or "a" added or missing. (i.e. title, Page 3 line 1, etc.).

Response Thanks, "a" and "the" were added or removed, as suggested in the title and Page3 line 1, and at other places in the manuscript.

**Reviewer's Comment**

Page 1 line 4, not climate change (aka CO2)-> rather, paleoclimate.

**Response**

Thanks, this was reworded to paleoclimate in manuscript.

**Reviewer's Comment**

Page 4, line 8 "Following such extreme setttings: :: ( what does that mean?)

**Response: Thanks, this has been clarified as "long dry periods"**

Reviewer's Comment Page 4, line 16-20 – please rewrite the objectives in a clearer manner. A hypothesis would be a good place to start.

Response: Line 16-20 related to objectives has been rewritten and hypothesis added. The paragraph now reads:

The objective of this research was to improve understanding of intact swamps under natural conditions by characterizing the sediments, waters and organic materials and developing the conceptual model for the swamp system. We hypothesise that groundwater is an important contributor to the swamp water balance and is connected to the regional groundwater system. Therefore, we investigate, for the first time using direct equilibration method, the vertical profiles of stable  $\delta^{18}$ O and  $\delta^{2}$ H isotopes of pore water within the swamp. We then compare those to stable isotopes of regional groundwater, rainfall and surface water as endpoint members. Supported by sediment lithology logs, organic and carbon content of sediments these stable isotope results enabled the development of a conceptual model of the swamp water cycle.

**Reviewer's Comment Page 8 line 25-26 - IRMS cannot be an LGR analyser!**

Response:

Thanks, this typing error is now corrected in manuscript to Off axis- integrated cavity output spectrometry (OA-ICOS). Pg9 Line 11-14

Reviewer's Comment Figure 6 caption error and use of d18O/d2H – the slash suggests 'or' when you mean'and'. Superscripts missing. Suggest using the same Y-axis scaling on all figures. Why are the symbols for the same thing different in each panel? Very confusing to look at and compare!

**Response**

Thanks, the caption has been changed to say  $\delta^{18}$ O and  $\delta^{2}$ H and superscripts added to Figure 6. Same y- scaling has been used on all figures as suggested, and all symbols for the same sample types are the same in each panel as recommended by the Reviewer3. Please see attached Figure 6 (now Figure 7)

**Application of the pore water stable isotope method and hydrogeological approaches to characterise a wetland system**

David, K.1; Timms, W.42; Hughes, C.E.23, Crawford, J23; McGeeney, D.34

1School of Minerals and Energy Resource engineering, and Connected Waters Initiative, University of New South Wales, Sydney, Australia

2 School of Engineering, Deakin University, Waurn Ponds, Australia

23Australian Nuclear Science and Technology Organisation, Sydney, Australia

10 34Australian Museum, Sydney, Australia

Correspondence to: Katarina David (k.david@unsw.edu.au)

**Abstract**

5

15 Three naturally intact wetland systems (swamps) were characterized based on sediment cores, analysis of surface water, swamp groundwater, regional groundwater and pore\_water stable isotopes. These swamps are classified as temperate highland peat swamps on sandstone (THPSS) and in Australia they are listed as threatened endangered\_ecological communities\_under State and Federal legislation.

This study is the first application of the stable isotope direct vapour equilibration method in a wetland, enabling-aiming at quantification of the contributions of evaporation, rainfall and groundwater to swamp water balance. This technique potentially enables understanding of the depth of evaporative losses and the relative importance of groundwater flow within the swamp environment without the need for intrusive piezometer installation at multiple locations and depths. Additional advantages of the stable isotope direct vapour equilibration technique include detailed spatial and vertical depth profiles of δ18O and δ2H, with good accuracy comparable to the other physical and chemical extraction methods. porewater

**25 compression technique.**

Depletion of  $\delta^{18}$ O and  $\delta^{2}$ H in pore\_water with increasing depth (to around 40-60 cm depth) was observed in two swamps, but remained uniform with depth in the third swamp. Within the upper surficial zone, the measurements respond to seasonal trends and are subject to evaporation in the capillary zone. Below this depth the pore water  $\delta^{18}$ O and  $\delta^{2}$ H signature approaches that of regional groundwater indicating lateral groundwater contribution. Significant differences were found in

30 stable pore water isotopes for-samples collected after dry weather period compared to wet periods where recharge of depleted rainfall was apparent.

The organic rich soil in the upper 40 to 60 cm retains significant saturation following precipitation events and maintains moisture necessary for ecosystem functioning. An important finding for wetland and ecosystem response to changing groundwater swamp groundwater conditions (and potential ground movement) are is the observations that basal sands are observed -underlayto underlay these swamps, allowing relatively rapid drainage at the base of the swamp and interaction

35 observed\_underlayto underlay these swamps, allowing relatively rapid drainage at the base of the swamp and interactively with-lateral groundwater contribution.

Based on the novel stable isotope direct vapour equilibration analysis of swamp sediment, our study identified the following important processes: rapid infiltration of rainfall to the water table with longer retention of moisture in the upper 40-60\_cm and lateral groundwater flow contribution at the base. This study also found, that evaporation estimated using stable isotope direct vapour equilibration method is more realistic compared to reference evapotranspiration (ET). Importantly, if swamp discharge data were available in combination with pore water isotope profiles, an appropriate transpiration rate\_could be determined for these swamps. Based on the results, the groundwater contribution to the swamp is a significant, and perhaps dominant component of the water balance. Our methods could complement other monitoring studies and numerical water balance models to improve prediction of the hydrological response of the swamp\_to changes in water conditions due to natural or anthropogenic influences.

**1** Introduction**

10

SThe stable isotopes of water (δ18O and δ2H) have been widely used to understand—the groundwater and surface water interaction and recharge processes in aquifer systems (Barnes and Allison, 1988; Cuthbert et al, 2014). The understanding of stable isotope variation and signatures in subsurface allows the identification of flowpaths (Person et al, 2012), hydrological processes on a catchment scale (Rodgers et al., 2005; Vitvar et al., 2005), climate change (Huneau et al., 2003; Tadros et al., 2016) and aquifer heterogeneity (Hendry and Wassenaar, 2009). Although less common than liquid water isotope studies, pore water (vapour) stable isotope techniques have been applied to investigate the—groundwater flux and interpret the paleoenvironment (Hendry et al., 2013; Harrington et al., 2013), determine slope runoff contribution to groundwater (Garvelman et al., 2012) and characterize multiti-layered sedimentatrysedimentary sequences (David et al., 2015). Pore water
stable isotope analysis was less common due to sampling difficulties (Sodenberg et al., 2011) and high cost (Harrington et al., 2015).

2013), until advances in laser spectroscopy improved the speed and accuracy of the analysis (Wassenaar and Hendry and Wassenaar, 20098; Hendry et al, 2015).

International eExamples of wetland research indicate that it remains challenging to quantify some components of the water balance (Bijoor et al., 2011), since only a few studies have investigated the groundwater contribution (Hunt et al., 19986).

- 25 The application of stable isotopes δ44O and δ2H is typically limited to surface water isotopes to understand the hydrology of the swamp system (Nyarko et al., 2010; Bijoor et al, 2011), the effect of transpiration (Wang and Yakir, 2000) and to improving the water balance study (Schwerdtfeger et al., 2014; Levy et al, 2016). Recent international research on wetland and lake systems focuses on stable water isotopes and environmental tracers (Mandl et al., 2017; Meier et al. 2015; Kaller, et al. 2015) to understand paleoclimate and processes in these systems.
- 30 The significance of groundwater in maintaining the function of swamp ecosystem function has been discussed in international the literature (Chang et al, 2009; Kaller et al, 2015). In Australia, swamp studies have evaluated geomorphology (Fryirs et al, 2016; Cowley et al, 2016), management (Kohlhagen et al, 2013), the relationship between

vegetation and groundwater (Hose et al, 2014), the processes that result in denudation and sedimentation in the headwaters of the swamps (Prosser et al, 1994), natural and anthropogenic vegetation change in swamps (Bickford and Gell, 2005) and the impact of mining subsidence (CoA, 2014b). However, there is limited literature on the importance of groundwater storage, flow and which source of water source contributes to maintain moisture in swamp systems.

- 5 The tremperate highland peat swamps on sandstone (THPSS) swamps in Eastern Australia are endangered ecological communities with endemic flora and fauna that are dependent on hydrological balance water balance. The direct influences on the water regime of these swamps are changes in weather patterns, natural storm activity (Smith-Smith et al. 2001), fire (Middleton and Kleinebecker, 2012; CoA, 2014a) and the effects of mining subsidence (CoA, 2014b). Ecologically, the THPSS swamps are sensitive to changing swamp moisture content (CoA 2014a; Young, 2017), and the importance of
- 10 groundwater in these systems has been discussed by Eamus and Froend (2006), Fryirs et al7 (2014) and7 Hose et al7 (2014). Based on the substantialThere is direct and indirect evidence of saturation within thethat the TPHSS swamps (Newnes Plateau shrub swamp) are mostly saturated, including \_such as-vegetation patterns, specific species of plants, piezometer records, presence of certain plant species, age dating (Benson and Baird, 2012) piezometer records (Benson and Baird, 2012) and spring discharge (Johnson, 20076). Maintaining , there is a clear indication that the maintenance of groundwater
- 15 groundwater levels and groundwater discharge to swamps is necessary for the health of for such a wetland systems (Clifton and Evans, 2001).

Despite the awareness of these factors, there is limited research to predict hydrogeological changes in swamps behaviour and ecological response under changing water conditions (Mitsch and Gosselink 2000; Cowley et al, 2016). Furthermore, studies describing the impact of environmental changes on swamp ecology are rare and there is insufficient understanding of natural

- 20 variation in swamp ecology over time (CoA, 2014a). Although groundwater is often assumed to be important for the sustainability of wetlands and swamp ecosystems, very little is known on how these systems would respond to changing swamp groundwater and regional groundwater conditions. The existing literature recognizes that natural variation in swamp ecology is not well understood given the complexity and interaction of swamp groundwater, groundwater and surface water. Long drought periods in Australia result in temporary drying of water bodies, pooling of water and reduction in baseflow
- 25 contribution to wetlands (Lake, 2003). Following such extreme settingslong dry periods, the rainfall may not be sufficient for a swamp to recover its original condition (Bond et al, 2008; Middleton and Kleinebecker, 2012+). For example, Smith et al. (2001) report loss of swamps in Africa and Australia as a result of climate change. Vegetation removal, drainage of swamp and undermining are known to be critical human impacts (Kohlhagen et al, 2013; Valentin et al, 2005). As such, mining and urbanisation have degraded and considerably damaged the TPHSS swamps (CoA, 2014b), although the actual impact often
- 30 cannot be quantified due to limited baseline and monitoring data (Paterson, 2004). However, it is generally recognized that rock fracturing, changes in elevation gradient and catchment conditions can compromise the stability and integrity of these swamps (CoA, 2014b).

The objective of this research was to improve understanding of intact swamps under natural conditions by characterizing the sediments, waters and organic materials and developing the conceptual model for the swamp system. We hypothesise that

groundwater is an important contributor to the swamp water balance and in-connectionis connected to-with the regional groundwater system. Therefore, For this we investigate for the first time using direct equilibration method, the vertical profiles of stable  $\delta^{18}$ O and  $\delta^{2}$ H isotopes of pore water within the swamp. We then compare those to stable isotopes of regional groundwater, rainfall and surface water as distinct-endpoint members. Along with Supported by logs of sediment lithology logs, organic and carbon content of sediments these stable isotope results enabled the development of a conceptual model of the swamp water cycle.

**2. Site description**

5

10

The research site is located west of Sydney, NSW, Australia, in the World Heritage listed Blue Mountains on the Newnes Plateau (**Fig.ure 1**) between Lithgow and Blue Mountains local government areas. The elevation of the plateau ranges from 1000-1200 m Australian Height Datum (mAHD).

---

## Author Response (AR3)

**Editor Decision: Publish subject to minor revisions (review by editor)** (09 Nov 2018) by Christine Stumpp

Comments to the Author:
Two referees reviewed the revised version of the manuscript. One reviewer pointed out a substantial improvement and only minor changes are required (mainly imprecise language, see reviewer comment). The still missing precise language was also pointed out by the other reviewer (not in the public report though).

Response

Thanks to reviewers for helping to improve this manuscript. The changes mentioned by Reviewer1 have now been addressed and are listed below in Response to Reviewer1.

Comment

This person also pointed out some other critical points, which at least should be discussed (volatile organic compounds) or where authors should be more precise (which information can be gained from hydro(geo)logical values like water balance and groundwater levels and for which information are the isotopes required) or less enthusiastic (first study using the direct vapor method in wetlands --> it has been used in unsaturated sediments, soils and in saturated sediments (aquifer, aquitard) before and a wetland isn't anything different). For details, please see report.

Response

Thanks to Reviewer2 for providing useful comments. Two critical points that reviewer2 mentioned have now both been addressed in separate document and discussion/ changes made in the text. These changes relate both to presence of VOC and clarification regarding the groundwater levels and application of and the reason for using the stable isotopes in this study. Please see below in Response to Reviewer2.

Comments

Additionally, (i) carefully check the entire reference list; e.g. Sodenberg et al., 2011, should be Soderberg et al., 2012;

Response

The reference list has been checked and changes made accordingly.

Comment

 (ii) check tenses and spelling;

Response

This has been checked throughout the document

Comment

(iii) define all abbreviations;

Response

Where abbreviations are used they have been defined first time they are mentioned.

 (iv) check y-axis in Figure 8: is it really 8000 cm? not 800 cm?;

Response

Thanks, the y-axis on Figure 8 has been updated to read 800 cm.

Comment

(v) the term "evaporation line" should not be used because it was later discussed that this is a result of differently evaporated water with different initial values.

Response

We agree that this is a complex evaporation process with sources that are a difficult to define with a mix of groundwater, rainwater and previously evaporated porewater. However, the fact that they fall along fairly consistent lines suggests there is actually some similarity in the original values of the water. We have removed the term evaporation line and made it clear that this is not a single source. We have modified this sentence to say: *Pore water samples at CC and GG in May 2016 were clearly evaporated, lying along lines with slopes of 4.2 and 4.6 respectively, even though no single initial value for porewater evaporation was discernible.*

If these changes requiring minor revisions are included into the manuscript, it can be accepted for publication in HESS.

Comment

There are still minor edits required around imprecise language. For example, isotopic delta values cannot be enriched or depleted (e.g. page 14, line 14). There are several instances of this misuse throughout the paper. Change to "enriched in 18O or 2H" , etc. Also several cases where they say the "water is enriched or depleted (but in what?) relative to rainfal/Xl" (e.g. page 15 line 6). Search the paper and fix all of these cases.

Response

Thanks for pointing out the minor edits relating to imprecise language.

The words "depleted and enriched" have now been corrected in the manuscript as several locations, these are: Page 1 Line 30, recharge of depleted rainfall (low $\delta^{18}O$ and $\delta^{2}H$) was apparent", Page 14 Line 9 "'plot within the lower $\delta^{18}O$ and $\delta^{2}H$ rainfall range'', Pg 15 Line 8 '' are lower relative to pore water samples for $\delta^{18}O$'', Pg 16 Line 13 '' have lower $\delta^{18}O$ after'', Pg 19 Line 12 '' based on lower $\delta^{18}O$ values'', Pg 21 Line 28 ''more isotopically depleted events'', Pg 22 Line 15 "' typically lower in $\delta^{18}O$ and $\delta^{2}H$'', Pg 22 Line 29 '' lower $\delta^{18}O$ and $\delta^{2}H$ values", Pg 14 Line 15 '' enriched in $^{2}H$", Pg 15 Line 5 '' enriched in $^{18}O$ and $^{2}H$'', Pg 16 Line 23 '' isotopically enriched in $^{18}O$ and $^{2}H$".

Response to Reviewer 2

Comment

Despite the major revisions David et al. have made, I regret to say that I am still unable to see the lessons that readers could learn from this manuscript. Therefore, I cannot recommend this manuscript for publication as explained in detail below.

The authors emphasize two main aspects: First, being the first research group to apply the direct vapor equilibration method to a wetland. And second, to characterize the status, namely the groundwater regime, of an intact, endangered wetland mainly by means of porewater stable isotope depth profiles.

Regarding the first aspect, as I mentioned before, the direct vapor equilibration method was never methodically restricted to non-wetlands and therefore does not constitute a challenge itself.

Response

We agree that the method has been used before in soils and its application to wetlands has not been restricted. In fact, the application of VE technique in association with other information and modelling for wetlands can be very beneficial. The VE technique can assist in estimating the evaporation and evapotranspiration in complex wetlands environments without the need of a detailed long term vegetation monitoring. It is for these reasons that we chose to apply the method in the wetlands and highlight that our study is the first to apply the method in wetlands.

This has been reworded as *This study applies the stable isotope direct vapour equilibration method…*

Comment

Nonetheless, the manuscript could have been a valuable contribution to existing isotope knowledge if the authors had proven that (or how) despite "strong organic smell", which does in fact describe the presence of volatile organic compounds, laser-based water-vapor isotope measurements yield trustworthy data for organic-rich sediments. This would actually have been new. The issue of laser-based water stable isotope analyses being flawed by VOCs has been discussed extensively in the existing literature (e.g. Brand et al., 2009, DOI: 10.1002/rcm.4083; West et al., 2011, DOI: 10.1002/rcm.5126; Martin-Gomez et al., 2015, doi: 10.1111/nph.13376; Orlowski et al., 2018, doi: 10.5194/hess-22-3619-2018; Millar et al., 2018, DOI: 10.1002/rcm.8136) and was made aware to the authors. For this purpose, however, a method comparison or structured laboratory experiments would have been necessary. Neither of which is/are presented or at least discussed. Further, the performed modeling efforts based on the presented, potentially flawed porewater isotope data remain therefore highly questionable.

Response

Thank you for your observations regarding VOCs. The 'strong organic smell', relates to presence of woody peat in the sample and in this case is not necessarily an indication of presence of VOCs. However, as pointed out by the reviewer, there are many recent studies which indicate spectral interference in OA-ISOS method from the presence of organic contaminants – VOCs and alcohols. These studies have been performed using mainly plant portions as samples, and not peat. Miller et al (2018) in their study indicated that more research is needed on the woody plants (ie more similar to woody peat). A research undertaken by Mezhibor and Bonn (2014) on peat indicated that the VOCs naturally present in peat are in very low vol% concentrations (0.2 ppb). In our study, we have used the post processing LGR software on water samples analysed from the same swamps where no spectral interference was noted.

We have added the text as follows to explain this (Pg 9 Line 26):

*The vapour equilibration method is sensitive to presence of volatile organic compounds (VOC) in samples (Millar et al, 2018) and may require spectral correction post analysis, however usually the presence of these hydrocarbons is not known (Hendry et al, 2015). There are limited studies related to quantification of VOC in peat (Mezhibor and Bonn, 2014) and impact of organic matter on isotopic composition (Orlowski et al, 2016). For water samples, the LGR's post analysis software automatically applied check for spectral interference. Similar approach was reported by Millar et al (2018), Schultz et al (2011) and Orlowski et al (2018). There was no evidence of spectral contamination. However, this analysis was not possible for vapour analysis of soil samples due to the different processing method. The similarity between the isotopic composition of water and vapour collected from the same horizon suggests a strong interaction between groundwater and porewater; as the groundwater analyses show no evidence of VOC contamination, there is no reason to suspect that the porewater samples would contain concerning levels of VOCs despite the high peat organic content. Furthermore, in one of the rare VOC quantification studies on peat, Mezhibor and Bonn (2014) found that peat had mean concentration of isoprene and acetaldehyde (VOCs characteristic for natural plant organic emission) up to 0.26 ppb. The ecosystem in their study is similar to this one and for such low volume % concentrations the spectral corrections are not considered to be necessary.*

Comment

Regarding the second aspect, the authors pointed out, that the health of a wetland depends on the persistence of high groundwater levels. These can be observed directly and in situ via the available piezometers or indirectly via vegetation patterns or specific species of plants. A wetland being subject to mining activities will first display changes in these phenomena. The groundwater regime, necessary for the maintenance of such high groundwater levels, can be attributed to stable piezometer levels and gradients. In this context, the assumption of lateral groundwater flow within the swamps is mandatory, given the observed gradients in piezometer levels. This strongly contradicts the model selection. Changes in the groundwater regimes would be indicated by changes in groundwater levels and gradients which can easily be monitored and even logged continuously. I am therefore unable to see what sense it makes to recommend putting that much effort into taking water and sediment samples on several occasions and having them analyzed at remote laboratories with ambiguous modeling results, i.e. similar isotope patterns (Fig. 7a) which led to unreasonably differing evaporation estimates (1 mm/day vs. 4-9 mm/day). The presented evaporation estimates led to the indirect and only qualitative conclusion of significant groundwater flow into the swamps. "Fresh" groundwater at down-gradient locations, however, indicates that evaporation must be considerably smaller than water export via groundwater discharge or surface water runoff. These components of the water balance are therefore more important than evaporation to be represented quantitatively in the desired conceptual model.

Response

We agree that standard hydrological monitoring methods have good applications here. However, access to this site and to monitoring infrastructure was restricted, given protection of the swamps under State and Federal legislation. This informed our research approach.

 Available research indicates that the persistence of groundwater, with the swamp at least partially saturated, is necessary for wetland health. Groundwater levels in the swamp vary over time in response to rainfall and evaporation, however, observations are limited to approved piezometer sites. The groundwater flow is assumed to be both vertical and lateral as presented in the conceptual model.

Although evaporation during dry periods is estimated at 4-9 mm/day, groundwater remains fresh due to contributions from regional groundwater which is below the 2 m depth of evaporation. The techniques applied in this paper can provide such additional spatial information to augment limited

monitoring infrastructure and our understanding of the relative importance of the water balance components.

The following paragraph has been added to provide clarification on the information required for the water balance and the use of stable isotopes (Pg 22, Line 10):

*It is clear that the water balance in swamp can be obtained if all components are known (rainfall, runoff, groundwater contribution, evaporation, evapotranspiration, discharge), however due to the THPSS swamps being difficult to access and being protected under State and Federal legislation it is not possible to undertake intrusive drilling to obtain all hydrogeological information. The application of stable isotopes has enabled estimation of evapotranspiration from the swamp and assisted in development of the conceptual model.*

**Application of the pore water stable isotope method and hydrogeological approaches to characterise a wetland system**

David, K.[1]; Timms, W.[2]; Hughes, C.E.[3], Crawford, J[3]; McGeeney, D.[4]

[1]School of Minerals and Energy Resource Engineering, and Connected Waters Initiative, University of New South Wales, Sydney, Australia
[2] School of Engineering, Deakin University, Waurn Ponds, Australia
[3]Australian Nuclear Science and Technology Organisation, Sydney, Australia
[4]Australian Museum, Sydney, Australia

*Correspondence to*: Katarina David (k.david@unsw.edu.au)

**Abstract**

Three naturally intact wetland systems (swamps) were characterized based on sediment cores, analysis of surface water, swamp groundwater, regional groundwater and pore water stable isotopes. These swamps are classified as temperate highland peat swamps on sandstone (THPSS) and in Australia they are listed as threatened endangered ecological communities under State and Federal legislation.

This study  applies the stable isotope direct vapour equilibration method in a wetland, aiming at quantification of the contributions of evaporation, rainfall and groundwater to swamp water balance. This technique potentially enables understanding of the depth of evaporative losses and the relative importance of groundwater flow within the swamp environment without the need for intrusive piezometer installation at multiple locations and depths. Additional advantages of the stable isotope direct vapour equilibration technique include detailed spatial and vertical depth profiles of $\delta^{18}O$ and $\delta^2H$, with good accuracy comparable to other physical and chemical extraction methods.

Depletion of $\delta^{18}O$ and $\delta^2H$ in pore water with increasing depth (to around 40-60 cm depth) was observed in two swamps but remained uniform with depth in the third swamp. Within the upper surficial zone, the measurements respond to seasonal trends and are subject to evaporation in the capillary zone. Below this depth the pore water $\delta^{18}O$ and $\delta^2H$ signature approaches that of regional groundwater indicating lateral groundwater contribution. Significant differences were found in stable pore water isotope samples collected after dry weather period compared to wet periods where recharge of depleted rainfall (with low $\delta^{18}O$ and $\delta^2H$ values) was apparent.

The organic rich soil in the upper 40 to 60 cm retains significant saturation following precipitation events and maintains moisture necessary for ecosystem functioning. An important finding for wetland and ecosystem response to changing swamp groundwater conditions (and potential ground movement) is that basal sands are observed to underlay these swamps, allowing relatively rapid drainage at the base of the swamp and lateral groundwater contribution.

Based on the novel stable isotope direct vapour equilibration analysis of swamp sediment, our study identified the following important processes: rapid infiltration of rainfall to the water table with longer retention of moisture in the upper 40-60 cm

and lateral groundwater flow contribution at the base. This study also found, that evaporation estimated using stable isotope direct vapour equilibration method is more realistic compared to reference evapotranspiration (ET). Importantly, if swamp discharge data were available in combination with pore water isotope profiles, an appropriate transpiration rate could be determined for these swamps. Based on the results, the groundwater contribution to the swamp is a significant, and perhaps

5    dominant component of the water balance. Our methods could complement other monitoring studies and numerical water balance models to improve prediction of the hydrological response of the swamp to changes in water conditions due to natural or anthropogenic influences.

**1 Introduction**

Stable isotopes of water ($\delta^{18}O$ and $\delta^2H$) have been widely used to understand groundwater and surface water interaction and

10    recharge processes in aquifer systems (Barnes and Allison, 1988; Cuthbert et al, 2014). Although less common than liquid water isotope studies, pore water (vapour) stable isotope techniques have been applied to investigate groundwater flux and interpret the paleoenvironment (Hendry et al, 2013; Harrington et al, 2013), determine slope runoff contribution to groundwater (Garvelman et al, 2012) and characterize multi-layered sedimentary sequences (David et al, 2015). Pore water stable isotope analysis was less common due to sampling difficulties (Sodernberg et al, 2012+) and high cost (Harrington et

15    al, 2013), until advances in laser spectroscopy improved the speed and accuracy of the analysis (Hendry and Wassenaar, 2009; Hendry et al, 2015).

Examples of wetland research indicate that it remains challenging to quantify some components of the water balance (Bijoor et al., 2011), since only a few studies have investigated the groundwater contribution (Hunt et al., 1998). The significance of groundwater in maintaining the swampthe swamp ecosystem function has been discussed in the literature (Chang et al,

20    2009; Kaller et al, 2015). In Australia, swamp studies have evaluated geomorphology (Fryirs et al, 2016; Cowley et al, 2016), management (Kohlhagen et al, 2013), the relationship between vegetation and groundwater (Hose et al, 2014), processes that result in denudation and sedimentation in the headwaters of the swamps (Prosser et al, 1994), natural and anthropogenic vegetation change in swamps (Bickford and Gell, 2005) and the impact of mining subsidence (CoA, 2014b). However, there is limited literature on the importance of groundwater storage, flow and which water source contributes to

25    maintain moisture in swamp systems.

The temperate highland peat swamps on sandstone (THPSS) swamps in Eastern Australia are endangered ecological communities with endemic flora and fauna that are dependent on water balance. The direct influences on the water regime of these swamps are changes in weather patterns, natural storm activity (Smith et al. 2001), fire (Middleton and Kleinebecker, 2012; CoA, 2014a) and the effects of mining subsidence (CoA, 2014b). Ecologically, the THPSS swamps are sensitive to

30    changing swamp moisture content (CoA 2014a; Young, 2017), and the importance of groundwater in these systems has been discussed by Eamus and Froend (2006), Fryirs et al (2014) and Hose et al (2014).

There is direct and indirect evidence that the THPPHSS swamps (Newnes Plateau shrub swamp) are mostly saturated, including vegetation patterns, specific species of plants, piezometer records (Benson and Baird, 2012) and spring discharge (Johnson, 2007). Maintaining groundwater levels is necessary for the health for such a wetland system (Clifton and Evans, 2001).

5 Despite the awareness of these factors, there is limited research to predict hydrogeological changes in swamps and ecological response under changing water conditions (Mitsch and Gosselink 2000; Cowley et al, 2016). Furthermore, studies describing the impact of environmental changes on swamp ecology are rare and there is insufficient understanding of natural variation in swamp ecology over time (CoA, 2014a). Although groundwater is often assumed to be important for the sustainability of wetlands and swamp ecosystems, very little is known on how these systems would respond to changing swamp groundwater

10 and regional groundwater conditions. The existing literature recognizes that natural variation in swamp ecology is not well understood given the complexity and interaction of swamp groundwater, groundwater and surface water.

Long drought periods in Australia result in temporary drying of water bodies, pooling of water and reduction in baseflow contribution to wetlands (Lake, 2003). Following such long dry periods, the rainfall may not be sufficient for a swamp to recover its original condition (Bond et al, 2008; Middleton and Kleinebecker, 2012). For example, Smith et al. (2001) report

15 loss of swamps in Africa and Australia as a result of climate change. Vegetation removal, drainage of swamp and undermining are known to be critical human impacts (Kohlhagen et al, 2013; Valentin et al, 2005). As such, mining and urbanisation have degraded and considerably damaged the TPHPSS swamps (CoA, 2014b), although the actual impact often cannot be quantified due to limited baseline and monitoring data (Paterson, 2004). However, it is generally recognized that rock fracturing, changes in elevation gradient and catchment conditions can compromise the stability and integrity of these

20 swamps (CoA, 2014b).

The objective of this research was to improve understanding of intact swamps under natural conditions by characterizing the sediments, waters and organic materials and developing the conceptual model for the swamp system. We hypothesise that groundwater is an important contributor to the swamp water balance and is connected to the regional groundwater system. Therefore, weTherefore, we investigate, for the first time using direct equilibration method, the vertical profiles of stable

25 $\delta^{18}$O and $\delta^2$H isotopes of pore water within the swamp. We then compare those to stable isotopes of regional groundwater, rainfall and surface water as endpoint members. Supported by sediment lithology logs, organic and carbon content of sediments these stable isotope results enabled the development of a conceptual model of the swamp water cycle.

**2. Site description**

The research site is located west of Sydney, NSW, Australia, in the World Heritage listed Blue Mountains on the Newnes

30 Plateau (**Fig. 1**) between Lithgow and Blue Mountains local government areas. The elevation of the plateau ranges from 1000-1200 m Australian Height Datum (mAHD).

[Figure]

**Figure 1: Map of selected Newnes Plateau swamps with location of samples and transects.**

The Triassic Narrabeen Sandstone outcrops over most of the study area. It comprises mainly quartzose sandstone and minor claystone and shale (Yoo et al, 2001). The swamps on the Newnes Plateau are classified as shrub swamps (OEH, 2017) based on the dominant shrub ecological community, and they occur at the highest elevation of any sandstone-based swamp in Australia. These swamps occur in low slope headwaters of the Newnes Plateau as narrow and elongated sites with impeded drainage (OEH, 2017) and are also classified as T~P~HPSS belonging to both headwater and valley infill types (CoA, 2014a). Mapping by Keith and Benson (1988) and Benson and Keith (1990) indicates that the shrub swamps cover 650 ha of land on the Newnes Plateau, with the largest swamp being 40 ha and average size less than 6 ha. Keith and Myerscough (1993) relate the swamps to other upland swamps in the Sydney Basin in terms of biogeography. However, the difference from other T~P~PHSS is the presence of a permanent water table (Benson and Baird, 2012).

The three swamps selected: identified as CC (swamp area 7 ha, catchment area 150 ha), GG (swamp area 11 ha, catchment area 190 ha) and GGSW (swamp area 5 ha, catchment area 57 ha), are in the upper Carne Creek catchment (**Fig. 1**). Carne Creek is a tributary of the Wolgan River (catchment area 5310 ha) that ultimately flows to the Hawkesbury River and Pacific Ocean. The three swamps selected thus have a total area approximately 0.043% of the Wolgan River catchment. Except for

the headwaters including these swamps, the Wolgan River has been designated part of the Colo Wild River area recognizing substantially unmodified conditions and high conservation value (NSW Government, 2008). The swamps in this study are located to the east of current underground mining operations, with coal extraction currently occurring on the western side of the GGSW and GG swamps though not directly below these swamps. The swamps are elongated with gentle gradient and typically terminate with a sandstone rockbar. The swamp groundwater level responds rapidly to rainfall recharge (Centennial Coal, 2016) and there is indication that swamp systems are fed by lateral groundwater inflow (Benson and Baird, 2012). Further indirect evidence for regional groundwater interaction with swamp sediments and long-term saturation are the consistently stable swamp groundwater levels over time (Centennial Coal, 2016). Chalson and Martin (2009) undertook radiocarbon dating on pollen from a swamp on the Newnes plateau and found that the calibrated ages were 11,000 to 7,500 years (sampling depth 55-90 cm) and decreasing to 1,800 years at 40 cm depth. These ages support the existence of the swamps during the early wetter and warmer Holocene, through to seasonally variable climate in the period from mid to late Holocene (Allen and Lindesay, 1998). Given the seasonality of rainfall events, groundwater interaction must have existed through the mid and late Holocene to enable the swamp survival during dry periods.

Two sedimentary formations underlay the Newnes Plateau swamps: the Burralow Formation and the Banks Wall Sandstone. The Burralow Formation underlies the upper headwater part of the swamps and comprises of an interbedded coarse-grained sandstone sequence with frequent fine-grained, clay-rich sandstone, siltstones, claystones and shales. The thickness of Burralow Formation ranges from around 40 m in the upgradient part of the swamps but is absent in downgradient parts of the swamps. Banks Wall Sandstone typically forms the base of lower parts of most swamps (McHugh, 2014).

Information on the natural groundwater regime in the swamps is very limited (CoA, 2014). It is generally considered that the sandstone underlying the swamps provides the barrier to water loss due to its relatively low permeability (CoA, 2014). However, in some cases, joints and bedding planes within sandstone can provide recharge to swamps (Coffey, 2008). NSW DP (2008) considers that headwater swamps, such as the ones described in this study, are likely to be perched above the regional water table. CoA (2014) indicates that regional groundwater in those swamps can interact with the swamp system, but where this occurs the connection is ephemeral as it is dependent on the perched aquifer. The groundwater residence time is short, and the water is fresh. Information available to date suggests that the dominant source of water to Sydney Basin (headwater) swamps is rainfall and run-off recharge (NSW PAC, 2009).

The climate on the Newnes Plateau is temperate with higher rainfall in November to March and lower rainfall from April to October. Average yearly rainfall at the closest long term meteorological station in Lidsdale (Bureau of Meteorology (BoM) Station SN63132, 12 km west of the study area) is 765 mm (890 mAHD) and 1270 mm at Mt Wilson (BoM Station 63246 21 km south-east (SE) of the study area) (1010 mAHD). The temperature varies from an average 19.6°C in summer to 5.8°C in winter (Lidsdale).

**3. Methods**

**3.1 Fieldwork and sampling**

Fieldwork was undertaken during 2016 and 2017 with the swamps in a natural state and recovered from earlier wildfire in 2013. The first sampling event 24th to 25th May 2016 occurred following an extremely dry weather period of four months below the long-term average rainfall (BoM Lithgow Station SN63226, 900 m-AHD, 13 km SW of the study area with 139 years of data records) for February to May (46.6 mm, 36.8 mm, 6.6 mm and 20.8 mm for each of the months). A total of 34 pore water samples and 5 surface and groundwater samples were collected. A repeat sampling on 25th to 26th October 2016 occurred after four months of above average rainfall from June to September (170.2 mm, 102 mm, 61.8 mm and 92 mm for each of the months). During October 2016 sampling event 14 pore water samples and 13 surface and groundwater samples were collected. Sampling on 30th May 2017 occurred under different climate conditions with both above and below average rainfall trend in the months preceding the sampling event. A total of 27 pore water samples, 3 surface and 3 groundwater samples were collected in May 2017. The spatial depth resolution varied from 10 to 20 cm depending on the penetration of the corer. **Figure 2** shows the variation in monthly long-term rainfall (139 years) and comparison with rainfall in 2016 and 2017.

[Figure]

**Figure 2: Long term average monthly rainfall at Lithgow station (Bureau of Meteorology Station No. 063226) compared to rainfall during 2016 and 2017.**

In total seven sediment cores were obtained by coring using a Russian D hand corer (40 mm diameter) to rock refusal (between 0.45 to 1.4 m), and three transects (CC, GG and GGSW) were prepared along the length of the swamps (**Fig 3,4 and 5**). Samples were geologically logged after extraction, by noting the lithology, grain size and roundness, matrix and colour. The hand cored holes were restored by returning soil material to the hole immediately after sampling. This was undertaken to ensure no change occurred to endangered and protected ecological system as a result of sampling. The coring

on CC transect was repeated in October 2016 at a distance of less than 0.5 m from the original hole. The coring locations were selected to represent swamp stratigraphy from upstream to downstream and to provide a spatial coverage across the three swamps. In addition, three cored locations were selected such that they were adjacent to an existing piezometer (CCG1 on transect CC and GGEG2A and GGEG4 on GG transect). The purpose of this, in addition to determining the stable isotope

5 profiles, was to enable comparison with the swamp groundwater measurements and to collect regional groundwater from the underlying sandstone aquifer where possible.

Sediment cores were divided into subsamples of 10-20 cm length, packed into Ziploc bags and kept in cool storage for later analysis of moisture content and organic matter content. The samples for pore water analysis were temporarily double packed in Ziploc bags by minimising the airspace in the bag, stored in the cooled ice box in accordance with the sampling

10 protocol developed by Wassenaar et al (2008) and further improved by Hendry et al (2015). The use of clear Ziplock bags for storage of samples for pore water analysis has been found (Hendry et al 2015) to result in evaporation loss and isotopic fractionation only after 10-15 days after sample collection. The same afternoon after collection, samples were packed in tough high-grade food storage plastic bags with air extracted, double sealed, separately stored in an additional plastic bag and were kept at a 4°C to prevent evaporation. Vacuum packing was required to minimise atmospheric moisture

15 contamination. All isotopic field controls during sampling and analysis were implemented; this included: quick storage in tough plastic bags, immediate double bagging during collection and vacuum packing the same afternoon. Storage time for samples after collection was 3 days in the cool environment (4°C) before they were analysed.

Swamp groundwater was sampled directly from the cored hole and field parameters were measured immediately (pH, electrical conductivity (EC), dissolved oxygen (DO), temperature). This was repeated for all three sampling events; however,

20 some bores were dry and some not accessible. Swamp groundwater and regional groundwater from existing piezometers (CCG1, GGEG2, GGEG5x, GGEG5 and GGSWG1) was gauged and sampled by bailing three volumes and then the same procedure was followed as the cored holes. Swamp and sandstone piezometers were installed by the mining company prior to our research study. Swamp piezometers were installed to the base of the swamp, where auger refusal did not allow further progress. The typical installation depth is around 1 m to 1.3 m. To minimise disturbance of the swamp, all swamp

25 piezometers were installed by manual coring an 80 mm diameter hole to refusal and pushing the slotted 50 mm diameter PVC tube into the hole. A full PVC casing was attached to the top of the pipe. The sandstone piezometer is 8.5 m depth with 50 mm diameter PVC casing that includes a 3-meter length of screen at the bottom of the hole. The piezometer installation was extended with casing to the top. The top was sealed by grout, and a steel monument constructed for protection. Surface water samples were collected at the downgradient end of the swamp but also at one upgradient location

30 (GGES2) where this was possible.

For this study the Australian Nuclear Science and Technology Organisation (ANSTO) provided event based $\delta^{18}O$ and $\delta^2H$ data for precipitation from Mt Werong for the period covered in this research. Mt Werong (Hughes and Crawford, 2013) is located around 70 km south of this research site, however, within the same climatic environment and at similar elevation to the investigated swamps.

**3.2 Sample analysis**

The swamp sediment samples were analysed for $\delta^{18}O$ and $\delta^2H$ by $H_2O_{(water)}$-$H_2O_{(vapour)}$ pore water equilibration (Wassenaar et al, 2008; Wassenaar and Hendry, 2008) and off-axis ICOS. The Los Gatos (LGR) water vapour analyser (WVIA RMT-EP

5    model 911-0004) located at University of NSW (UNSW), Australia was used for sample analysis. All samples and standards have been stored at 4°C prior to the analysis, and all have been allowed the same time on the laboratory bench in the temperature-controlled laboratory during preparation and have followed the same treatment. Samples (n=34, 14 and 27 for each of the sampling events) were prepared in the lab by transferring the samples to a tough Ziploc bag. The 1 L sample bags were inflated with dry air and left on the laboratory bench within the controlled temperature for a period of between 17 to 24

10   hours to allow vapour equilibration. Timing of vapour equilibration is dependent on compactness of the core sample, whether it is broken in pieces and if it is unconsolidated (Wassenaar et al, 2008). The timing varies for different geologic materials and must be determined experimentally (Hendry et al, 2015) for each material. Work by Wassenaar et al (2008) and David et al (2015) indicates that for compact, low permeability, consolidated materials around 3 days is required for core samples equilibration. The samples in this research are broken down, unconsolidated, saturated and high permeability

15   therefore shorter equilibration time is considered justified. In addition, the optimal equilibration time in this research is considered to be achieved when headspace water content of 23,000 to 28,000 ppm $H_2O$ was measured in the bag. This headspace water content is important for accurate sampling (Hendry et al, 2015). Once the sample has reached complete isotopic equilibrium, the vapour was collected by perforating the bag containing sample with a sharp needle and transferring it directly from the bag to the LGR vapour analyser. The connection between the needle and the LGR inlet fitting was via a

20   flexible, thick wall, soft plastic tube, fitted tightly with fittings on both sides. The tight fitting was required to limit the atmospheric air ingress into the LGR. The contamination by atmospheric air during sampling is considered negligible. This is based on the measurement of ambient air moisture of around 14,000 to 15,000 ppm, while the headspace for samples had a range of 23,000 to 28,000 ppm $H_2O$.

Analysis of the vapour sample was undertaken along with the standards (1 ml) prepared in the similar manner to the core

25   samples. The equilibration time for standards was around 20 minutes based on literature and air moisture (Wassenaar et al, 2008). A new set of three standards (one primary and two secondary) were run after every third sample. It is not possible to sample the headspace repeatedly using this technique, as 1 L headspace only allows sampling once (60-90 seconds). Repeated inflating of the same sample with dry air results in incorrect readings. Following each set of samples and standards, the analysis was suspended for a period of around 10-15 minutes, to allow the LGR to reach the stable atmospheric air

30   readings and reduce any memory effect. Linear regression for $\delta^{18}O$ and $\delta^2H$ was established between the liquid values for standards and raw headspace vapour (fractionation factor at 25°C) readings for the same standards. Regression was used to calibrate the vapour results for samples. Calibration was undertaken with two secondary $\delta^{18}O$ and $\delta^2H$ standards (Los Gatos 2A -16.14 ‰ $\delta^{18}O$ and -123.6 ‰ $\delta^2H$ and 5A -2.80 ‰ $\delta^{18}O$ and 9.5 ‰ $\delta^2H$) and normalised with one primary

VSMOW/VSMOW2 standard run during the analysis. LGR standards were stored in accordance with the protocol, at 4°C, and ampules were fully sealed to prevent any exchange with the atmosphere. The use of LRG standards as secondary standards has been used in other studies such as Penna et al (2010) on reproducibility and repeatability of the laser absorption spectroscopy measurements and was found that LGR standards performed satisfactorily.

5 Replicate sample analyses using direct vapour equilibration method (mean difference of 6 samples) indicate reproducibility of results in our research within an uncertainty of 0.68 ‰ for $\delta^2H$ and 0.04 ‰ for $\delta^{18}O$. Reported instrument precision of 0.5 ‰ $\delta^2H$ and 0.15 ‰ $\delta^{18}O$ over 10 seconds and drift of 0.75 ‰ $\delta^2H$ and 0.3 ‰ $\delta^{18}O$ over 15 minutes was minimised by correcting the readings. The data set for each sample was corrected for drift by back correction using standards within each set and then applying the same regression analysis to the relevant samples. For each sample the standard deviation and

10 instrument drift error were calculated. Following the standard operating procedures, the precision in this research was 0.6 ‰ for $\delta^2H$ and 0.23 ‰ for $\delta^{18}O$ over 60 seconds.

Hendry et al (2015) report the analytical precision of the vapour equilibration method (±0.40 ‰ for $\delta^{18}O$ and ±2.1 ‰ for $\delta^2H$) to be comparable or better than physical extraction from cores using high-speed centrifugation, cryogenic micro-distillation, azeotropic and microwave distillation or isotope ratio mass spectrometry (IRMS) based direct equilibration

15 methods as discussed in Kelln et al (2001). Based on work by Allison and Hughes (1983) and Revesz and Woods (1990), the direct vapour equilibration method precision is also better than for methods obtained by chemical water extractions (Hendry et al, 2015). This is achieved through limiting fractionation losses by short storage time, single procedure once the samples are in the laboratory and use of standards and water isotopic data as a cross check.

Water samples (surface water, swamp groundwater and regional groundwater, n=21) were analysed for $\delta^{18}O$ and $\delta^2H$ by the

20 off axis- integrated cavity output spectrometry (OC-ICOS) technique using an LGR analyser located at UNSW Australia. Two secondary standards and VSMOW/VSMOW2 standard were used to calibrate and normalise the samples.

Gravimetric water content (ASTM D2974-14, 2014 and ASTM D2216-10, 2010) was measured by weighing the sediment samples (n=70), drying at 100°C for 24 hours and re-weighing (Reynolds, 1970). 100% gravimetric water content relates to water holding capacity and organic content of the material. The analysis was undertaken at the School of Mining

25 Engineering, UNSW Australia. Organic matter content was measured by loss on ignition method (LOI), by weighing (following initial drying at 100°C) and drying in furnace oven at 550°C (Heiri et al, 2001). The analysis was conducted at the Water Research Laboratory, UNSW Australia.

The vapour equilibration method is sensitive to presence of volatile organic compounds (VOC) in samples (Millar et al, 2018) and may require spectral correction post analysis, however usually the presence of these hydrocarbons is not known

30 (Hendry et al, 2015). There are limited studies related to quantification of VOC in peat (Mezhibor and Bonn, 2014) and impact of organic matter on isotopic composition (Orlowski et al, 2016). For water samples, the LGR's post analysis software automatically applied check for spectral interference. Similar approach was reported by Millar et al (2018), Schultz et al (2011) and Orlowski et al (2018). There was no evidence of spectral contamination. However, this analysis was not possible for vapour analysis of soil samples due to the different processing method. The similarity between the isotopic

composition of water and vapour collected from the same horizon suggests a strong interaction between groundwater and porewater; as the groundwater analyses show no evidence of VOC contamination, there is no reason to suspect that the porewater samples would contain concerning levels of VOCs despite the high peat organic content. Furthermore, in one of the rare VOC quantification studies on peat, Mezhibor and Bonn (2014) found that peat had mean concentration of isoprene

5  and acetaldehyde (VOCs characteristic for natural plant organic emission) up to 0.26 ppb. The ecosystem in their study is similar to this one and for such low volume % concentrations the spectral corrections are not considered to be necessary.

Precipitation samples were analysed at the ANSTO Environmental Isotope Laboratory using a cavity ring-down spectroscopy method on a Picarro L2120-I Water Analyser (reported accuracy of $\pm1.0$, $\pm0.2$ ‰ for $\delta^2$H and $\delta^{18}$O respectively). The lab runs a minimum of two in-house standards calibrated against VSMOW/VSMOW2 and SLAP/SLAP2

10  with samples in each batch.

For simple statistical analysis of moisture content, precipitation and organic matter content, an XLStat software package (XLStat, 2017) was used. The Barnes and Allison (1988) model was implemented for this project using R (R core team, 2013), to investigate the evaporative losses based on isotopic composition of water. For the Barnes and Allison (1988) model volumetric water content was calculated from the measured gravimetric water content and bulk density. Bulk density was

15  obtained from known lithology and measured data (Cowley et al, 2016) and porosity data from a swamp study by Walczak et al (2002). To estimate effective liquid diffusivity of isotopes, particle size and tortuosity values were obtained from the literature (Maidment, 1993; Shackelford and Daniel, 1991; Barnes and Allison, 1988).

**4. Results**

**4.1.  Stratigraphy, organic matter and moisture content**

20  Four stratigraphic units are recognised along both Newnes Plateau swamp transects CC, GG and GGSW (**Fig. 3 to 5**), similar to a general classification derived by Fryirs et al (2014) for T~P~HP~S~S in Blue Mountains and Southern Highlands region. The cross-sections presented in Fig. 3 to 5 were prepared on the basis of logged cores extracted as part of this research. These units are typically from the base upward medium to coarse sand, medium sand to clayey sand, silt to sandy clay and organic rich soil (sandy) at the top.

[Figure]

**Figure 3: Interpreted long–section of swamp GG with swamp groundwater levels as measured in May 2016.**

[Figure]

**Figure 4: Interpreted long–section of swamp CC with swamp groundwater levels as measured in May 2016.**

[Figure]

**Figure 5: Interpreted long–section of swamp GGSW with swamp groundwater levels as measured in October 2016.**

The base of the swamp is comprised of quartz sandstone, the Banks Wall Sandstone of the Narrabeen Group. The alluvial
5    sands (with sub-angular quartz grains) overlying the sandstone are off-white opaque to transparent, medium to coarse
grained with occasional quartz grains up to 2.5 mm in diameter. The term ''sub-angular'' defines the roundness of quartz or
any other sediment grain. This is important as it points to material transport information, angular grains have been subject to
limited transport.

These sands are overlain by medium sand grading to fine sand in the GG transect, with 15% organic matter and a minor clay
10    component. However, in the CC transect this layer is missing and sand transitions upwards to clayey sand with iron staining.
The total thickness of these two sandy units varies from 10 to 50 cm increasing in downgradient direction. At the most
downgradient site on GG transect the sand layer is absent. Typically, the basal sand is overlain by a silt and silty clay that is
thickest in the middle of the swamp (20–45 cm). The silt is dark grey in colour and contains approximately 40% organic
matter with strong- organic smell. Organic smell relates to  high percentage of organic matter (peat). The uppermost
15    unit is an organic rich soil or peat (20-60 cm thick), occasionally silty with abundant roots.

The swamp groundwater level is shallow, and it varied in piezometers (installed to 1.5 m depth) in May 2016 from 0.35 m
below ground level (bgl) in GGEG2 to 0.47 m bgl in CCG1 (**Fig. 3 and 4**). The swamp groundwater level in cored holes was
similar to that in shallow piezometers, however there was a significant difference represented by a rise of up to 0.4 m at all
measured locations following the wetter period. The initial rise is mainly attributed to rainfall. During this wetter period,
20    swamp groundwater levels recorded at GGSWG1 and GGEG2 were 0.05 m bgl and 0.09 m bgl, respectively.  No overland
flow was observed at any time, and the swamps did not have a formed channel. The only surface water observed in the
swamps was at the lower edge of the swamp and flowing over the rockbar.

The swamp sediments are variably saturated, with gravimetric water content measurements exceeding 100% weight (dry
mass basis) in the top 30 cm. This is typical for high organic matter proportion (GG samples) (**Fig. 6**). Within the same

vertical profile, the organic matter content varied with depth and decreased from 60 % to 10 %. At a depth from 60 to 120 cm the gravimetric water content decreased to an average of 17 % for CC and 32 % for GG swamp during both May and Oct 2016 sampling periods. The average organic matter decreased to 3.7% for all swamp locations below 80 cm depth.

During May 2016, following the dry period, upgradient and downgradient samples in CC swamp had similar gravimetric water content. A clear distinction was observed after wet weather period between the upgradient CCG2, having overall lower gravimetric water content, and downgradient CCG3, with higher gravimetric water content. A trend with an increase in moisture content downstream has been observed in all three swamps. However, at GGEG, the undulating topographic gradient means that changing moisture conditions exist along the length of the swamp. An overall increase in moisture content to around 80 cm depth in CCG3, was also recorded following the wet weather period although the increase was not statistically significant ($p>0.05$).

[Figure]

Figure 6: Gravimetric water content (% weight) for May16 (a), Oct16 (b) and May17 (c) and organic matter content in CC and GG swamps (d) shown with depth.

**4.2 Stable isotopes of water and pore water**

The relationship between surface water, swamp groundwater, regional groundwater and swamp pore water $\delta^{18}$O and $\delta^2$H data is presented in **Fig. 7**. This figure also shows the local meteoric water line (LMWL) for Lithgow ($\delta^2$H=7.99$\delta^{18}$O+16.6; Hughes and Crawford, 2013) and weighted rainfall average for Mt Werong which is based on the past 12 years of data ($\delta^{18}$O=-6.87 ‰, $\delta^2$H=-37.3 ‰). The $\delta^{18}$O of rainfall varies seasonally with higher values in summer and lower in winter. Stable isotope data from precipitation events at Mt Werong  are plotted (excluding the rainfall below 5 mm) for three periods (Jan to May 2016, May to October 2016, and Jan to May 2017). The stable isotope data for these events plot on or close to previously defined LMWL for Lithgow (note that the LMWL for Mt Werong of $\delta^2$H=8.08 $\delta^{18}$O+16.6 (Hughes and Crawford, 2013) and has a similar slope but higher intercept than that for Lithgow.

[Figure]

Figure 7: Stable $\delta^{18}$O and $\delta^2$H composition of surface water, swamp groundwater, regional groundwater, swamp pore water, weighted rainfall average for Mt Werong (2005-2017) and LMWL for Lithgow (Hughes and Crawford, 2013) May 2016 (a), October (2016) (b) and May 2017 (c).

For May 2016 with dry and warm antecedent conditions, pore water stable isotope ranges  are -7.20 to 3.10 ‰ $\delta^{18}$O and -45.7 to -22.3 ‰ $\delta^2$H (**Fig. 7a**). Pore water samples at CC and GG in May 2016 were clearly evaporated, lying along lines with slopes of 4.2 and 4.6 respectively, even though no single initial value for porewater evaporation was discernible.  Two major rainfall periods (27 mm two weeks prior to May 2016 sampling and 153.5 mm in January 2016) had no noticeable influence on the swamp pore water isotope composition. The intersection points of the regressed trend lines of pore water and LMWL plot within the  lower $\delta^{18}$O and $\delta^2$H rainfall range.

Stable isotopes for swamp pore water collected in October 2016 (**Fig. 7b**) range from -4.50 to -7.50 ‰ $\delta^{18}$O and -25.0 to -47.0 ‰ $\delta^2$H. Pore water stable isotope values from samples collected in the wet and cool antecedent conditions, plot along the LMWL very close to the weighted rainfall average. This is consistent with a winter rainfall signature.

The pore water samples collected in May 2017 from GGSW swamp lie along a slope of 6 which aligns with a wetter period in early 2017 compared to 2016. Samples from CC swamp collected in May 2017 are more enriched in  (i.e. have a higher D-excess (*d*), defined as *d*= $\delta^2$H-8$\delta^{18}$O (Dansgaard, 1964)) than previously collected samples indicating greater evaporative influence. Rainfall samples for bigger rainfall events in the period from December 2016 to May 2017 plot along the LMWL, except events in the April prior to the 2017 sampling which have a significantly higher D-excess (*d*=24.5 ‰, rainfall of 68 mm). The pore water returned to the LMWL between May and October 2016 and shifted to the left of the LMWL for the May 2017 sampling.

Swamp groundwater samples collected in October 2016 and May 2017 are enriched in $^{18}$O and $^2$H relative to the rainfall weighted average for Mt Werong (2005-2017). Surface water samples collected mainly at the downstream point of the swamp plot close to the LMWL and are  lower in $\delta^{18}$O and $\delta^2$H relative to pore water samples , and relative to large rainfall events preceding the sampling event.

Surface water samples (-6.50 to -7.70 ‰ $\delta^{18}$O and -37.0 and -44.4 ‰ $\delta^2$H) plot within the range of $\delta^{18}$O and $\delta^2$H for swamp groundwater samples (-6.2 to -7.9 ‰ $\delta^{18}$O and -32.4 and -44.7 ‰ $\delta^2$H). The statistical significance of the difference between the isotopic composition of surface water and swamp groundwater on both GG and GGSW transects was analysed by comparing the means of $\delta^{18}$O and $\delta^2$H (October 2016 and May 2017) for these two datasets using a *t*-test. Based on the mean, we test the hypothesis that there is no statistical difference between the datasets (surface water and swamp water). The calculated *p*-value was significantly more than 0.05 (for $\delta^{18}$O p=0.34 and for $\delta^2$H p=0.27;(n=20), indicating that the null hypothesis cannot be rejected and there is no significant difference between these two datasets.

[Figure]

[Figure]

**Figure 8:** $\delta^{18}O$ and $\delta^2H$ variation with depth in GG and GGSW swamps (May 2016) with typical lithology log. Regional groundwater sample was collected at the downstream point of the GG swamp (a and b). $\delta^{18}O$ and $\delta^2H$ variation with season and depth in CC swamp (May and October16 and May17) with typical lithology log. Swamp groundwater represents cumulative water through the swamp within shallow piezometers and cored holes (c and d). Swamp groundwater samples were not collected at all locations in May 2016 due to dry conditions. Depth of augured holes was not exactly the same in all sampling events.

The $\delta^{18}O$ and $\delta^2H$ data for pore water is plotted with depth along with surface water and groundwater from GG and GGSW swamps (**Fig. 8a and 8b**). Seasonal pore water and swamp groundwater variations (May and October 2016 sampling) for CC swamp are compared to rainfall isotopic signature collected at Lithgow (Hughes and Crawford, 2013) (**Fig. 8c and 7d**). The $\delta^{18}O$ values of pore water (May 2016) in GG and GGSW swamps (**Fig. 8**) show a tendency of depletion with depth with greater variability at a depth of 40-65 cm. Below 100 cm depth, the $\delta^{18}O$ values of pore water approach the swamp groundwater and regional groundwater signature.

It can be observed that pore water samples from CC swamp from both upstream (location CCG2) and downstream (location CCG3) have lower $\delta^{18}O$become depleted after longer wet and cool antecedent conditions with a $\delta^{18}O$ shift of around 1-3 ‰ (**Fig. 8c and 8d**). $\delta^{18}O$ and $\delta^2H$ for pore water at CCG2 during May and October 2016 shows a statistically significant difference between the wet and dry period ($\delta^{18}O$ (p=0.003) and $\delta^2H$ (p=0.02)), similar to $\delta^{18}O$ of pore water at CCG3 (p=0.01). The CC samples collected in May 2017 have lower $\delta^{18}O$ values of pore water compared to October 2016 samples and are similar to significant rainfall in March 2017.The CC samples collected in May 2017 have similar $\delta^{18}O$ values of pore water to October 2016 samples however are slightly more depleted and similar to significant rainfall in March 2017.

Swamp groundwater samples collected from piezometers screened across both top of sandstone and the base of swamp sediments (CCG1 and GGEG2) had a similar $\delta^{18}O$ signature to pore water at a depth below 110 cm. Surface water $\delta^2H$ for October 2016 is more negative than the pore water value (-37.7 ‰ $\delta^2H$) in the upper 70 cm and is similar to the typical winter rainfall signature.

**4.3 Water balance**

During dry periods swamp pore water is subject to evaporation and becomes  enriched in $^{18}O$ and $^2H$. Therefore, the fractional loss of water through evaporation can be quantified if other water loss processes do not isotopically fractionate (Gonfiantini, 1986) or/and if the stable isotope composition of inflow and outflow and site weather data is known (Lawrence et al, 2007). To evaluate the evaporative losses based on isotopic composition of water, we used the Barnes and Allison (1988) analytical model to represent the change in isotopic profile in unsaturated soils due to evaporation. This model, based on deterministic approach, was selected because the stable isotopes diffusivities vary slowly with water content and a relatively good agreement is reported with experimental results (Barnes and Allison, 1988; Shanafield et al, 2015). The disadvantage of using the soil profile to estimate evaporation is that an assumption of steady state is needed and there is some uncertainty in dispersivity and tortuosity values (Shanafield et al, 2015). The support for the selection of the Barnes and Allison (1988) model for the vegetated wetland environment is shown in the recent work undertaken by Piayda et al (2017). They found that regardless of the presence of vegetation or bare soil, the total evapotranspirative water loss of soil and understorey remains unchanged. Furthermore, the modelling is considered to be applicable by focusing on vertical flow. Modelling included the samples from the base of the swamp (not sides) where vertical flow is dominant due to high permeability of the peat.

We applied the model to pore water data from all three sampling periods considering realistic input variables into the model as given in Table 1. The model ran with the evaporation factor adjusted such that it matched the observed data; all other parameters remain constant. A linear relationship was identified between particle size and tortuosity, and the final estimated tortuosity values are given in Supplement (**Table 1**).

The results for unsaturated soil modelling at all sampled depth points based on $\delta^{18}O$ and $\delta^2H$ indicate an evaporative loss in the unsaturated zone of 4 mm to 9 mm/day in May 2016 (dry) period, and <1 mm/day for the wetter and cooler period between May and October 2016. Evaporation of less than 1 mm was estimated in CC swamp in both wet and dry periods, and at the upstream point on GGSW swamp.

The model was not sensitive to temperature; modelling at both 21.9°C and 10°C resulted in only minor differences in evaporation (<0.04 mm/day). Model results for drier and wetter periods are presented in Supplement (**Fig. 1**). The data for the May 2016 period (dry) shows a clear evaporative enrichment profile towards the surface (upper 0.4 to 0.6 m) and uniform $\delta^2H$ with depth (Supplement, **Fig. 1a and 1b**). No changes in isotopic composition  are observed below a depth of 0.6 m.

The water balance was prepared such that it incorporates the following parameters: rainfall, runoff from each of the swamps, evaporation. The deficit in the water balance  is attributed to groundwater contribution. Two options are considered in the water balance with respect to evaporation: evaporation based on unsaturated soil model results (E) and reference data (ET$_c$). The rainfall data from Lithgow BoM Station 63132 and reference evapotranspiration (ET) data from Nullo Mountain

5    BoM Station 62100 (94 km north of the study site in the same mountain range and similar elevation and climate) indicate that in the dry period (February to May 2016) the ET significantly exceeded the rainfall (Table 1). The ET represents evapotranspiration computed from reference surface (grass) using meteorological data (Allen at el, 1998). Crop evapotranspiration (ET$_c$) calculation incorporates the ground cover, canopy properties and aerodynamic resistance for the specific crop into the calculation. In our case the ET$_C$ is applied to a wetland system.

**Table 1: Water/mass balance components: measured rainfall and ET data (Lithgow and Nullo Mountain), runoff, measured E and estimated regional groundwater contribution (negative values are regional groundwater contribution).**

**Figure 9** shows the water deficit and estimated regional groundwater contribution to each of the swamps for the ET$_C$ and E

15    methods. The relative regional groundwater contribution is dominant in the dry weather period when it exceeds total rainfall. This regional groundwater contribution range represents the minimum and conservative value given that discharge from the swamp is not included in the water balance and that the estimates based on the E method do not include transpiration losses.

[Figure]

**Figure 9: Water balance during dry period estimated using ETc and E for each of the swamps.**

**Discussion**

**5.1 Swamp stratigraphy, geomorphology and groundwater condition**

5    Swamp sediments are thin (less than 1.5 m) and are deposited directly on the sandstone basement. Typically, the organic soil or peat is 40-60 cm thick, underlain by unconsolidated alluvial sand and sandy silt with organic rich thin bands. The geomorphology of the Newnes swamps is consistent with the intact swamp classification as reported by Fryirs et al. (2016), and with moisture and organic matter content as reported in Blue Mountain swamps by Cowley et al. (2016). The lithology indicates that the sediment transport is alluvial, however limited and occurring over relatively short distances (length of the

10    swamp).

An important finding of this research is that no evidence was observed for a clay rich layer with sealing properties at the base of these swamps. A conceptual model of swamp sediments that are hydraulically connected with the underlying sandstone is proposed (**Fig. 10**). However, there is likely to be a decrease in permeability at this interface. A degree of hydraulic connection between the regional groundwater and these elongated gentle gradient shrub swamps (50 mm/m average (Cardno,

15    2014) is further supported by gravimetric water content results. The stable gravimetric water content below 0.4 m depth in

CC and 0.6 m depth in GG and GGSW swamps, indicates stable saturated conditions likely supported by lateral groundwater inflow.

Groundwater levels in the swamps were observed to be similar to regional groundwater level within the underlying sandstone (monitoring screen at a depth of around 10 m bgl) at the downstream end of GG swamp indicating that these two units could be hydraulically connected. Typically, the swamp groundwater levels in THPS (CC swamp) rise and decline in response to rainfall recharge (Centennial Coal, 2016) with very little lag time. Rapid infiltration and discharge in the swamp groundwater system is indicated by low swamp groundwater salinity (measured in this study) (David et al, 2018). Given high moisture and organic matter content and evidence of seasonal precipitation in $\delta^{18}O$ and $\delta^2H$ profiles (p<0.05) in the upper swamp horizons, we conclude that in this zone the high water holding capacity increases residence time following the initial infiltration (vertical swamp groundwater flow). The $\delta^{18}O$ and $\delta^2H$ of pore water in this variably saturated zone exhibits summer evaporation trends and a winter rainfall signature. The lateral groundwater discharge to the swamp is characterised by longer residence time compared to water exchange through the swamp based on  lower $\delta^{18}O$ values and minor change between the sampling events. The similarity in EC and pH values between surface and swamp groundwater (David et al, 2018) further supports relatively rapid infiltration and possibility of both lateral and upward local groundwater inflow that provides baseflow to the swamp. However, local differences in swamp strata do exist: e.g. the difference in gravimetric water content in the CC swamp between the upgradient and downgradient location. This difference can be explained by higher permeability in the upgradient part of the swamp resulting in quicker drainage, increased groundwater contribution in the lower part of the swamp and/or lateral throughflow.

[Figure]

Figure 10: Conceptual representation of water dynamics in the swamp system.

To validate this conceptual model, a simple water balance was completed based on the evaporative losses estimated by analytical model (Barnes and Allison, 1988). Using the results from the dry weather period February to May 2016, we obtain evaporation estimates ranging from 1 to 9 mm/day. The evaporation occurs in the top 0.4 m of the vertical profile, with an absence of fractionation below this depth where pore water isotopes values are similar to swamp water and regional

5    groundwater. These evaporation rates (1 to 9 mm/day) suggest high evaporation compared to rainfall in the same time period (**Table 1**). During the wet period (Supplement **Fig. 1c and 1d**) we observe the lower $\delta^2$H  at the surface. this  is related to a big rainfall event, 10 days before sampling (**Fig. 7c**).

With an $ET_c$ ranging from 1.7 to 4.4 mm/day and E ranging mainly from 4 to 9 mm/d, the $ET_c$/E ratio for these swamps would be 0.7 to 0.3. This ratio is at the lower end of measured $ET_c$/E ratio for typical wetlands indicating that reference ET

10   could underestimate that based on realistic evaporation rates obtained by matching the modelled to observed data. The $ET_c$ for typical wetland vegetation (sedge) in temperate climates ranges from 0.8 to 1.2 (Allen et al, 1998; Mohamed et al, 2012) and 0.7 was reported in a swamp in the Murrumbidgee, Australia (Linacre et al, 1967). The $ET_c$ in our case is less than the estimated E based on stable isotope data. As transpiration does not fractionate, the actual evapotranspiration in the dry and warm period would have to be greater than the estimated evaporation. This would result in higher water balance losses,

15   requiring more water be supplied from other sources.

Runoff represents only a small component of the water budget for several reasons. Firstly, the 10% slope gradient of the ridges, 3% slope gradient along the swamp floor and densely vegetated sides and base of the swamp minimize the runoff significantly. Secondly, the upper soil layer is peat with significant water holding capacity compared to other soil types, and as indicated by the gravimetric water content measured in CC and GG swamps.

20   A simple mass balance comprising the rainfall (input), runoff (input) from the catchment considered two different approaches in dry period using E or $ET_c$. When $ET_c$ (output), was used, March had excess water with a deficit in February, April and May of between 10 and 60 mm. However, the same mass balance calculated with E using 4 mm/day, has water deficit of between 10 and 113 mm/month for any month. If E of 9 mm/day is used in the water balance, water deficit occurs in every month in the range from 10 to 260 mm/month (Table 1). Either way, two important output components are not

25   considered in this mass balance:  transpiration and discharge at the rockbar downgradient of the swamp. The estimation of these two components is uncertain, but inclusion in the water balance would increase the water deficit further. Importantly, if swamp discharge data were available in combination with pore water isotope profiles, an appropriate crop transpiration could be determined for these swamps, a factor that is typically a large unknown in water balance studies.

It is evident that given the water deficit, even without two output components, an additional water source must have

30   maintained the swamp groundwater levels. We therefore conclude that groundwater is a significant contributor to swamp water balance, particularly during dry periods. For example, in GG swamp the swamp groundwater levels are in the range from 0.28 to 0.38m bgl and if groundwater inflow was not occurring under same evaporation conditions the depth to water in the swamp would be greater.

Furthermore, measured loss of moisture as shown in **Fig. 6** indicates that significant loss occurs in such a dry weather period in the top 40 cm (up to 150 % by weight), while lower parts of the swamp remain saturated. The estimate of groundwater contribution in the drier period (February to May 2016) ranges from 10 to over 113 mm/month if calculated using 4 mm/day of evaporation, and up to -260 mm/day if E of 9 mm/day is used. The water balance was undertaken for dry period only as

5    evaporation from soil profile using stable isotopes  is considered to be most accurate during that period. Thus, even in the months where water balance is positive, i. e. groundwater contribution is likely, as evident from discharge at the rockbar observed at the end of dry period. Although there is a compelling explanation for significant groundwater contribution to the swamp water balance, the actual volume of groundwater cannot be estimated without knowledge of swamp groundwater and regional groundwater recession rate and/or measurement of discharge from the swamp.

10   It is clear that the water balance in swamp can be obtained if all components are known (rainfall, runoff, groundwater contribution, evaporation, evapotranspiration, discharge), however due to the THPSS swamps being difficult to access and being protected under State and Federal legislation it is not possible to undertake intrusive drilling to obtain all hydrogeological information. The application of stable isotopes has enabled estimation of evapotranspiration from the swamp and assisted in development of the conceptual model.

**5.2 Swamp groundwater and regional groundwater movement within the swamp system**

The vertical depth profiles of pore water $\delta^{18}O$ and $\delta^2H$ can provide time series information by tracing the influence of the rainfall isotopic signature in recharging water. Pore water direct vapour equilibration method is used  in a

20   swamp environment and results compared with end members which included surface water, rainfall and groundwater. Although stable isotope data in precipitation changes in the short term, this end member is well constrained based on the good quality dataset for precipitation. Constraining the interpretation of isotope results with these end members enabled groundwater inputs to be identified.

The evaporation response in the upper 40 cm is consistent with depth of penetration dependent on evaporation rate, soil type

25   and time between rainfall events (Mathieu and Bariac, 1996; Melayah et al, 1996; dePaolo et al., 2004). As evaporation proceeds, capillary rise of swamp groundwater reduces the $\delta^{18}O$ enrichment closer to the surface. Moisture content data reveals variability at 30-70 cm depth, which is also observed in $\delta^{18}O$ and $\delta^2H$ profiles and is related to interlayering of fine and coarser grained material, consistent with other studies (dePaolo et al., 2004). The  pore water regression line intercepts  the LMWL  lower $\delta^{18}O$ and $\delta^2H$ value  than weighted average rainfall. The

30   isotope signature in the partially saturated zone (variable from 0.05 m bgl to 0.4 m bgl in the swamp) in the summer period (May 2016 sampling event) is a result of evaporation as observed from depth profiles and moisture content. This agrees with numerical experiments conducted  by Benettin et al (2018) where the soil water samples trend lines were found to be products of seasonality of evaporative fractionation. Swamp pore water in May 2016  has lower $\delta$ values

than rainfall and therefore likely to be from bigger, more isotopically depleted events in the autumn and winter of the prior year which are lower in $\delta^{18}O$ and $\delta^{2}H$ (including 230 mm in April 2015: -7.60 ‰ $\delta^{18}O$, -39.2 ‰ $\delta^{2}H$; 108 mm in August 2015: -9.8 ‰ $\delta^{18}O$, -61.6 ‰ $\delta^{2}H$; and two smaller but highly $^{18}O$ and $^{2}H$ depleted events in June and July). This agrees with annual weighted averages at Mt Werong of -34.9 and -46.5 ‰ $\delta^{2}H$ in 2015 and 2016, respectively. A major rainfall event in

5   June 2016 (92.8 mm at Lithgow and 109 mm at Mt Werong, -17.70 ‰ $\delta^{18}O$, -126.3 ‰ $\delta^{2}H$) had not obviously affected swamp pore water.

Although the same rainfall events generally affect both Mt Werong and Newnes, and occur at the same time, the amount of rainfall at Newnes is typically smaller than at Mt Werong. Whilst we would expect that larger rainfall events would lead to the most significant infiltration and recharge of swamp groundwater and regional groundwater, and therefore influence the

10   pore water signature more, the data seems to suggest that small recent rainfall events are very important in October 2016 and May 2017, following the wetter conditions experienced in the second half of 2016 and early 2017. The importance of smaller rainfall events on recharge is also consistent with gravimetric water content data which remained stable throughout the wetter and drier period at depth below 0.8 m in CC and 0.6 m in GG and GGSW swamps. Another contributing factor may be that groundwater provides a moderating effect particularly during wetter periods, reducing the effects that evaporation has

15   on pore water isotope composition.

Statistically there is no difference between the mean of the surface water and swamp groundwater stable isotope samples for GG and GGSW swamp. The reason for similarity of surface and swamp groundwater samples is assumed to be short infiltration time to water table and/or mixing with lateral regional groundwater with surface water sample points being located largely in the groundwater discharge zone. There is a difference in $\delta^{18}O$ and $\delta^{2}H$ values (p<0.05, n=18) between

20   samples collected after dry and warm versus wet and cool antecedent conditions. The October 2016 (cool weather) samples from CC swamp are typically depleted lower in $\delta^{18}O$ and $\delta^{2}H$ and we conclude that these values are within the range of winter rainfall isotope values. Below 100 cm depth the pore water values of $\delta^{18}O$ remain uniform and consistent with the regional groundwater value but also with surface water. We infer this to represent swamp groundwater derived from vertical infiltration and laterally from sandstone, respectively. We therefore consider the main processes to be rapid infiltration

25   through the swamp sediments to water table but at the same time high water retention in the upper horizons, and slow lateral exchange of pore water below the vadose zone.

The vertical topographic difference from swamp headwaters to the downstream end of the swamp (typically a sandstone rockbar) is around 40 m. This elevation difference is too small to result in any difference in isotopic signature of precipitation, therefore, given the spatial response and assuming a homogeneous environment with vertical flow, pore water

30   $\delta^{18}O$ and $\delta^{2}H$ should be similar (Garvelman et al, 2012). However, observed variation in profiles is not uniform, and is caused by vertical rainfall infiltration in the upper part of the profile and lateral flow at the base. The lateral flow within the swamp sediments is further enhanced by regional groundwater flow contribution from the valley sides. Such lateral flow is reported in these swamps where sandstone is underlain by a claystone layer (Corbett et al, 2014).

Factors such as fine-grained content of lithological units, reported by other studies (dePaolo et al., 2012), have been found to result in a bigger shift to  lower $\delta^{18}O$ and $\delta^2H$ values and variation in isotope signature with depth. The reason for this is related to hydraulic conductivity of the unconsolidated soil. For example, the biggest variation in $\delta^{18}O$ and $\delta^2H$ was observed in silt and clayey sand units (**Fig. 8a** and **8b**) which contain higher percentage of particles <2 µm. Contrary to

5     observations by Garvelman et al (2012), we did not find the variability in $\delta^{18}O$ and $\delta^2H$ to be a result of soil saturation and depth of vadose zone only, but also as a function of lithology and different grain size material (peat, organic soil with sand and silt). Variations in particle size, porosity and permeability would then influence groundwater flow and storage.

**Conclusion**

The hydrogeological and isotopic characterisation of these swamp environments provides a baseline understanding for future

10     comparison of any hydrological changes due to natural or human activities. This study  applies the vapour equilibration method for determining stable isotopes of pore water in a wetland system. This unique pore water isotope approach combined with other data and information has significantly improved a conceptual model of wetland hydrology. As found by Wassenaar et al (2008), the pore water stable isotope method allows efficient sample collection without permanent disturbance, collection of vertically discretized data at any practicable frequency and without the need for more

15     complex methods of water extraction.

This study found, for several upland peat swamps, that swamp groundwater is a dominant component of the water balance, its contribution being larger than rainfall during dry weather period. This finding is consistent with environmental tracer studies suggesting that 19-80 % of water in Blue Mountains swamps is from groundwater, particularly in steeper and rounder catchments (Young, 2017). Furthermore, these swamp groundwater systems appeared to be in hydraulic connection with the

20     underlying sandstone regional groundwater, given similar groundwater levels, and the lack of a clayey layer at the base of the swamp. Although rainfall infiltration to water table occurs rapidly, the high water holding capacity of upper organic rich layers maintains the moisture for long periods. These processes are confirmed by the results of the water balance, in particular during dry periods. The majority of flow through the swamp system is via lateral groundwater flow where flow rate depends on heterogeneity within this layer and hydraulic conditions. Under natural intact conditions, upward or

25     downward flow between the swamp system and underlying rock is controlled by groundwater heads, the slope, and the hydraulic conductivity contrast at the interface.

The conceptual model presented here provides a valuable benchmark from which to evaluate potential changes in swamps following underground mining and forestry activity. The improved understanding in the water balance in these swamps also has implications in other areas of the Blue Mountains where urbanization has a significant impact on upland swamps. The

30     role that catchments have on the health of a swamp is important in supporting its flora and fauna, with groundwater likely to be a primary factor that contributes to the long-term survival of the ecosystem (Gorissen et al, 2017). The protection of this

ecological community is therefore dependent on maintenance of catchment stability and groundwater baseflow contribution if forestry activity and ground movement or deformation due to mining occur in the swamp catchment.

Measurement of pore water stable isotopes of peat and sediment within the swamp ecosystem provides direct information on the depth at which the evaporation occurs and understanding of the water cycle. Evaporation obtained from stable isotope direct equilibration method was found to be more realistic than reference evapotranspiration. In particular, based on current research of the water balance in wetland and swamp systems and ecology around the world, the application of this method could be beneficial to define water availability for flora and fauna in swamps where a thick organic soil/peat and sedimentary layer exists.

**Data availability**

The underlying research data can be accessed in the registry for research data repositories at https://www.dataarchive.unsw.edu.au/.

**Acknowledgement**

This research did not receive any grants from funding agencies in the public, commercial, or not-for-profit sectors. The authors would like to acknowledge the support of Centre for Water Initiative and School of Biological and Earth Sciences for assistance with sample analysis. Rainfall isotope analysis was funded independent by ANSTO. We thank Prof A. Baker and anonymous referees for providing constructive suggestions to improve this paper. We acknowledge Bob Cullen for collecting rainfall samples at Mt Werong, Barbara Gallagher, Jennifer van Holst (ANSTO) and Fang Bian (UNSW) for analysis of rainfall samples, the Sydney Catchment Authority for providing rainfall data, and Karina Meredith (ANSTO) for advice on evaporation modelling.

**Table 1: Water/mass balance components: measured rainfall and ET data (Lithgow and Nullo Mountain), runoff, measured E and estimated balance deficit (negative values are groundwater contribution).**

|  | Feb-16 | Mar-16 | Apr-16 | May-16 |
|---|---|---|---|---|
| Total monthly rainfall Lithgow station (SN63132) (mm/month) | 28.8 | 61.2 | 6.2 | 26 |
| Reference ET Nullo Mountain (SN62100) (mm/month) | 119.9 | 92 | 76.6 | 51 |
| Evaporation (pore water stable isotope profiles) mm/month | 117-267 | 123-273 | 120-270 | 123-273 |
| Runoff estimate |  |  |  |  |
| CC | 31 | 65.6 | 6.6 | 28 |
| GG | 25 | 53 | 5.4 | 22.5 |
| GGSW | 16.4 | 35 | 3.5 | 15 |
| Balance deficit (groundwater component) (ETc) |  |  |  |  |
| CC | -60.2 | 34.8 | -63.8 | 2.9 |
| GG | -66.2 | 22.1 | -65.0 | -2.5 |
| GGSW | -74.7 | 4.1 | -66.9 | -10.2 |
| Balance deficit (groundwater component) (4 mm) |  |  |  |  |
| CC | -56.3 | 2.8 | -107.2 | -70.1 |
| GG | -62.3 | -9.9 | -108.4 | -75.5 |
| GGSW | -70.8 | -27.9 | -110.3 | -83.2 |
| Balance deficit (groundwater component) (9 mm) |  |  |  |  |
| CC | -201.3 | -152.2 | -257.2 | -225.1 |
| GG | -207.3 | -164.9 | -258.4 | -230.5 |
| GGSW | -215.8 | -182.9 | -260.3 | -238.2 |